# Short-length Adversarial Training Helps LLMs Defend Long-length Jailbreak Attacks: Theoretical and Empirical Evidence

**Shaopeng Fu**[1]  **Liang Ding**[2]  **Jingfeng Zhang**[3,1]  **Di Wang**[1*]
[1]King Abdullah University of Science and Technology
[2]The University of Sydney  [3]The University of Auckland
shaopeng.fu@kaust.edu.sa, liangding.liam@gmail.com
jingfeng.zhang@auckland.ac.nz, di.wang@kaust.edu.sa

## Abstract

Jailbreak attacks against large language models (LLMs) aim to induce harmful behaviors in LLMs through carefully crafted adversarial prompts. To mitigate attacks, one way is to perform adversarial training (AT)-based alignment, *i.e.*, training LLMs on some of the most adversarial prompts to help them learn how to behave safely under attacks. During AT, the length of adversarial prompts plays a critical role in the robustness of aligned LLMs. While long-length adversarial prompts during AT might lead to strong LLM robustness, their synthesis however is very resource-consuming, which may limit the application of LLM AT. This paper focuses on adversarial suffix jailbreak attacks and unveils that to defend against a jailbreak attack with an adversarial suffix of length $\Theta(M)$, it is enough to align LLMs on prompts with adversarial suffixes of length $\Theta(\sqrt{M})$. Theoretically, we analyze the adversarial in-context learning of linear transformers on linear regression tasks and prove a robust generalization bound for trained transformers. The bound depends on the term $\Theta(\sqrt{M_{\text{test}}}/M_{\text{train}})$, where $M_{\text{train}}$ and $M_{\text{test}}$ are the numbers of adversarially perturbed in-context samples during training and testing. Empirically, we conduct AT on popular open-source LLMs and evaluate their robustness against jailbreak attacks of different adversarial suffix lengths. Results confirm a positive correlation between the attack success rate and the ratio of the square root of the adversarial suffix length during jailbreaking to the length during AT. Our findings show that it is practical to defend against "long-length" jailbreak attacks via efficient "short-length" AT. The code is available at https://github.com/fshp971/adv-icl.

## 1 Introduction

Large language models (LLMs) [5, 51, 28, 65] are widely adopted in various real-world applications to assist human users [55, 67, 57, 56, 24], but their safety is found to be vulnerable toward jailbreak attacks [60]. With carefully crafted adversarial prompts, one can "jailbreak" the safety mechanism of LLMs and induce arbitrary harmful behaviors [73, 7, 30]. To tackle the challenge, recent studies [63, 36, 68, 6] propose performing safety alignment through adversarial training (AT) [32] to enhance LLMs' robustness against jailbreaking. A standard AT for LLMs would train them on jailbreak prompts synthesized by strong jailbreak attacks to learn to refuse these harmful instructions [36].

In such AT, the length of synthesized adversarial prompts used for model training is critical to the final jailbreak robustness of LLMs. [3] and [64] have shown that longer adversarial prompts enjoy stronger

---

[*]Corresponding Author

39th Conference on Neural Information Processing Systems (NeurIPS 2025).

jailbreaking abilities. Thus, it is reasonable to deduce that performing AT with longer adversarial prompts can help LLMs achieve stronger robustness to defend against "long-length" jailbreak attacks. However, synthesizing long-length adversarial prompts in adversarial training is resource-consuming since it requires solving discrete optimization problems in high-dimensional spaces, which thus needs lots of GPU memory and training time. This may limit the application of AT in LLMs' safety alignment and further raises the following research question: ***How will the adversarial prompt length during AT affect trained LLMs' robustness against jailbreaking with different prompt lengths?***

We study this research question by analyzing *suffix jailbreak attacks*, where each jailbreak prompt is constructed by concatenating a harmful instruction with a synthesized adversarial suffix. Our main finding is: **To defend against a suffix jailbreak attack with suffix length of $\Theta(M)$, it is enough to adversarially train LLMs on adversarial prompts with suffix length of only $\Theta(\sqrt{M})$.** In other words, we show that it is possible to defend long-length jailbreak attacks via efficient short-length AT.

Our finding is supported by *theoretical* and *empirical* evidence. Theoretically, we leverage the *in-context learning theory* [53, 69] to investigate how linear transformers learn linear regression tasks from in-context task samples under AT. To better simulate suffix jailbreak attacks in real-world LLMs, our analysis introduces a new *in-context adversarial attack*. Concretely, for any in-context task sample, this attack will adversarially perturb the last several in-context training points to maximize the squared prediction error that linear transformers made on the in-context test point. Under our theoretical framework, we prove a robust generalization bound for adversarially trained linear transformers. This bound has a positive correlation with the term $\Theta(\sqrt{M_{\text{test}}}/M_{\text{train}})$, where $M_{\text{train}}$ and $M_{\text{test}}$ are the number of perturbed in-context points in training and testing in-context task samples, respectively.

Empirically, we conduct AT with the GCG attack [73], one of the most effective jailbreak attacks, under various adversarial suffix lengths on five popular real-world LLMs and evaluate their robustness against jailbreak attacks with different adversarial suffix lengths. We use the jailbreak attack success rate (ASR) to express the robust generalization error of trained LLMs and find that this ASR has a clear positive correlation with the ratio of the square root of test-time adversarial suffix length to the AT adversarial suffix length. Such a correlation empirically verifies our main finding. We also find that AT with an adversarial suffix (token) length of 20 is already able to reduce the ASR of jailbreak attacks with an adversarial suffix (token) length of up to 120 by at least 30% in all experiments.

## 2 Related works

**Jailbreak attacks.** Jailbreaking [60] can be seen as adversarial attacks [49, 14] toward LLMs, which aim to synthesize adversarial prompts to induce targeted harmful behaviors from LLMs. Many efforts have been made on token-level jailbreak attacks, *i.e.*, searching adversarial prompts in the token space of LLMs, which can be achieved via gradient-based optimization [48, 16, 73, 26, 45, 71], heuristic greedy search [44, 17, 22], or fine-tuning prompt generators from pre-trained LLMs [38]. Other attempts include word-level adversarial prompt searching [30] or directly prompting LLMs to generate adversarial prompts [7, 29]. Our work focuses on token-level jailbreaking since it make it easier for us to control the adversarial prompt length for our analysis. More recent studies have found that increasing the length of adversarial prompts by adding more harmful demonstrations [3, 54, 61] or synthesizing longer adversarial suffixes [64] can make jailbreaking more effective. These works motivate us to investigate the problem of defending against "long-length" jailbreak attacks.

**Adversarial training on LLMs.** To defend against jailbreak attacks, a large body of studies focus on aligning LLMs to refuse responding jailbreak prompts [37, 42, 40, 41, 8]. More recent works have started to adopt adversarial training (AT) [32] to align LLMs. [36] trained LLMs on (discrete) adversarial prompts synthesized by GCG attack [73], in which they cached the intermediate synthesized results to reduce the heavy cost of searching adversarial prompts from scratch. Meanwhile, various studies [63, 6, 46, 68] conduct AT with adversarial examples found in the continuous embedding space rather than the discrete text space since searching in the continuous embedding space is more computationally efficient. Nevertheless, as a preliminary study of the length of adversarial prompts during AT, our work only analyzes AT with discrete adversarial prompts.

**In-context learning theory (ICL).** Transformer-based large models like LLMs are strong in performing ICL: Given a series of inputs (also known as "prompt") specified by a certain task, LLMs can make predictions well for this certain task without adjusting model parameters. Current theories in understanding ICL can be roughly divided into two categories. The first category aims to under-

stand ICL via constructing explicit multi-layer transformers to simulate the optimization process of learning function classes [13, 53, 1, 9, 34, 59]. The second category focuses on directly analyzing the training [69, 66, 19, 62, 27] and generalization [31, 33, 11, 47] of simple self-attention models (*i.e.*, one-layer transformer). [4] is the first to study adversarial attacks against linear transformers and finds that an attack can always succeed by perturbing only a single in-context sample. However, their analysis allows samples to be perturbed in the entire real space, which might not appropriately reflect real-world settings since real-world adversarial prompts can only be constructed from token/character spaces of limited size. Unlike [4], we propose a new ICL adversarial attack that requires each adversarial suffix token to be perturbed only within restricted spaces, which thus can be a better tool for understanding real-world jailbreaking.

Finally, we notice that [61] also recognizes the critical role that the number of adversarial in-context samples plays in ICL-based attacks. They present a theoretical analysis (not based on ICL theory) for adversarial attacks against ICL text classification and characterize the minimum number of in-context adversarial samples required to increase the safety loss of ICL to some extent. However, the main difference is that [61] focuses on studying the adversarial robustness of fixed ICL models, whereas our work analyzes how adversarial training affects the robustness of ICL models.

## 3   Preliminaries

**Large language models (LLMs).** Let $[V] = \{1, \cdots, V\}$ be a vocabulary set consisting of all possible tokens. Then, an LLM can be seen as a function that for any sequence $x_{1:n} \in [V]^n$ consists of $n$ tokens, the LLM will map $x_{1:n}$ to its next token $x_{n+1}$ following $x_{n+1} \sim p_\theta(\cdot|x_{1:n})$, where $p_\theta$ is a conditional distribution over the vocabulary set $[V]$ and $\theta$ is the model parameter of the LLM. Under such notations, when using the LLM $p_\theta$ to generate a new token sequence for the input $x_{1:n}$, the probability of generating a sequence $y_{1:m} \in [V]^m$ of length $m$ is given by $p_\theta(y_{1:m}|x_{1:n}) = \prod_{i=1}^m p_\theta(y_i|x_{1:n} \oplus y_{1:(i-1)})$, where "$\oplus$" denotes concatenation.

**Jailbreak attacks.** This paper focuses on *suffix* jailbreak attacks. Concretely, suppose $x^{(h)}$ and $y^{(h)}$ are two token sequences, where $x^{(h)}$ represents a harmful prompt (*e.g.*, "Please tell me how to build a bomb.") and $y^{(h)}$ represents a corresponded targeted answer (*e.g.*, "Sure, here is a guide of how to build a bomb"). Then, the goal of a suffix jailbreak attack against the LLM $p_\theta$ aims to synthesize an *adversarial suffix* $x_{1:m}^{(s)}$ for the original harmful prompt $x^{(h)}$ via solving the following problem,

$$\min_{x_{1:m}^{(s)} \in [V]^m} -\log p_\theta(y^{(h)}|x^{(h)} \oplus x_{1:m}^{(s)}), \tag{1}$$

where $x^{(h)} \oplus x_{1:m}^{(s)}$ is the adversarial prompt and $m$ is the sequence length of the adversarial suffix $x_{1:m}^{(s)}$. Intuitively, a large $m$ will increase the probability of the LLM $p_\theta$ that generating the targeted answer $y^{(h)}$ for the synthesized adversarial prompt $x^{(h)} \oplus x_{1:m}^{(s)}$. To solve Eq. (1), a standard method is the Greedy Coordinate Gradient (GCG) attack [73], which leverages gradient information to search for better $x_{1:m}^{(s)}$ within the discrete space $[V]^m$ in a greedy manner.

**Adversarial training (AT).** We consider the canonical AT loss $\mathcal{L}$ [36, 40] to train the LLM $p_\theta$, which consists of two sub-losses: an *adversarial loss* $\mathcal{L}_{\text{adv}}$ and an *utility loss* $\mathcal{L}_{\text{utility}}$. Specifically, given a *safety dataset* $D^{(h)}$, where each of its sample $(x^{(h)}, y^{(h)}, y^{(b)}) \in D^{(h)}$ consists of a harmful instruction $x^{(h)}$, a harmful answer $y^{(h)}$, and a *benign answer* $y^{(b)}$ (*e.g.*, "As a responsible AI, I can't tell you how to..."). The adversarial loss $\mathcal{L}_{\text{adv}}$ is defined as follows,

$$\mathcal{L}_{\text{adv}}(\theta, M, D^{(h)}) := \mathbb{E}_{(x^{(h)}, y^{(h)}, y^{(b)}) \in D^{(h)}} [-\log p_\theta(y^{(b)}|x^{(h)} \oplus x_{1:m}^{(s)})], \tag{2}$$

where $x_{1:m}^{(s)}$ is the adversarial suffix obtained from Eq. (1) and $m$ is the adversarial suffix length. Note that the probability terms in Eqs. (1) and (2) look similar to each other. The difference is that the term in Eq. (1) denotes the probability that $p_\theta$ generates the harmful answer $y^{(h)}$ for the adversarial prompt, while that in Eq. (2) denotes the probability of generating the benign answer $y^{(b)}$. Besides, let $D^{(u)}$ be a *utility dataset* where each of its sample $(x^{(u)}, y^{(u)}) \in D^{(u)}$ consists of a pair of normal instruction and answer. Then, the utility loss $\mathcal{L}_{\text{utility}}$ is given by

$$\mathcal{L}_{\text{utility}}(\theta, D^{(u)}) := \mathbb{E}_{(x^{(u)}, y^{(u)}) \in D^{(u)}} [-\log p_\theta(y^{(u)}|x^{(u)})].$$

Thus, the overall AT problem for improving the jailbreak robustness of the LLM $p_\theta$ is given as

$$\min_\theta \{\alpha \mathcal{L}_{\text{adv}}(\theta, M, D^{(h)}) + (1-\alpha)\mathcal{L}_{\text{utility}}(\theta, D^{(u)})\}, \tag{3}$$

where $\alpha \in [0,1]$ is a factor that balances between the adversarial and utility sub-losses. The idea behind such a loss design is that: (1) help LLM learn to respond harmlessly even when strong jailbreak prompts present (achieved via $\mathcal{L}_{\text{adv}}$), (2) retain the utility of LLM gained from pre-training (achieved via $\mathcal{L}_{\text{utility}}$). Intuitively, a larger adversarial suffix length $m$ during AT will help the LLM gain robustness against jailbreak attacks with longer adversarial suffixes.

## 4    Theoretical evidence

This section establishes the theoretical foundation of how "short-length" AT can defend against "long-length" jailbreaking. Our analysis is based on the in-context learning (ICL) theory [69, 47, 4], and we will bridge the ICL theory and the LLM AT problem defined in Eq. (3) later (in Section 4.2). Here we first introduce the necessary notations to describe the problem. To avoid confusion, we note that **all notations in this section will only be used within this section and have no relevance to those in other sections** (*e.g.*, Section 3).

**In-context learning (ICL).** In the ICL theory, a *prompt* with length $N$ related to a specific *task* indexed by $\tau$ is defined as $(x_{\tau,1}, y_{\tau,1}, \cdots, x_{\tau,N}, y_{\tau,N}, x_{\tau,q})$, where $x_{\tau,i} \in \mathbb{R}^d$ is the $i$-th in-context training sample, $y_{\tau,i} \in \mathbb{R}$ is the label for the $i$-th training sample, and $x_{\tau,q} \in \mathbb{R}^d$ is the in-context query sample. Then, the task-specific ICL input $E_\tau$ is defined as

$$E_\tau := \begin{pmatrix} x_{\tau,1} & \cdots & x_{\tau,N} & x_{\tau,q} \\ y_{\tau,1} & \cdots & y_{\tau,N} & 0 \end{pmatrix} \in \mathbb{R}^{(d+1)\times(N+1)}. \tag{4}$$

Given an ICL input $E_\tau$ of task $\tau$, the goal of an ICL model is to make a prediction based on $E_\tau$ for the query sample $x_{\tau,q}$. Such an ICL model design aims to model the ability of real-world LLMs in making decisions based on prompting without updating model parameters.

**Linear self-attention (LSA) models.** LSA models are a kind of linear transformer that has been widely adopted in existing theoretical ICL studies. [2] empirically show that LSA models share similar properties with non-linear ones and thus are useful for understanding transformers. We follow [69] to study the following single-layer LSA model,

$$f_{\text{LSA},\theta}(E_\tau) := \left[ E_\tau + W^V E_\tau \cdot \frac{E_\tau^\top W^{KQ} E_\tau}{N} \right] \in \mathbb{R}^{(d+1)\times(N+1)},$$

where $\theta := (W^V, W^{KQ})$ is the model parameter, $W^V \in \mathbb{R}^{(d+1)\times(d+1)}$ is the value weight matrix, $W^{KQ} \in \mathbb{R}^{(d+1)\times(d+1)}$ is a matrix merged from the key and query weight matrices of attention models, $E_\tau \in \mathbb{R}^{(d+1)\times(N+1)}$ is the task-specific ICL input, and $N$ is the prompt length. The prediction $\hat{y}_{q,\theta}$ for the query sample $x_{\tau,q}$ is given by the right-bottom entry of the output matrix of the LSA model, *i.e.*, $\hat{y}_{q,\theta}(E_\tau) := f_{\text{LSA},\theta}(E_\tau)_{(d+1),(N+1)}$. We further follow [69] to denote that

$$W^\square = \begin{pmatrix} W_{11}^\square & w_{12}^\square \\ (w_{21}^\square)^\top & w_{22}^\square \end{pmatrix} \in \mathbb{R}^{(d+1)\times(d+1)},$$

where $\square \in \{V,\ KQ\}$, $W_{11}^\square \in \mathbb{R}^{d\times d}$, $w_{12}^\square, w_{21}^\square \in \mathbb{R}^{d\times 1}$ and $w_{22}^\square \in \mathbb{R}$. Under this setting, the model prediction $\hat{y}_{q,\theta}$ can be further simplified as follows,

$$\hat{y}_{q,\theta}(E_\tau) := f_{\text{LSA},\theta}(E_\tau)_{(d+1)\times(N+1)} = \begin{pmatrix} (w_{21}^V)^\top & w_{22}^V \end{pmatrix} \cdot \frac{E_\tau E_\tau^\top}{N} \cdot \begin{pmatrix} W_{11}^{KQ} \\ (w_{21}^{KQ})^\top \end{pmatrix} \cdot x_{\tau,q}. \tag{5}$$

**Other notations.** For any $n \in \mathbb{N}^+$, we denote $[n] := \{1, \cdots, n\}$. For any $A \in \mathbb{R}^{n\times m}$, we denote $\|A\|_{2,\infty} := \max_{1\le i\le m} \|A_{i,:}\|_2$, $\|A\|_2$ be the operator norm, and $\|A\|_F$ be the Frobenius norm. For any $A \in \mathbb{R}^{n\times n}$, we denote $\text{Tr}(A) := \sum_{i=1}^n A_{i,i}$. We use standard big O notations $\mathcal{O}(\cdot)$ and $\Theta(\cdot)$.

### 4.1   Problem definition for adversarial ICL

We now formalize the AT problem in ICL with the previously introduced notations. We focus on the linear regression task and introduce a novel ICL "suffix" adversarial attack, where in-context adversarial points are appended to the end of ICL inputs, to analyze the robustness of LSA models.

**In-context linear regression.** For any task indexed by $\tau$, we assume that there is a task weight $w_\tau \in \mathbb{R}^d$ drawn from $w_\tau \sim \mathcal{N}(0, I_d)$. Besides, for any in-context training point $x_{\tau,i}$ $(1 \leq i \leq N)$ and the query point $x_{\tau,q}$ (see Eq. (4)), we assume that they are drawn from $x_{\tau,i}, x_{\tau,q} \sim \mathcal{N}(0, \Lambda)$, where $\Lambda \in \mathbb{R}^{d \times d}$ is a positive-definite covariance matrix. Moreover, the ground-truth labels of training points $x_{\tau,i}$ and the query point $x_{\tau,q}$ are given by $y_{\tau,i} = w_\tau^\top x_{\tau,i}$ and $y_{\tau,q} = w_\tau^\top x_{\tau,q}$.

**ICL suffix adversarial attack.** Our novel adversarial attack against ICL models is launched via concatenating (clean) prompt embedding matrices with adversarial embedding suffixes. Specifically, for an ICL input $E_\tau$ of length $N$ (see Eq. (4)), we will form its corresponding adversarial ICL input $E_{\tau,M}^{\mathrm{adv}} \in \mathbb{R}^{(d+1) \times (N+M+1)}$ by concatenating $E_\tau$ with an adversarial suffix of length $M$ as follows,

$$E_{\tau,M}^{\mathrm{adv}} := \left( \underbrace{\begin{pmatrix} X_\tau \\ Y_\tau \end{pmatrix}}_{\text{Training Data of Length } N} \quad \underbrace{\begin{pmatrix} X_\tau^{\mathrm{sfx}} + \Delta_\tau \\ Y_\tau^{\mathrm{sfx}} \end{pmatrix}}_{\text{Adversarial Suffix of Length } M} \quad \underbrace{\begin{pmatrix} x_{\tau,q} \\ 0 \end{pmatrix}}_{\text{Query Sample From } E_\tau} \right), \qquad (6)$$

where $X_\tau := (x_{\tau,1} \cdots x_{\tau,N}) \in \mathbb{R}^{d \times N}$ and $Y_\tau := (y_{\tau,1} \cdots y_{\tau,N}) \in \mathbb{R}^{1 \times N}$ denote the $N$ original training samples and labels, and $X_\tau^{\mathrm{sfx}} := (x_{\tau,1}^{\mathrm{sfx}} \cdots x_{\tau,M}^{\mathrm{sfx}}) \in \mathbb{R}^{d \times M}$, $Y_\tau^{\mathrm{sfx}} := (y_{\tau,1}^{\mathrm{sfx}} \cdots y_{\tau,M}^{\mathrm{sfx}}) \in \mathbb{R}^{1 \times M}$, and $\Delta_\tau^{\mathrm{sfx}} := (\delta_{\tau,1} \cdots \delta_{\tau,M}) \in \mathbb{R}^{d \times M}$ denote the new $M$ clean suffix samples, clean suffix labels, and adversarial perturbations. The clean suffix samples $X_\tau^{\mathrm{sfx}}$ and labels $Y_\tau^{\mathrm{sfx}}$ here follow the same distribution as those in-context data in the embedding $E_\tau$, *i.e.*, $x_{\tau,i}^{\mathrm{sfx}} \sim \mathcal{N}(0, \Lambda)$ and $y_{\tau,i}^{\mathrm{sfx}} = w_\tau^\top x_{\tau,i}^{\mathrm{sfx}}$ hold for every $i \in [M]$. For the adversarial perturbation matrix $\Delta_\tau$, we require each perturbation $\delta_{\tau,i}$ is restricted within a ball-sphere as $\|\delta_{\tau,i}\|_2 \leq \epsilon$, where $\epsilon > 0$ is the perturbation radius. This aims to simulate that in jailbreak attacks, and each adversarial token is searched within a token vocabulary set of limited size.

The goal of the ICL adversarial attack is to add an optimal suffix adversarial perturbation matrix $\Delta_\tau$ to maximize the difference between the model prediction $\hat{y}_q(E_\tau^{\mathrm{adv}})$ based on the adversarial ICL input $E_\tau^{\mathrm{adv}}$ and the ground-truth query label $y_{\tau,q}$. We adopt the squared loss to measure such a prediction difference, which thus leads to the robust generalization error for the model $f_\theta^{\mathrm{LSA}}$ as

$$\mathcal{R}^{\mathrm{adv}}(\theta, M) = \mathbb{E}_\tau \max_{\|\Delta_\tau^\top\|_{2,\infty} \leq \epsilon} \frac{1}{2} |\hat{y}_{q,\theta}(E_{\tau,M}^{\mathrm{adv}}) - y_{\tau,q}|^2, \qquad (7)$$

where $M$ is the length of the adversarial suffix and the expectation $\mathbb{E}_\tau[\cdot]$ is calculated over the randomness of $w_\tau$, $X_\tau$, $X_\tau^{\mathrm{sfx}}$, and $x_{\tau,q}$. As we aim to understand how the adversarial prompt length in AT would affect the robustness of LLM, Eq. (7) will also only focus on how the adversarial suffix length $M$ in ICL adversarial attacks would affect the robust generalization error $\mathcal{R}^{\mathrm{adv}}(\theta, M)$.

**Adversarial in-context learning.** Following previous studies on minimax AT [32, 20, 43, 12, 58], here we adopt a minimax AT loss to train the LSA model. Concretely, we first use the aforementioned ICL adversarial attack to synthesize adversarial prompts and then update the LSA model based on these adversarial prompts to help the model gain robustness against them. We further assume that the adversarial suffix length is fixed during AT, which thus leads to the following ICL AT problem,

$$\min_\theta \mathcal{L}^{\mathrm{adv}}(\theta) := \min_\theta \mathcal{R}^{\mathrm{adv}}(\theta, M_{\mathrm{train}}) = \min_\theta \left\{ \mathbb{E}_\tau \max_{\|\Delta_\tau^\top\|_{2,\infty} \leq \epsilon} \frac{1}{2} |\hat{y}_{q,\theta}(E_{\tau,M_{\mathrm{train}}}^{\mathrm{adv}}) - y_{\tau,q}|^2 \right\}, \qquad (8)$$

where $\mathcal{L}^{\mathrm{adv}}(\theta) := \mathcal{R}^{\mathrm{adv}}(\theta, M_{\mathrm{train}})$ is the AT loss in ICL and $M_{\mathrm{train}} \in \mathbb{N}^+$ is the fixed adversarial suffix length during AT. We will perform AT with continuous gradient flow, and further following [69] to make the following assumption on the LSA model parameter initialization.

**Assumption 1** (c.f. Assumption 3 in [69]). *Let $\sigma > 0$ be a parameter and $\Theta \in \mathbb{R}^{d \times d}$ be any matrix satisfying $\|\Theta\Theta^\top\|_F = 1$ and $\Theta\Lambda \neq 0_{d \times d}$. We assume*

$$W^V(0) = \begin{pmatrix} 0_{d \times d} & 0_{d \times 1} \\ 0_{1 \times d} & \sigma \end{pmatrix}, \quad W^{KQ}(0) = \begin{pmatrix} \sigma\Theta\Theta^\top & 0_{d \times 1} \\ 0_{1 \times d} & 0 \end{pmatrix}.$$

Recall in Eq. (5), $w_{12}^V$, $w_{12}^{KQ}$, and $w_{22}^{KQ}$ do not contribute to the model prediction $\hat{y}_{q,\theta}(\cdot)$. Assumption 1 thus directly sets them to be zero at initialization. To ensure symmetric initialization, it further sets $w_{21}^V(0)$ and $w_{21}^{KQ}(0)$ to zero. We will see how Assumption 1 helps simplify the analysis of ICL AT.

## 4.2 Bridging ICL AT and LLM AT

We now discuss similarities between the ICL AT problem defined in Eq. (8) and the LLM AT problem defined in Eq. (3) to help motivate why ICL AT can be a good artifact to theoretically study LLM AT.

**Firstly, in-context inputs (*i.e.*, $E_\tau$ defined in Eq. (4)) for LSA models are similar to prompt inputs for real-world LLMs.** If we replace each token in an LLM prompt with its *one-hot encoding* defined over the token vocabulary space, then these one-hot encodings would be similar to in-context samples $x_i$ in Eq. (4) since both of them are now "feature vectors". Besides, we note that each in-context label $y_i$ in Eq. (4) can be seen as the "next-token prediction label" in real-world LLMs. The main difference is that in LLMs, the $i$-th token in a prompt can be seen as the $i$-th input token and the $(i-1)$-th next-token prediction label simultaneously, while in LSA models, the $i$-th in-context input and the $(i-1)$-th in-context label are explicitly separated into two terms $x_i$ and $y_{i-1}$.

**Secondly, the search for adversarial in-context samples (see Eq. (6)) in the ICL suffix adversarial attack is similar to the search for adversarial tokens in suffix jailbreak attacks.** We note that each adversarial token in jailbreak prompts can be seen as replacing the "padding token". Thereby, from the point of view of one-hot encoding, searching for an adversarial token can thus be seen as applying an $\ell_2$-norm adversarial perturbation within a radius of $\sqrt{2}$ to transform the one-hot encoding of the padding token to that of the adversarial token. This process is the same as the search for adversarial in-context samples in the ICL suffix adversarial attack defined in Eq. (7), which would perturb each in-context suffix sample $x_{\tau,i}^{\text{sfx}}$ within an $\ell_2$-norm ball-sphere under a given radius $\epsilon > 0$.

**Thirdly, motivations behind ICL AT and LLM AT are also similar to each other.** Both of the two AT problems aim to enhance models' robustness via training them on some synthesized adversarial inputs. The adversarial inputs syntheses in ICL AT and LLM AT are also similar, as both of them aim to make targeted models behave wrongly via manipulating suffixes of input prompts. The difference is that suffix jailbreak attacks are *targeted adversarial attacks* aimed at inducing LLMs to generate *specified* harmful content while our ICL attack is an *untargeted adversarial attack* aimed at reducing the utility of linear regression prediction made by LSA models.

## 4.3 Training dynamics of adversarial ICL

We now start to analyze the training dynamics of the minimax ICL AT problem formalized in Eq. (8). The main technical challenge is that to solve the inner maximization problem in Eq. (8), one needs to analyze the optimization of the adversarial perturbation matrix $\Delta_\tau$. However, the matrix $\Delta_\tau$ along with the clean data embedding $E_\tau$ and the clean adversarial suffix $(X_\tau^{\text{sfx}}, Y_\tau^{\text{sfx}})$ are entangled together within the adversarial ICL input $E_{\tau,M_{\text{train}}}^{\text{adv}}$, which makes it very difficult to solve the inner maximization problem and further analyze the ICL AT dynamics.

To tackle such a challenge, we propose to instead study the dynamics of a *closed-form upper bound* of the original AT loss $\mathcal{L}^{\text{adv}}(\theta)$. Formally, we will analyze the following surrogate AT problem:

$$\min_\theta \tilde{\mathcal{L}}^{\text{adv}}(\theta) := \min_\theta \Big\{ \sum_{i=1}^{4} \ell_i(\theta) \Big\}, \tag{9}$$

where $\tilde{\mathcal{L}}^{\text{adv}}(\theta) := \sum_{i=1}^{4} \ell_i(\theta)$ is the surrogate AT loss, $E_{\tau,M_{\text{train}}}^{\text{clean}} := \begin{pmatrix} X_\tau & X_\tau^{\text{sfx}} & x_{\tau,q} \\ Y_\tau & Y_\tau^{\text{sfx}} & 0 \end{pmatrix}$, and

$$\ell_1(\theta) = 2 \mathop{\mathbb{E}}_\tau \Big[ \big( (w_{21}^V)^\top \quad w_{22}^V \big) \frac{E_{\tau,M_{\text{train}}}^{\text{clean}} E_{\tau,M_{\text{train}}}^{\text{clean}\top}}{N + M_{\text{train}}} \begin{pmatrix} W_{11}^{KQ} \\ (w_{21}^{KQ})^\top \end{pmatrix} x_{\tau,q} - y_{\tau,q} \Big]^2,$$

$$\ell_2(\theta) = \frac{2\epsilon^4 M_{\text{train}}^2}{(N + M_{\text{train}})^2} \|w_{21}^V\|_2^2 \mathop{\mathbb{E}}_\tau \Big[ \|W_{11}^{KQ} x_{\tau,q}\|_2^2 \Big],$$

$$\ell_3(\theta) = \frac{2\epsilon^2 M_{\text{train}}}{(N + M_{\text{train}})^2} \mathop{\mathbb{E}}_\tau \Big[ \|W_{11}^{KQ} x_{\tau,q}\|_2^2 \cdot \| \big( (w_{21}^V)^\top \quad w_{22}^V \big) \begin{pmatrix} X_\tau^{\text{sfx}} \\ Y_\tau^{\text{sfx}} \end{pmatrix} \|_2^2 \Big],$$

$$\ell_4(\theta) = \frac{2\epsilon^2 M_{\text{train}}}{(N + M_{\text{train}})^2} \|w_{21}^V\|_2^2 \cdot \mathop{\mathbb{E}}_\tau \Big[ \| \begin{pmatrix} X_\tau^{\text{sfx}} \\ Y_\tau^{\text{sfx}} \end{pmatrix}^\top \begin{pmatrix} W_{11}^{KQ} \\ (w_{21}^{KQ})^\top \end{pmatrix} x_{\tau,q} \|_2^2 \Big].$$

The surrogate AT loss $\tilde{\mathcal{L}}^{\text{adv}}(\theta)$ in Eq. (9) is the closed-form upper bound for the original AT loss $\mathcal{L}^{\text{adv}}(\theta)$ in Eq. (8), as illustrated in the below Proposition 1 (see Appendix A.2 for the proof).

**Proposition 1** (Uniform upper bound for $\mathcal{L}^{\mathrm{adv}}(\theta)$). *For the AT loss function $\mathcal{L}^{\mathrm{adv}}(\theta)$ defined in Eq. (8) and the surrogate AT loss function $\tilde{\mathcal{L}}^{\mathrm{adv}}(\theta)$ defined in Eq. (9), for any model parameter $\theta := (W^V, W^{KQ})$ of the LSA model $f_{\mathrm{LSA},\theta}$, we uniformly have that: $\mathcal{L}^{\mathrm{adv}}(\theta) \leq \tilde{\mathcal{L}}^{\mathrm{adv}}(\theta)$.*

This result indicates that when we are training the LSA model via solving the surrogate AT problem Eq. (9), we are also reducing the model training loss in the original AT problem Eq. (8). Thus, solving the surrogate AT problem will also intuitively improve the robustness of the model.

Based on our previous analysis, we now turn to study the training dynamics of surrogate AT defined in Eq. (9). To better describe our results, we define two functions $\Gamma(\cdot) : \mathbb{N} \to \mathbb{R}^{d \times d}$ and $\psi(\cdot) : \mathbb{N} \to \mathbb{R}$, both of which depend on the adversarial suffix length $M$, as follows,

$$\Gamma(M) := \frac{N + M + 1}{N + M}\Lambda + \frac{\mathrm{Tr}(\Lambda)}{N + M}I_d \in \mathbb{R}^{d \times d}, \quad \psi(M) := \frac{M^2 \mathrm{Tr}(\Lambda)}{(N + M)^2} \in \mathbb{R}, \qquad (10)$$

where $N$ is the prompt length of the original ICL input $E_\tau$ (see Eq. (4)) and $\Lambda$ is the covariance matrix of in-context linear regression samples. The closed-form surrogate AT dynamics of the LSA model $f_{\mathrm{LSA},\theta}$ is then given in the following Theorem 1 (see Appendix A.3 for the proof).

**Theorem 1** (Closed-form Surrogate AT Dynamics). *Suppose Assumption 1 holds and $f_{\mathrm{LSA},\theta}$ is trained from the surrogate AT problem defined in Eq. (9) with continuous gradient flow. When the $\sigma$ in Assumption 1 satisfies $\sigma < \sqrt{\frac{2}{d \cdot \|(\Gamma(M_{\mathrm{train}})\Lambda + \epsilon^2 \psi(M_{\mathrm{train}})I_d)\Lambda^{-1}\|_2}}$, after training for infinite long time, the model parameter $\theta$ will converge to $\theta_*(M_{\mathrm{train}}) := (W_*^V(M_{\mathrm{train}}), W_*^{KQ}(M_{\mathrm{train}}))$, satisfying: $w_{*,12}^{KQ} = w_{*,21}^{KQ} = w_{*,12}^V = w_{*,21}^V = 0_{d \times 1}$, $w_{*,22}^{KQ} = 0$, $W_{*,11}^V = 0_{d \times d}$, and*

$$w_{*,22}^V W_{*,11}^{KQ} = \left(\Gamma(M_{\mathrm{train}})\Lambda + \epsilon^2 \psi(M_{\mathrm{train}})I_d\right)^{-1}\Lambda.$$

**Remark 1.** *When the $l_2$-norm adversarial perturbation radius $\epsilon$ is zero, the closed-form AT solution $\theta_*$ derived in Theorem 1 degenerates to that obtained without AT (see Theorem 4.1 in [69]). Thus, a sufficient large adversarial perturbation $\epsilon$ is a key to helping the LSA model $f_{\mathrm{LSA},\theta}$ obtain significant adversarial robustness. This will be further justified in the next section.*

### 4.4 Robust generalization upper-bound

With the closed-form AT solution $\theta_*(M_{\mathrm{train}})$ in Theorem 1, we now analyze the robustness of the trained LSA model. All proofs in this section are presented in Appendix A.4. We study how a LSA model adversarially trained under a fixed adversarial suffix length $M_{\mathrm{train}}$ can defend against the ICL adversarial attack with a different adversarial suffix length $M_{\mathrm{test}}$. That is, we aim to analyze the magnitude of the robust generalization error $\mathcal{R}^{\mathrm{adv}}(\theta_*(M_{\mathrm{train}}), M_{\mathrm{test}})$ for the converged robust model parameter $\theta_*(M_{\mathrm{train}})$. Here, we prove an upper-bound for it in the following theorem.

**Theorem 2** (Surrogate AT Robust Generalization Bound). *Suppose all conditions in Theorem 1 hold and $\theta_*(M_{\mathrm{train}})$ is the surrogate AT solution in Theorem 1. We have*

$$\mathcal{R}^{\mathrm{adv}}(\theta_*(M_{\mathrm{train}}), M_{\mathrm{test}}) \leq 2\mathrm{Tr}\left[\Lambda^3\left(\Gamma_{\mathrm{test}}\Lambda + \epsilon^2 \psi_{\mathrm{test}}I_d\right)\left(\Gamma_{\mathrm{train}}\Lambda + \epsilon^2 \psi_{\mathrm{train}}I_d\right)^{-2} + \Lambda\right],$$

*where $M_{\mathrm{train}}$ is the adversarial suffix length in the ICL adversarial attack, and $\Gamma_{\mathrm{train}} := \Gamma(M_{\mathrm{train}})$, $\Gamma_{\mathrm{test}} := \Gamma(M_{\mathrm{test}})$, $\psi_{\mathrm{train}} := \psi(M_{\mathrm{train}})$, and $\psi_{\mathrm{test}} := \psi(M_{\mathrm{test}})$ are functions in Eq. (10).*

We further adopt Assumption 2 to help us better understand our robust generalization bound.

**Assumption 2.** *For adversarial suffix lengths during AT and testing, we assume that $M_{train}, M_{test} \leq \mathcal{O}(N)$, where $N$ is the original ICL prompt length. Besides, for the $l_2$-norm adversarial perturbation radius, we assume that $\epsilon = \Theta(\sqrt{d})$, where $d$ is the ICL sample dimension.*

In the above Assumption 2, the assumption made on adversarial suffix lengths means that they should not be too long to make the model "forget" the original ICL prompt. Besides, the assumption made on the perturbation radius $\epsilon$ ensures that it is large enough to simulate the large (but limited) token vocabulary space of real-world LLMs to help model gain robustness.

**Corollary 1.** *Suppose Assumption 2 and all conditions in Theorem 2 hold. Suppose $\|\Lambda\|_2 \leq \mathcal{O}(1)$. Then, we have the following robust generalization bound,*

$$\mathcal{R}^{\mathrm{adv}}(\theta_*(M_{\mathrm{train}}), M_{\mathrm{test}}) \leq \mathcal{O}(d) + \mathcal{O}\left(\frac{d^2}{N}\right) + \mathcal{O}\left(N^2 \cdot \frac{M_{\mathrm{test}}^2}{M_{\mathrm{train}}^4}\right).$$

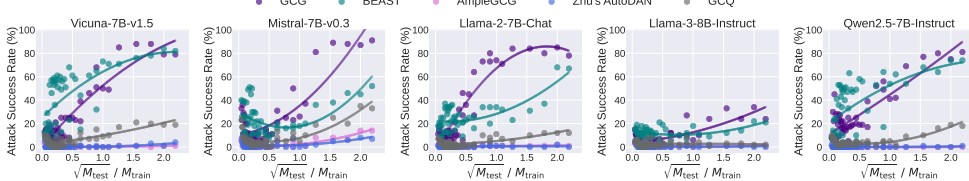

Figure 1: Scatter plots of ASR to the ratio $\sqrt{M_{\text{test}}}/M_{\text{train}}$. For each pair of base model and attack, 48 points are plotted. A high ASR indicates a weak jailbreak robustness.

Corollary 1 is our main theoretical result, which clearly show that for an adversarially trained LSA model, its robust generalization bound depends on the term $\Theta(\sqrt{M_{\text{test}}}/M_{\text{train}})$, where $M_{\text{train}}$ and $M_{\text{test}}$ are the number of adversarially perturbed in-context samples during training and testing. In other words, for an ICL adversarial attack with an adversarial suffix length $\Theta(M)$, to maintain the order of the robust generalization bound, it is enough to perform surrogate AT with only an adversarial suffix length $\Theta(\sqrt{M})$. Such an observation is useful in practice, since one can thus leverage a "short-length" AT, which requires less GPU memory and training time, to defend against "long-length" jailbreakings.

# 5 Empirical evidence

In this section, we follow Eq. (3) to perform AT on LLMs and investigate the relationship between adversarial suffix lengths during LLM AT and jailbreak attacks.

## 5.1 Experimental setup

**Models&datasets.** We adopt five pre-trained LLMs, which are: Vicuna-7B-v1.5 [70], Mistral-7B-Instruct-v0.3 [21], Llama-2-7B-Chat [52], Llama-3-8B-Instruct [15], and Qwen2.5-7B-Instruct [65]. For data in AT, we use the training set from Harmbench [36] as the safety dataset and Alpaca [50] as the utility dataset. For data in the robustness evaluation, we construct a test set of size 100 that consists of the first 50 samples from the test set of Harmbench [36] and the first 50 samples from AdvBench [73]. For data in the utility analysis, we use the benchmark data from AlpacaEval [10].

**Adversarial training.** We leverage GCG [73], a token-level jailbreak attack, to synthesize (suffix) jailbreak prompts, in which the adversarial suffix length $M_{\text{train}}$ is fixed to one of $\{5, 10, 20, 30, 40, 50\}$ during AT. To reduce computational complexity of tuning LLMs, LoRA [18] is applied to all query and key projection matrices in attentions. In every AT experiment, we follow Eq. (3) to perform AT with Adam. Please refer to Appendix B.2 for omitted settings.

**Jailbreak attacks.** We use both suffix and non-suffix jailbreak attacks to evaluate the adversarial robustness of trained LLMs. Specifically, five token-level suffix jailbreak attacks are adopted, which are GCG [73], BEAST [44], AmpleGCG [26], Zhu's AutoDAN [71], and GCQ [17]. The adversarial suffix token length $M_{\text{test}}$ is varied within $\{5, 10, 20, 40, 60, 80, 100, 120\}$. Meanwhile, two non-suffix jailbreak attacks are leveraged, which are PAIR [7] and DeepInception [25]. Please refer to Appendix B.3 for full implementation details of all used jailbreak attacks.

**Evaluations.** We focus on evaluating the jailbreak robustness and the utility of trained LLMs. For the robustness evaluation, we report the **Attack Success Rate (ASR)** of jailbreak attacks. An LLM-based judger from [36] is used to determine whether a jailbreak attack succeeds or not. For the utility evaluation, we use the AlpacaEval2 [10] to report the **Length-controlled WinRate (LC-WinRate)** of targeted models against a reference model Davinci003 evaluated under the Llama-3-70B model. An LC-WinRate of $50\%$ means that the output qualities of the two models are equal, while an LC-WinRate of $100\%$ means that the targeted model is consistently better than the reference Davinci003. Please refer to Appendix B.3 for the detailed settings of model evaluations.

## 5.2 Results analysis

**Correlation between the suffix jailbreak robustness and the ratio $\sqrt{M_{\text{test}}}/M_{\text{train}}$.** We plot the ASR of models trained and attacked with different adversarial suffix lengths in Figure 1. This results in 48 points for each pair of base model and jailbreak attack. The Pearson correlation coefficient (PCC)

Table 1: PCCs and $p$-values calculated between ASR and ratio $\sqrt{M_{\text{test}}}/M_{\text{train}}$. A high PCC (within $[-1, 1]$) means a strong correlation between ASR and the ratio. $p < 5.00 \times 10^{-2}$ means that the observation is considered statistically significant.

| Model | GCG Attack | | BEAST Attack | | AmpleGCG Attack | | Zhu's AutoDAN | | GCQ Attack | |
|---|---|---|---|---|---|---|---|---|---|---|
| | PCC(↑) | $p$-value(↓) | PCC(↑) | $p$-value(↓) | PCC(↑) | $p$-value(↓) | PCC(↑) | $p$-value(↓) | PCC(↑) | $p$-value(↓) |
| Vicuna-7B | 0.93 | $\mathbf{4.7 \times 10^{-21}}$ | 0.63 | $\mathbf{1.4 \times 10^{-6}}$ | 0.19 | $\underline{1.9 \times 10^{-1}}$ | 0.51 | $\mathbf{2.5 \times 10^{-4}}$ | 0.82 | $\mathbf{1.4 \times 10^{-12}}$ |
| Mistral-7B | 0.86 | $\mathbf{4.0 \times 10^{-15}}$ | 0.29 | $\mathbf{4.4 \times 10^{-2}}$ | 0.74 | $\mathbf{1.5 \times 10^{-9}}$ | 0.49 | $\mathbf{3.7 \times 10^{-4}}$ | 0.70 | $\mathbf{2.6 \times 10^{-8}}$ |
| Llama-2-7B | 0.88 | $\mathbf{9.0 \times 10^{-17}}$ | 0.67 | $\mathbf{1.7 \times 10^{-7}}$ | 0.37 | $\mathbf{1.0 \times 10^{-2}}$ | 0.13 | $\underline{3.8 \times 10^{-1}}$ | 0.71 | $\mathbf{2.1 \times 10^{-8}}$ |
| Llama-3-8B | 0.76 | $\mathbf{2.8 \times 10^{-10}}$ | 0.26 | $\underline{7.7 \times 10^{-2}}$ | −0.07 | $\underline{6.2 \times 10^{-1}}$ | −0.12 | $\underline{4.1 \times 10^{-1}}$ | 0.0 | $\underline{9.7 \times 10^{-1}}$ |
| Qwen2.5-7B | 0.87 | $\mathbf{1.1 \times 10^{-15}}$ | 0.58 | $\mathbf{1.0 \times 10^{-5}}$ | −0.24 | $\underline{1.0 \times 10^{-1}}$ | 0.16 | $\underline{2.6 \times 10^{-1}}$ | 0.72 | $\mathbf{1.1 \times 10^{-8}}$ |

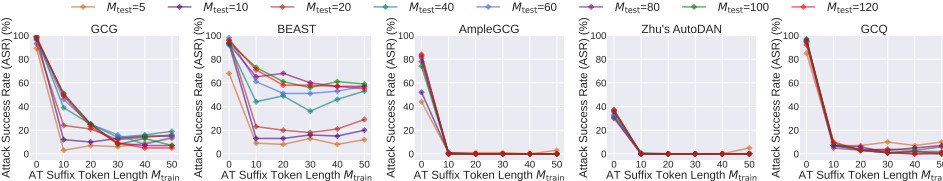

Figure 2: ASR versus $M_{\text{train}}$ on Vicuna-7B-v1.5 under jailbreaking with different $M_{\text{test}}$. $M_{\text{train}} = 0$ means that AT is not performed on the evaluated model. A low ASR indicates a strong robustness.

and the corresponding $p$-value between the ratio $\sqrt{M_{\text{test}}}/M_{\text{train}}$ and the ASR are calculated in Table 1, where **bold $p$-values** indicate that observations are statistically significant (*i.e.*, $p < 0.05$), while underlined ones indicate they are not significant.

When the jailbreak attack used during AT is the same as that used during robustness evaluation (*i.e.*, GCG), one can observe from Figure 1 that a clear positive correlation between the ratio $\sqrt{M_{\text{test}}}/M_{\text{train}}$ and the ASR for all evaluated base models. Further, high PCCs ($> 0.7$) and low $p$-values ($< 0.05$) in Table 1 also confirm that the observed positive correlation is statistically significant.

Besides, when the jailbreak attack is BEAST and GCQ, which is different from that used during AT, the significant positive correlation between the ratio $\sqrt{M_{\text{test}}}/M_{\text{train}}$ and the ASR can only be observed from some of the base models. This may be due to the fact that AT with only a single jailbreak attack may not help the model generalize well to unseen attacks. Therefore, it might be necessary to adopt multiple attacks when performing AT-based alignment on LLMs. Nevertheless, from Figure 1, we find that for those models where the correlation is not significant (*i.e.*, Mistral-7B, and Llama-3-8B), GCG-based AT can still suppress the ASR to no more than $50\%$, which indicates that it can still help models gain a certain degree of robustness against unseen attacks.

Finally, for AmpleGCG and Zhu's AutoDAN attacks, we notice that the correlation between the ratio $\sqrt{M_{\text{test}}}/M_{\text{train}}$ and the ASR cannot be observed on most of the base models. However, this is simply due to AT being too effective in defending against these two attacks: from Figure 1, one can observe that AT effectively reduces ASRs of AmpleGCG and Zhu's AutoDAN to nearly zero in most cases.

**Relationship between adversarial suffix lengths in AT (*i.e.*, $M_{\text{train}}$) and suffix jailbreaking (*i.e.*, $M_{\text{test}}$).** We plot curves of the ASR on Vicuna-7B versus the adversarial suffix token length during AT in Figure 2. Results on remaining base models are presented in Figure 4 in Appendix B.4. From these figures, we find that as the adversarial suffix token length during AT increases, AT can effectively reduce the ASR of all analyzed attacks. Furthermore, when the AT adversarial suffix token length is

Table 2: Time cost (hrs) of LLM AT with different adversarial suffix lengths.

| Model | Adversarial Suffix Token Length $M_{\text{train}}$ in AT | | | | | |
|---|---|---|---|---|---|---|
| | 5 | 10 | 20 | 30 | 40 | 50 |
| Vicuna-7B | 10.2h | 11.3h | 13.8h | 16.0h | 18.2h | 20.4h |
| Mistral-7B | 8.9h | 9.9h | 12.0h | 14.3h | 16.6h | 19.0h |
| Llama-2-7B | 9.9h | 11.0h | 13.2h | 15.5h | 18.1h | 20.0h |
| Llama-3-8B | 9.7h | 10.8h | 13.1h | 15.3h | 17.7h | 20.2h |
| Qwen2.5-7B | 9.1h | 9.9h | 11.7h | 13.9h | 16.4h | 18.4h |

set to 20, AT is already able to reduce the ASR by at least $30\%$ under all settings. All these results demonstrate the effectiveness of defending against long-length jailbreaking with short-length AT.

**Time cost of LLM AT with different adversarial suffix lengths $M_{\text{train}}$.** We then present the time costs of performing LLM AT in Table 2. From the table, we find that when the adversarial suffix

Table 3: ASR(%) of non-suffix jailbreak attacks versus models adversarially trained with different adversarial suffix length $M_{\text{train}}$. A low ASR indicates a strong robustness.

| Attack | Model | Adversarial Suffix Token Length $M_{\text{train}}$ in AT | | | | | | |
|--------|-------|-----------|------|------|------|------|------|------|
| | | 0 (No AT) | 5 | 10 | 20 | 30 | 40 | 50 |
| PAIR | Vicuna-7B | 84.0 | 53.0 | 48.0 | 42.0 | 50.0 | 44.0 | 55.0 |
| | Qwen2.5-7B | 71.0 | 20.0 | 17.0 | 25.0 | 19.0 | 24.0 | 26.0 |
| DeepInception | Vicuna-7B | 76.0 | 39.0 | 15.0 | 0.0 | 0.0 | 0.0 | 0.0 |
| | Qwen2.5-7B | 89.0 | 0.0 | 0.0 | 0.0 | 0.0 | 0.0 | 0.0 |

length $M_{\text{train}}$ during AT is as long as 50, the time cost of AT can reach around 20 hours, which is around 30% to 60% longer than that when $M_{\text{train}}$ is set to 20 or 30. Meanwhile, according to Figure 2 in this section and Figure 4 in Appendix B.4, AT with a short adversarial suffix length of 20 or 30 can already enable trained LLMs to achieve strong jailbreak robustness. These results demonstrate the advantages of using short-length AT instead of long-length AT.

**Robustness of jailbreak attacks beyond suffix attacks.**
We also calculate the ASR of two non-suffix jailbreak attacks, PAIR and DeepInception attacks, against LLM AT in Table 3. From the table, one can observe that: (1) For the DeepInception attack, LLM AT with a short adversarial suffix length ($M_{\text{train}} = 20$) can already suppress its ASR to 0%. (2) For the PAIR attack, while LLM AT with a short adversarial suffix length can reduce its ASR from 84% to around 50% against the Vicuna-7B model and from 71% to around 25% against the Qwen2.5-7B model, further increasing the suffix length does not help much to improve LLM robustness against PAIR. These results suggest that the mechanisms behind suffix-based and non-suffix-based jailbreak attacks might have different properties.

**Utility analysis.** Finally, we plot the LC-WinRate of models trained under different adversarial suffix token lengths and the original model (*i.e.*, $M_{\text{train}} = 0$) in Figure 3. We find that while AT reduces the utility of models, they

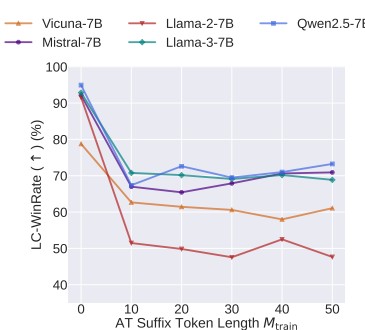

Figure 3: Utility analysis based on LC-WinRate against the referenced Davinci003 model. A high LC-WinRate indicates strong model utility.

can still achieve WinRates close to or more than 50% against the reference model Davinci003. This means that these adversarially trained models achieve utility comparable to Davinci003.

## 6 Conclusion

We study the AT problem in LLMs and unveils that to defend against a suffix jailbreak attack with suffix length of $\Theta(M)$, it is sufficient to perform AT on jailbreak prompts with suffix length of $\Theta(\sqrt{M})$. The finding is supported by both theoretical and empirical evidence. Theoretically, we define a new AT problem in the ICL theory and prove a robust generalization bound for adversarially trained linear transformers. This bound has a positive correlation with $\Theta(\sqrt{M_{\text{test}}}/M_{\text{train}})$. Empirically, we conduct AT on real-world LLMs and confirm a clear positive correlation between the jailbreak ASR and the ratio $\sqrt{M_{\text{test}}}/M_{\text{train}}$. Our results indicate that it is possible to conduct efficient short-length AT against strong long-length jailbreaking.

## Acknowledgements

Di Wang and Shaopeng Fu are supported in part by the funding BAS/1/1689-01-01 and funding from KAUST - Center of Excellence for Generative AI, under award number 5940.

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

# A   Proofs

This section collects all the proofs in this paper.

## A.1   Technical lemmas

This section presents several technical lemmas that will be used in our proofs.

**Lemma A.1** (c.f. Lemma D.2 in [69]). *If $x \in \mathbb{R}^{d \times 1}$ is Gaussian random vector of $d$ dimension, mean zero and covariance matrix $\Lambda$, and $A \in \mathbb{R}^{d \times d}$ is a fixed matrix. Then*

$$\mathbb{E}[xx^\top A xx^\top] = \Lambda(A + A^\top)\Lambda + \mathrm{Tr}(A\Lambda)\Lambda.$$

**Lemma A.2.** *If $x \in \mathbb{R}^{d \times 1}$ is Gaussian random vector of $d$ dimension, mean zero and covariance matrix $\Lambda$, and $A \in \mathbb{R}^{d \times d}$ is a fixed matrix. Then*

$$\mathbb{E}[x^\top A x] = \mathrm{Tr}(A\Lambda).$$

*Proof.* Since

$$\mathbb{E}[x^\top A x] = \mathbb{E}\Big[\sum_{i,j} x_i A_{i,j} x_j\Big] = \sum_{i,j} A_{i,j} \cdot \mathbb{E}[x_i x_j] = \sum_{i,j} A_{i,j} \cdot \Lambda_{i,j} = \sum_{i=1}^{d}(A\Lambda^\top)_{i,i} = \mathrm{Tr}(A\Lambda),$$

which completes the proof. $\qquad\square$

**Lemma A.3.** *For any matrices $A \in \mathbb{R}^{n \times m}$ and $B \in \mathbb{R}^{m \times n}$, we have*

$$\mathrm{Tr}(AB) = \mathrm{Tr}(BA).$$

*Proof.* Since

$$\mathrm{Tr}(AB) = \sum_{i=1}^{n}(AB)_{i,i} = \sum_{i=1}^{n}\sum_{j=1}^{m} A_{i,j} B_{j,i} = \sum_{j=1}^{m}\sum_{i=1}^{n} B_{j,i} A_{i,j} = \sum_{j=1}^{m}(BA)_{j,j} = \mathrm{Tr}(BA),$$

which completes the proof. $\qquad\square$

**Lemma A.4** (From Lemma D.1 in [69]; Also in [39]). *Let $X \in \mathbb{R}^{n \times m}$ be a variable matrix and $A \in \mathbb{R}^{a \times n}$ and $B \in \mathbb{R}^{n \times m}$ be two fixed matrices. Then, we have*

$$\partial_X \mathrm{Tr}(BX^\top) = B \in \mathbb{R}^{n \times m},$$
$$\partial_X \mathrm{Tr}(AXBX^\top) = (AXB + A^\top X B^\top) \in \mathbb{R}^{n \times m}.$$

**Lemma A.5** (Von Neumann's Trace Inequality; Also in Lemma D.3 in [69]). *Let $A \in \mathbb{R}^{n \times m}$ and $B \in \mathbb{R}^{m \times n}$ be two matrices. Suppose $(\sigma_1(A), \cdots \sigma_{\min\{n,m\}}(A))$ and $(\sigma_1(B), \cdots \sigma_{\min\{n,m\}}(B))$ are all the singular values of $A$ and $B$, respectively. We have*

$$\mathrm{Tr}(AB) \le \sum_{i=1}^{\min\{n,m\}} \sigma_i(A)\sigma_i(B) \le \sum_{i=1}^{\min\{n,m\}} \|A\|_2 \cdot \|B\|_2 = \min\{n,m\} \cdot \|A\|_2 \cdot \|B\|_2.$$

## A.2   Proof of Proposition 1

This section presents the proof of Proposition 1.

*Proof of Proposition 1.* For the AT loss $\mathcal{L}(\theta)$ defined in Eq. (8), we have that

$$\mathcal{L}^{\mathrm{adv}}(\theta) := \mathcal{R}^{\mathrm{adv}}(\theta, M_{\mathrm{train}}) = \mathbb{E}_\tau \max_{\|\Delta_\tau^\top\|_{2,\infty} \le \epsilon} |\hat{y}_{q,\theta}(E_{\tau,M_{\mathrm{train}}}^{\mathrm{adv}}) - y_{\tau,q}|^2$$

$$= \mathbb{E}_\tau \left\{ \max_{\|\Delta_\tau^\top\|_{2,\infty} \le \epsilon} \frac{1}{2} \left| \begin{pmatrix} (w_{21}^V)^\top & w_{22}^V \end{pmatrix} \cdot \frac{E_{\tau,M_{\mathrm{train}}}^{\mathrm{adv}} E_{\tau,M_{\mathrm{train}}}^{\mathrm{adv},\top}}{N + M_{\mathrm{train}}} \cdot \begin{pmatrix} W_{11}^{KQ} \\ (w_{21}^{KQ})^\top \end{pmatrix} \cdot x_{\tau,q} - y_{\tau,q} \right|^2 \right\}. \quad (\text{A.1})$$

Then, the term $E_{\tau,M_{\text{train}}}^{\text{adv}} E_{\tau,M_{\text{train}}}^{\text{adv},\top}$ can be decomposed as follows,

$$
\begin{aligned}
E_{\tau,M_{\text{train}}}^{\text{adv}} E_{\tau,M_{\text{train}}}^{\text{adv},\top} &= \left( \begin{pmatrix} X_\tau \\ Y_\tau \end{pmatrix} \quad \begin{pmatrix} X_\tau^{\text{sfx}} + \Delta_\tau \\ Y_\tau^{\text{sfx}} \end{pmatrix} \quad \begin{pmatrix} x_{\tau,q} \\ 0 \end{pmatrix} \right) \cdot \left( \begin{pmatrix} X_\tau \\ Y_\tau \end{pmatrix} \quad \begin{pmatrix} X_\tau^{\text{sfx}} + \Delta_\tau \\ Y_\tau^{\text{sfx}} \end{pmatrix} \quad \begin{pmatrix} x_{\tau,q} \\ 0 \end{pmatrix} \right)^\top \\
&= \begin{pmatrix} X_\tau & X_\tau^{\text{sfx}} & x_{\tau,q} \\ Y_\tau & Y_\tau^{\text{sfx}} & 0 \end{pmatrix} \begin{pmatrix} X_\tau & X_\tau^{\text{sfx}} & x_{\tau,q} \\ Y_\tau & Y_\tau^{\text{sfx}} & 0 \end{pmatrix}^\top + \begin{pmatrix} 0_{d\times N} & \Delta_\tau & 0_{d\times 1} \\ 0_{1\times N} & 0_{1\times M_{\text{train}}} & 0 \end{pmatrix} \begin{pmatrix} 0_{d\times N} & \Delta_\tau & 0_{d\times 1} \\ 0_{1\times N} & 0_{1\times M_{\text{train}}} & 0 \end{pmatrix}^\top \\
&\quad + \begin{pmatrix} X_\tau & X_\tau^{\text{sfx}} & x_{\tau,q} \\ Y_\tau & Y_\tau^{\text{sfx}} & 0 \end{pmatrix} \begin{pmatrix} 0_{d\times N} & \Delta_\tau & 0_{d\times 1} \\ 0_{1\times N} & 0_{1\times M_{\text{train}}} & 0 \end{pmatrix}^\top + \begin{pmatrix} 0_{d\times N} & \Delta_\tau & 0_{d\times 1} \\ 0_{1\times N} & 0_{1\times M_{\text{train}}} & 0 \end{pmatrix} \begin{pmatrix} X_\tau & X_\tau^{\text{sfx}} & x_{\tau,q} \\ Y_\tau & Y_\tau^{\text{sfx}} & 0 \end{pmatrix}^\top \\
&= \begin{pmatrix} X_\tau & X_\tau^{\text{sfx}} & x_{\tau,q} \\ Y_\tau & Y_\tau^{\text{sfx}} & 0 \end{pmatrix} \begin{pmatrix} X_\tau & X_\tau^{\text{sfx}} & x_{\tau,q} \\ Y_\tau & Y_\tau^{\text{sfx}} & 0 \end{pmatrix}^\top + \begin{pmatrix} \Delta_\tau \\ 0_{1\times M_{\text{train}}} \end{pmatrix} \begin{pmatrix} \Delta_\tau \\ 0_{1\times M_{\text{train}}} \end{pmatrix}^\top \\
&\quad + \begin{pmatrix} X_\tau^{\text{sfx}} \\ Y_\tau^{\text{sfx}} \end{pmatrix} \begin{pmatrix} \Delta_\tau \\ 0_{1\times M_{\text{train}}} \end{pmatrix}^\top + \begin{pmatrix} \Delta_\tau \\ 0_{1\times M_{\text{train}}} \end{pmatrix} \begin{pmatrix} X_\tau^{\text{sfx}} \\ Y_\tau^{\text{sfx}} \end{pmatrix}^\top ,
\end{aligned}
$$

which further means that

$$
\begin{aligned}
&\left( (w_{21}^V)^\top \quad w_{22}^V \right) \cdot \frac{E_{\tau,M_{\text{train}}}^{\text{adv}} E_{\tau,M_{\text{train}}}^{\text{adv},\top}}{N + M_{\text{train}}} \cdot \begin{pmatrix} W_{11}^{KQ} \\ (w_{21}^{KQ})^\top \end{pmatrix} \cdot x_{\tau,q} \\
&= \left( (w_{21}^V)^\top \quad w_{22}^V \right) \cdot \frac{\begin{pmatrix} X_\tau & X_\tau^{\text{sfx}} & x_{\tau,q} \\ Y_\tau & Y_\tau^{\text{sfx}} & 0 \end{pmatrix} \begin{pmatrix} X_\tau & X_\tau^{\text{sfx}} & x_{\tau,q} \\ Y_\tau & Y_\tau^{\text{sfx}} & 0 \end{pmatrix}^\top}{N + M_{\text{train}}} \cdot \begin{pmatrix} W_{11}^{KQ} \\ (w_{21}^{KQ})^\top \end{pmatrix} \cdot x_{\tau,q} \\
&\quad + (w_{21}^V)^\top \cdot \frac{\Delta_\tau \Delta_\tau^\top}{N + M_{\text{train}}} \cdot W_{11}^{KQ} x_{\tau,q} + \left( (w_{21}^V)^\top \quad w_{22}^V \right) \cdot \frac{\begin{pmatrix} X_\tau^{\text{sfx}} \\ Y_\tau^{\text{sfx}} \end{pmatrix} \Delta_\tau^\top}{N + M_{\text{train}}} \cdot W_{11}^{KQ} x_{\tau,q} \\
&\quad + (w_{21}^V)^\top \cdot \frac{\Delta_\tau \begin{pmatrix} X_\tau^{\text{sfx}} \\ Y_\tau^{\text{sfx}} \end{pmatrix}^\top}{N + M_{\text{train}}} \cdot \begin{pmatrix} W_{11}^{KQ} \\ (w_{21}^{KQ})^\top \end{pmatrix} x_{\tau,q}.
\end{aligned} \tag{A.2}
$$

Inserting Eq. (A.2) into Eq. (A.1) and applying the inequality that $|a + b|^2 \le 2 \cdot (a^2 + b^2)$, $\mathcal{L}^{\text{adv}}(\theta)$ can thus be bounded as

$$
\begin{aligned}
\mathcal{L}^{\text{adv}}(\theta) \le{}& 2 \cdot \mathbb{E}_\tau \left[ \left( (w_{21}^V)^\top \quad w_{22}^V \right) \cdot \frac{\begin{pmatrix} X_\tau & X_\tau^{\text{sfx}} & x_{\tau,q} \\ Y_\tau & Y_\tau^{\text{sfx}} & 0 \end{pmatrix} \begin{pmatrix} X_\tau & X_\tau^{\text{sfx}} & x_{\tau,q} \\ Y_\tau & Y_\tau^{\text{sfx}} & 0 \end{pmatrix}^\top}{N + M_{\text{train}}} \cdot \begin{pmatrix} W_{11}^{KQ} \\ (w_{21}^{KQ})^\top \end{pmatrix} \cdot x_{\tau,q} - y_{\tau,q} \right]^2 \\
&+ \underbrace{2 \cdot \mathbb{E}_\tau \max_{\|\Delta_\tau^\top\|_{2,\infty} \le \epsilon} \left[ (w_{21}^V)^\top \cdot \frac{\Delta_\tau \Delta_\tau^\top}{N + M_{\text{train}}} \cdot W_{11}^{KQ} x_{\tau,q} \right]^2}_{:=A_1(\theta)} \\
&+ \underbrace{2 \cdot \mathbb{E}_\tau \max_{\|\Delta_\tau^\top\|_{2,\infty} \le \epsilon} \left[ \left( (w_{21}^V)^\top \quad w_{22}^V \right) \cdot \frac{\begin{pmatrix} X_\tau^{\text{sfx}} \\ Y_\tau^{\text{sfx}} \end{pmatrix} \Delta_\tau^\top}{N + M_{\text{train}}} \cdot W_{11}^{KQ} x_{\tau,q} \right]^2}_{:=A_2(\theta)} \\
&+ \underbrace{2 \cdot \mathbb{E}_\tau \max_{\|\Delta_\tau^\top\|_{2,\infty} \le \epsilon} \left[ (w_{21}^V)^\top \cdot \frac{\Delta_\tau \begin{pmatrix} X_\tau^{\text{sfx}} \\ Y_\tau^{\text{sfx}} \end{pmatrix}^\top}{N + M_{\text{train}}} \cdot \begin{pmatrix} W_{11}^{KQ} \\ (w_{21}^{KQ})^\top \end{pmatrix} x_{\tau,q} \right]^2}_{:=A_3(\theta)}.
\end{aligned} \tag{A.3}
$$

We then bound terms $A_1(\theta)$, $A_2(\theta)$, and $A_3(\theta)$ in Eq. (A.3) seprately. For the term $A_1(\theta)$ in Eq. (A.3), we have

$$A_1(\theta) := \frac{2}{(N + M_{\text{train}})^2} \cdot \mathbb{E}_\tau \max_{\|\Delta_\tau^\top\|_{2,\infty} \leq \epsilon} \left[ (w_{21}^V)^\top \cdot \sum_{i=1}^{M_{\text{train}}} \delta_{\tau,i} \delta_{\tau,i}^\top \cdot W_{11}^{KQ} x_{\tau,q} \right]^2$$

$$\leq \frac{2}{(N + M_{\text{train}})^2} \cdot \mathbb{E}_\tau \max_{\|\Delta_\tau^\top\|_{2,\infty} \leq \epsilon} \underbrace{\left[ \sum_{i=1}^{M_{\text{train}}} [(w_{21}^V)^\top \delta_{\tau,i}]^2 \cdot \sum_{i=1}^{M_{\text{train}}} [\delta_{\tau,i}^\top W_{11}^{KQ} x_{\tau,q}]^2 \right]}_{\text{by Cauchy-Schwarz Inequality}}$$

$$\leq \frac{2}{(N + M_{\text{train}})^2} \cdot \mathbb{E}_\tau \left[ \sum_{i=1}^{M_{\text{train}}} \max_{\|\delta_{\tau,i}\|_2 \leq \epsilon} [(w_{21}^V)^\top \delta_{\tau,i}]^2 \cdot \sum_{i=1}^{M_{\text{train}}} \max_{\|\delta_{\tau,i}\|_2 \leq \epsilon} [\delta_i^\top W_{11}^{KQ} x_{\tau,q}]^2 \right]$$

$$= \frac{2}{(N + M_{\text{train}})^2} \cdot \mathbb{E}_\tau \left[ \sum_{i=1}^{M_{\text{train}}} [\|w_{21}^V\|_2 \cdot \epsilon]^2 \cdot \sum_{i=1}^{M_{\text{train}}} [\|W_{11}^{KQ} x_{\tau,q}\|_2 \cdot \epsilon]^2 \right]$$

$$= \frac{2\epsilon^4 M_{\text{train}}^2}{(N + M_{\text{train}})^2} \cdot \|w_{21}^V\|_2^2 \cdot \mathbb{E}_\tau \|W_{11}^{KQ} x_{\tau,q}\|_2^2. \tag{A.4}$$

For the term $A_2(\theta)$ in Eq. (A.3), we have

$$A_2(\theta) := \frac{2}{(N + M_{\text{train}})^2} \cdot \mathbb{E}_\tau \max_{\|\Delta_\tau^\top\|_{2,\infty} \leq \epsilon} \left[ \begin{pmatrix} (w_{21}^V)^\top & w_{22}^V \end{pmatrix} \cdot \sum_{i=1}^{M_{\text{train}}} \begin{pmatrix} x_{\tau,i}^{\text{sfx}} \\ y_{\tau,i}^{\text{sfx}} \end{pmatrix} \delta_{\tau,i}^\top \cdot W_{11}^{KQ} x_{\tau,q} \right]^2$$

$$\leq \frac{2}{(N + M_{\text{train}})^2} \cdot \mathbb{E}_\tau \max_{\|\Delta_\tau^\top\|_{2,\infty} \leq \epsilon} \underbrace{\left[ \sum_{i=1}^{M_{\text{train}}} \left[ \begin{pmatrix} (w_{21}^V)^\top & w_{22}^V \end{pmatrix} \begin{pmatrix} x_{\tau,i}^{\text{sfx}} \\ y_{\tau,i}^{\text{sfx}} \end{pmatrix} \right]^2 \cdot \sum_{i=1}^{M_{\text{train}}} [\delta_{\tau,i}^\top W_{11}^{KQ} x_{\tau,q}]^2 \right]}_{\text{by Cauchy-Schwarz Inequality}}$$

$$= \frac{2}{(N + M_{\text{train}})^2} \cdot \sum_{i=1}^{M_{\text{train}}} \mathbb{E}_\tau \left[ \begin{pmatrix} (w_{21}^V)^\top & w_{22}^V \end{pmatrix} \begin{pmatrix} x_{\tau,i}^{\text{sfx}} \\ y_{\tau,i}^{\text{sfx}} \end{pmatrix} \right]^2 \cdot \sum_{i=1}^{M_{\text{train}}} \mathbb{E}_\tau \left[ \max_{\|\delta_{\tau,i}\|_2 \leq \epsilon} [\delta_{\tau,i}^\top W_{11}^{KQ} x_{\tau,q}]^2 \right]$$

$$= \frac{2}{(N + M_{\text{train}})^2} \cdot \sum_{i=1}^{M_{\text{train}}} \mathbb{E}_\tau \left[ \begin{pmatrix} (w_{21}^V)^\top & w_{22}^V \end{pmatrix} \begin{pmatrix} x_{\tau,i}^{\text{sfx}} \\ y_{\tau,i}^{\text{sfx}} \end{pmatrix} \right]^2 \cdot \sum_{i=1}^{M_{\text{train}}} \mathbb{E}_\tau [\|W_{11}^{KQ} x_{\tau,q}\|_2 \cdot \epsilon]^2$$

$$= \frac{2\epsilon^2 M_{\text{train}}}{(N + M_{\text{train}})^2} \cdot \mathbb{E}_\tau \|W_{11}^{KQ} x_{\tau,q}\|_2^2 \cdot \sum_{i=1}^{M_{\text{train}}} \mathbb{E}_\tau \left[ \begin{pmatrix} (w_{21}^V)^\top & w_{22}^V \end{pmatrix} \begin{pmatrix} x_{\tau,i}^{\text{sfx}} \\ y_{\tau,i}^{\text{sfx}} \end{pmatrix} \right]^2. \tag{A.5}$$

For the term $A_3(\theta)$ in Eq. (A.3), we have

$$A_3(\theta) := \frac{2}{(N + M_{\text{train}})^2} \cdot \mathbb{E}_\tau \max_{\|\Delta_\tau^\top\|_{2,\infty} \leq \epsilon} \left[ (w_{21}^V)^\top \cdot \sum_{i=1}^{M_{\text{train}}} \delta_{\tau,i} \begin{pmatrix} x_{\tau,i}^{\text{sfx}} \\ y_{\tau,i}^{\text{sfx}} \end{pmatrix}^\top \cdot \begin{pmatrix} W_{11}^{KQ} \\ (w_{21}^{KQ})^\top \end{pmatrix} x_{\tau,q} \right]^2$$

$$\leq \frac{2}{(N + M_{\text{train}})^2} \cdot \mathbb{E}_\tau \max_{\|\Delta_\tau^\top\|_{2,\infty} \leq \epsilon} \underbrace{\left[ \sum_{i=1}^{M_{\text{train}}} [(w_{21}^V)^\top \delta_{\tau,i}]^2 \cdot \sum_{i=1}^{M_{\text{train}}} \left[ \begin{pmatrix} x_{\tau,i}^{\text{sfx}} \\ y_{\tau,i}^{\text{sfx}} \end{pmatrix}^\top \begin{pmatrix} W_{11}^{KQ} \\ (w_{21}^{KQ})^\top \end{pmatrix} x_{\tau,q} \right]^2 \right]}_{\text{by Cauchy-Schwarz Inequality}}$$

$$= \frac{2}{(N + M_{\text{train}})^2} \cdot \mathbb{E}_\tau \left[ \sum_{i=1}^{M_{\text{train}}} \max_{\|\delta_{\tau,i}\|_2 \leq \epsilon} [(w_{21}^V)^\top \delta_{\tau,i}]^2 \cdot \sum_{i=1}^{M_{\text{train}}} \left[ \begin{pmatrix} x_{\tau,i}^{\text{sfx}} \\ y_{\tau,i}^{\text{sfx}} \end{pmatrix}^\top \begin{pmatrix} W_{11}^{KQ} \\ (w_{21}^{KQ})^\top \end{pmatrix} x_{\tau,q} \right]^2 \right]$$

$$= \frac{2}{(N + M_{\text{train}})^2} \cdot \mathbb{E}_\tau \left[ \sum_{i=1}^{M_{\text{train}}} [\|w_{21}^V\|_2 \cdot \epsilon]^2 \cdot \sum_{i=1}^{M_{\text{train}}} \left[ \begin{pmatrix} x_{\tau,i}^{\text{sfx}} \\ y_{\tau,i}^{\text{sfx}} \end{pmatrix}^\top \begin{pmatrix} W_{11}^{KQ} \\ (w_{21}^{KQ})^\top \end{pmatrix} x_{\tau,q} \right]^2 \right]$$

$$= \frac{2\epsilon^2 M_{\text{train}}}{(N + M_{\text{train}})^2} \cdot \|w_{21}^V\|_2^2 \cdot \sum_{i=1}^{M_{\text{train}}} \mathbb{E}_\tau \left[ \begin{pmatrix} x_{\tau,i}^{\text{sfx}} \\ y_{\tau,i}^{\text{sfx}} \end{pmatrix}^\top \begin{pmatrix} W_{11}^{KQ} \\ (w_{21}^{KQ})^\top \end{pmatrix} x_{\tau,q} \right]^2. \tag{A.6}$$

As a result, by inserting Eqs. (A.4), (A.5), and (A.6) into Eq. (A.3), we finally have that

$$
\begin{aligned}
\mathcal{L}^{\mathrm{adv}}(\theta) \leq 2 \cdot \underset{\tau}{\mathbb{E}} \Bigg[ \big( (w_{21}^V)^\top \quad w_{22}^V \big) \cdot \frac{\begin{pmatrix} X_\tau & X_\tau^{\mathrm{sfx}} & x_{\tau,q} \\ Y_\tau & Y_\tau^{\mathrm{sfx}} & 0 \end{pmatrix} \begin{pmatrix} X_\tau & X_\tau^{\mathrm{sfx}} & x_{\tau,q} \\ Y_\tau & Y_\tau^{\mathrm{sfx}} & 0 \end{pmatrix}^\top}{N + M_{\mathrm{train}}} \cdot \begin{pmatrix} W_{11}^{KQ} \\ (w_{21}^{KQ})^\top \end{pmatrix} \cdot x_{\tau,q} - y_{\tau,q} \Bigg]^2 \\
+ \frac{2\epsilon^4 M_{\mathrm{train}}^2}{(N + M_{\mathrm{train}})^2} \cdot \| w_{21}^V \|_2^2 \cdot \underset{\tau}{\mathbb{E}} \, \| W_{11}^{KQ} x_{\tau,q} \|_2^2 \\
+ \frac{2\epsilon^2 M_{\mathrm{train}}}{(N + M_{\mathrm{train}})^2} \cdot \underset{\tau}{\mathbb{E}} \, \| W_{11}^{KQ} x_{\tau,q} \|_2^2 \cdot \sum_{i=1}^{M_{\mathrm{train}}} \underset{\tau}{\mathbb{E}} \Bigg[ \big( (w_{21}^V)^\top \quad w_{22}^V \big) \begin{pmatrix} x_{\tau,i}^{\mathrm{sfx}} \\ y_{\tau,i}^{\mathrm{sfx}} \end{pmatrix} \Bigg]^2 \\
+ \frac{2\epsilon^2 M_{\mathrm{train}}}{(N + M_{\mathrm{train}})^2} \cdot \| w_{21}^V \|_2^2 \cdot \sum_{i=1}^{M_{\mathrm{train}}} \underset{\tau}{\mathbb{E}} \Bigg[ \begin{pmatrix} x_{\tau,i}^{\mathrm{sfx}} \\ y_{\tau,i}^{\mathrm{sfx}} \end{pmatrix}^\top \begin{pmatrix} W_{11}^{KQ} \\ (w_{21}^{KQ})^\top \end{pmatrix} x_{\tau,q} \Bigg]^2 .
\end{aligned}
\tag{A.7}
$$

The right-hand-side of Eq. (A.7) is exactly the surrogate AT loss $\tilde{\mathcal{L}}^{\mathrm{adv}}(\theta)$ in Eq. (9), which thus completes the proof. □

### A.3 Proof of Theorem 1

This section presents the proof of Theorem 1, which is inspired by that in [69]. Specifically:

1. we first prove that terms $w_{21}^V$ and $w_{21}^{KQ}$ stay zero during the surrogate AT (Lemma A.6) via continuous gradient-flow, which thus can simplify the surrogate AT loss $\tilde{\mathcal{L}}^{\mathrm{adv}}(\theta)$ defined in Eq. (9) (Lemma A.7).

2. We then calculate a closed-form solution $\theta_*$ for the surrogate AT problem based on the simplified $\tilde{\mathcal{L}}^{\mathrm{adv}}(\theta)$ (Lemma A.8), which is exactly the solution given in Theorem 1.

3. Finally, we prove that under the continuous gradient flow, the LSA model starts from the initial point defined in Assumption 1 can indeed converge to the closed-form solution $\theta_*$ (Lemma A.12), which thus completes the proof of Theorem 1.

We now start to prove the following Lemma A.6.

**Lemma A.6.** *Suppose Assumption 1 holds and the LSA model $f_{\mathrm{LSA},\theta}$ is trained via minimizing surrogate AT loss $\tilde{\mathcal{L}}^{\mathrm{adv}}(\theta)$ in Eq. (9) with continuous gradient flow. Then, for any continuous training time $t \geq 0$, we uniformly have that $w_{21}^V(t) = w_{21}^{KQ}(t) = 0_{d \times 1}$.*

*Proof.* When the LSA model $f_{\mathrm{LSA},\theta}$ is trained with continuous gradient-flow, the updates of $w_{21}^V$ and $w_{21}^{KQ}$ with respect to the continuous training time $t \geq 0$ are given by

$$
\begin{aligned}
\partial_t w_{21}^V(t) &:= -\partial_{w_{21}^V} \tilde{\mathcal{L}}^{\mathrm{adv}}(\theta), \\
\partial_t w_{21}^{KQ}(t) &:= -\partial_{w_{21}^{KQ}} \tilde{\mathcal{L}}^{\mathrm{adv}}(\theta).
\end{aligned}
$$

Meanwhile, since Assumption 1 assumes that $w_{21}^V(0) = W_{21}^{KQ}(0) = 0_{d \times 1}$, therefore, to complete the proof, we only need to show that $\partial_t w_{21}^V(t) = \partial_t W_{21}^{KQ}(t) = 0_{1 \times d}$ as long as $w_{21}^V(t) = W_{21}^{KQ}(t) = 0_{d \times 1}$ for any $t \geq 0$. In other words, below we need to show that $w_{21}^V = W_{21}^{KQ} = 0_{d \times 1}$ indicates $\partial_{w_{21}^V} \tilde{\mathcal{L}}^{\mathrm{adv}}(\theta) = \partial_{w_{21}^{KQ}} \tilde{\mathcal{L}}^{\mathrm{adv}}(\theta) = 0_{1 \times d}$.

Toward this end, we adopt the notation in Eq. (9) to decompose the surrogate AT loss $\tilde{\mathcal{L}}(\theta)$ as follows,

$$
\tilde{\mathcal{L}}^{\mathrm{adv}}(\theta) := [\ell_1(\theta) + \ell_2(\theta) + \ell_3(\theta) + \ell_4(\theta)],
$$

where

$$\ell_1(\theta) = 2\,\mathbb{E}_\tau\Big[((w_{21}^V)^\top \quad w_{22}^V) \frac{\begin{pmatrix} X_\tau & X_\tau^{\text{sfx}} & x_{\tau,q} \\ Y_\tau & Y_\tau^{\text{sfx}} & 0 \end{pmatrix}\begin{pmatrix} X_\tau & X_\tau^{\text{sfx}} & x_{\tau,q} \\ Y_\tau & Y_\tau^{\text{sfx}} & 0 \end{pmatrix}^\top}{N + M_{\text{train}}} \begin{pmatrix} W_{11}^{KQ} \\ (w_{21}^{KQ})^\top \end{pmatrix} x_{\tau,q} - y_{\tau,q}\Big]^2,$$
(A.8)

$$\ell_2(\theta) = \frac{2\epsilon^4 M_{\text{train}}^2}{(N+M_{\text{train}})^2}\|w_{21}^V\|_2^2\,\mathbb{E}_\tau\Big[\|W_{11}^{KQ}x_{\tau,q}\|_2^2\Big],$$
(A.9)

$$\ell_3(\theta) = \frac{2\epsilon^2 M_{\text{train}}}{(N+M_{\text{train}})^2}\,\mathbb{E}_\tau\Big[\|W_{11}^{KQ}x_{\tau,q}\|_2^2 \cdot \|((w_{21}^V)^\top \quad w_{22}^V)\begin{pmatrix} X_\tau^{\text{sfx}} \\ Y_\tau^{\text{sfx}} \end{pmatrix}\|_2^2\Big],$$
(A.10)

$$\ell_4(\theta) = \frac{2\epsilon^2 M_{\text{train}}}{(N+M_{\text{train}})^2}\|w_{21}^V\|_2^2 \cdot \mathbb{E}_\tau\Big[\|\begin{pmatrix} X_\tau^{\text{sfx}} \\ Y_\tau^{\text{sfx}} \end{pmatrix}^\top \begin{pmatrix} W_{11}^{KQ} \\ (w_{21}^{KQ})^\top \end{pmatrix} x_{\tau,q}\|_2^2\Big].$$
(A.11)

In the remaining of this proof, we will show that when $w_{21}^V = w_{21}^{KQ} = 0_{d\times 1}$ holds, one has: (1) $\partial_{W_{21}^V}\ell_1(\theta) = \partial_{W_{21}^{KQ}}\ell_1(\theta) = 0_{1\times d}$, (2) $\partial_{W_{21}^V}\ell_2(\theta) = \partial_{W_{21}^{KQ}}\ell_2(\theta) = 0_{1\times d}$, (3) $\partial_{W_{21}^V}\ell_3(\theta) = \partial_{W_{21}^{KQ}}\ell_3(\theta) = 0_{1\times d}$, and (4) $\partial_{W_{21}^V}\ell_4(\theta) = \partial_{W_{21}^{KQ}}\ell_4(\theta) = 0_{1\times d}$, which thus automatically indicates that $\partial_{W_{21}^V}\tilde{\mathcal{L}}^{\text{adv}}(\theta) = \partial_{W_{21}^{KQ}}\tilde{\mathcal{L}}^{\text{adv}}(\theta) = 0_{1\times d}$.

**Step 1: Show that $w_{21}^V = w_{21}^{KQ} = 0_{d\times 1}$ indicates $\partial_{W_{21}^V}\ell_1(\theta) = \partial_{W_{21}^{KQ}}\ell_1(\theta) = 0_{1\times d}$.** Such a claim can be directly obtained from the proofs in [69]. Specifically, when setting the (original) ICL prompt length from $N$ to $(N + M_{\text{train}})$, the ICL training loss $L$ in [69] is equivalent to our $\ell_1(\theta)$ defined in Eq. (A.8). Therefore, one can then follow the same procedures as those in the proof of Lemma 5.2 in [69] to show that the continuous gradient flows of $W_{21}^V$ and $W_{21}^{KQ}$ are zero when Assumption 1 holds. Please refer accordingly for details.

**Step 2: Show that $w_{21}^V = w_{21}^{KQ} = 0_{d\times 1}$ indicates $\partial_{w_{21}^V}\ell_2(\theta) = \partial_{w_{21}^{KQ}}\ell_2(\theta) = 0_{1\times d}$.** Since the term $w_{21}^{KQ}$ does not exist in the expression of $\ell_2(\theta)$ in Eq. (A.9), we directly have that $\partial_{w_{21}^{KQ}}\ell_2(\theta) = 0_{1\times d}$. Besides, for the derivative $\partial_{w_{21}^V}\ell_2(\theta)$, based on Eq. (A.9) we further have that

$$\partial_{w_{21}^V}\ell_2(\theta)\Big|_{w_{21}^V=0_{d\times 1}} = \partial_{w_{21}^V}\Big[\frac{2\epsilon^4 M_{\text{train}}^2}{(N+M_{\text{train}})^2}\cdot\|w_{21}^V\|_2^2\cdot\mathbb{E}_\tau\|W_{11}^{KQ}x_{\tau,q}\|_2^2\Big]\Big|_{w_{21}^V=0_{d\times 1}}$$

$$= \Big[\frac{4\epsilon^4 M_{\text{train}}^2}{(N+M_{\text{train}})^2}\cdot\mathbb{E}_\tau\|W_{11}^{KQ}x_{\tau,q}\|_2^2\cdot(w_{21}^V)^\top\Big]\Big|_{w_{21}^V=0_{d\times 1}}$$

$$= \frac{4\epsilon^4 M_{\text{train}}^2}{(N+M_{\text{train}})^2}\cdot\mathbb{E}_\tau\|W_{11}^{KQ}x_{\tau,q}\|_2^2\cdot 0_{d\times 1}^\top = 0_{1\times d},$$

which justifies our claim in Step 2.

**Step 3: Show that $w_{21}^V = w_{21}^{KQ} = 0_{d\times 1}$ indicates $\partial_{w_{21}^V}\ell_3(\theta) = \partial_{w_{21}^{KQ}}\ell_3(\theta) = 0_{1\times d}$.** We first rewrite $\ell_3(\theta)$ that defined in Eq. (A.10) as follows,

$$\ell_3(\theta) = \frac{2\epsilon^2 M_{\text{train}}}{(N+M_{\text{train}})^2}\cdot\mathbb{E}_\tau\Big[\|W_{11}^{KQ}x_{\tau,q}\|_2^2\cdot\|((w_{21}^V)^\top \quad w_{22}^V)\begin{pmatrix} X_\tau^{\text{sfx}} \\ Y_\tau^{\text{sfx}} \end{pmatrix}\|_2^2\Big]$$

$$= \frac{2\epsilon^2 M_{\text{train}}}{(N+M_{\text{train}})^2}\cdot\mathbb{E}_\tau\Big[\|W_{11}^{KQ}x_{\tau,q}\|_2^2\Big]\cdot\sum_{i=1}^{M_{\text{train}}}\mathbb{E}_\tau\Big[((w_{21}^V)^\top \quad w_{22}^V)\cdot\begin{pmatrix} x_{\tau,i}^{\text{sfx}} \\ y_{\tau,i}^{\text{sfx}} \end{pmatrix}\begin{pmatrix} x_{\tau,i}^{\text{sfx}} \\ y_{\tau,i}^{\text{sfx}} \end{pmatrix}^\top\cdot((w_{21}^V)^\top \quad w_{22}^V)^\top\Big]$$

$$= \frac{2\epsilon^2 M_{\text{train}}}{(N+M_{\text{train}})^2}\cdot\mathbb{E}_\tau\Big[\|W_{11}^{KQ}x_{\tau,q}\|_2^2\Big]\cdot((w_{21}^V)^\top \quad w_{22}^V)\cdot\Big(\sum_{i=1}^{M_{\text{train}}}\mathbb{E}_\tau\Big[\begin{pmatrix} x_{\tau,i}^{\text{sfx}} \\ y_{\tau,i}^{\text{sfx}} \end{pmatrix}\begin{pmatrix} x_{\tau,i}^{\text{sfx}} \\ y_{\tau,i}^{\text{sfx}} \end{pmatrix}^\top\Big]\Big)\cdot((w_{21}^V)^\top \quad w_{22}^V)^\top.$$
(A.12)

Then, for any $i \in [M]$ we have

$$\mathbb{E}_\tau\left[\begin{pmatrix} x_{\tau,i}^{\text{sfx}} \\ y_{\tau,i}^{\text{sfx}} \end{pmatrix}\begin{pmatrix} x_{\tau,i}^{\text{sfx}} \\ y_{\tau,i}^{\text{sfx}} \end{pmatrix}^\top\right] = \mathbb{E}_{w_\tau, x_{\tau,i}^{\text{sfx}}}\begin{pmatrix} x_{\tau,i}^{\text{sfx}} \cdot (x_{\tau,i}^{\text{sfx}})^\top & x_{\tau,i}^{\text{sfx}} \cdot (w_\tau^\top x_{\tau,i}^{\text{sfx}})^\top \\ w_\tau^\top x_{\tau,i}^{\text{sfx}} \cdot (x_{\tau,i}^{\text{sfx}})^\top & w_\tau^\top x_{\tau,i}^{\text{sfx}} \cdot (w_\tau^\top x_{\tau,i}^{\text{sfx}})^\top \end{pmatrix}$$

$$= \begin{pmatrix} \Lambda & \Lambda \cdot 0_{d\times 1} \\ 0_{1\times d} \cdot \Lambda & \mathbb{E}_{w_\tau}\left[w_\tau^\top \Lambda w_\tau\right] \end{pmatrix} = \begin{pmatrix} \Lambda & 0_{d\times 1} \\ 0_{1\times d} & \underbrace{\text{Tr}(I_d \Lambda)}_{\text{by Lemma A.2}} \end{pmatrix} = \begin{pmatrix} \Lambda & 0_{d\times 1} \\ 0_{1\times d} & \text{Tr}(\Lambda) \end{pmatrix}. \qquad \text{(A.13)}$$

Finally, by inserting Eq. (A.13) into Eq. (A.12), $\ell_3(\theta)$ can thus be simplified as follows,

$$\ell_3(\theta) = \frac{2\epsilon^2 M_{\text{train}}}{(N + M_{\text{train}})^2} \cdot \mathbb{E}_\tau\left[\|W_{11}^{KQ} x_{\tau,q}\|_2^2\right] \cdot ((w_{21}^V)^\top \quad w_{22}^V) \cdot \left(\sum_{i=1}^{M_{\text{train}}}\begin{pmatrix} \Lambda & 0_{d\times 1} \\ 0_{1\times d} & \text{Tr}(\Lambda) \end{pmatrix}\right) \cdot ((w_{21}^V)^\top \quad w_{22}^V)^\top$$

$$= \frac{2\epsilon^2 M_{\text{train}}^2}{(N + M_{\text{train}})^2} \cdot \mathbb{E}_\tau\left[\|W_{11}^{KQ} x_{\tau,q}\|_2^2\right] \cdot \left((w_{21}^V)^\top \Lambda w_{21}^V + \text{Tr}(\Lambda)(w_{22}^V)^2\right). \qquad \text{(A.14)}$$

According to Eq. (A.14), $\ell_3(\theta)$ does not depend on $w_{21}^{KQ}$, which means that $\partial_{w_{21}^{KQ}} \ell_3(\theta) = 0_{1\times d}$. On the other hand, based on Eq. (A.14), when $w_{21}^V = 0$, the derivative of $\ell_3(\theta)$ with respect to $w_{21}^V$ is calculated as follows,

$$\partial_{w_{21}^V} \ell_3(\theta)\Big|_{w_{21}^V=0} = \partial_{w_{21}^V}\left[\frac{2\epsilon^2 M_{\text{train}}^2}{(N + M_{\text{train}})^2} \cdot \mathbb{E}_\tau\left[\|W_{11}^{KQ} x_{\tau,q}\|_2^2\right] \cdot \left((w_{21}^V)^\top \Lambda w_{21}^V + \text{Tr}(\Lambda)(w_{22}^V)^2\right)\right]\Big|_{w_{21}^V=0}$$

$$= \frac{2\epsilon^2 M_{\text{train}}^2}{(N + M_{\text{train}})^2} \cdot \mathbb{E}_\tau\left[\|W_{11}^{KQ} x_{\tau,q}\|_2^2\right] \cdot \partial_{w_{21}^V}\left[(w_{21}^V)^\top \Lambda w_{21}^V\right]\Big|_{w_{21}^V=0}$$

$$= \frac{4\epsilon^2 M_{\text{train}}^2}{(N + M_{\text{train}})^2} \cdot \mathbb{E}_\tau\left[\|W_{11}^{KQ} x_{\tau,q}\|_2^2\right] \cdot \left[(w_{21}^V)^\top \Lambda\right]\Big|_{w_{21}^V=0}$$

$$= \frac{4\epsilon^2 M_{\text{train}}^2}{(N + M_{\text{train}})^2} \cdot \mathbb{E}_\tau\left[\|W_{11}^{KQ} x_{\tau,q}\|_2^2\right] \cdot 0_{d\times 1}^\top \Lambda = 0_{1\times d},$$

which justifies our claim in Step 3.

**Step 4: Show that $w_{21}^V = w_{21}^{KQ} = 0_{d\times 1}$ indicates $\partial_{w_{21}^V} \ell_4(\theta) = \partial_{w_{21}^{KQ}} \ell_4(\theta) = 0_{1\times d}$.** When $w_{21}^V = w_{21}^{KQ} = 0_{d\times 1}$, based on the expression of $\ell_4(\theta)$ given in Eq. (A.11), the derivative of $\ell_4(\theta)$ with respect to $w_{21}^V$ is calculated as follows,

$$\partial_{w_{21}^V} \ell_4(\theta)\Big|_{w_{21}^V=w_{21}^{KQ}=0_{d\times 1}} = \partial_{w_{21}^V}\left[\frac{2\epsilon^2 M_{\text{train}}}{(N + M_{\text{train}})^2}\|w_{21}^V\|_2^2 \cdot \mathbb{E}_\tau\left\|\begin{pmatrix} X_\tau^{\text{sfx}} \\ Y_\tau^{\text{sfx}} \end{pmatrix}^\top\begin{pmatrix} W_{11}^{KQ} \\ (w_{21}^{KQ})^\top \end{pmatrix} x_{\tau,q}\right\|_2^2\right]\Big|_{w_{21}^V=w_{21}^{KQ}=0_{d\times 1}}$$

$$= \left[\frac{4\epsilon^2 M_{\text{train}}}{(N + M_{\text{train}})^2} \cdot \mathbb{E}_\tau\left\|\begin{pmatrix} X_\tau^{\text{sfx}} \\ Y_\tau^{\text{sfx}} \end{pmatrix}^\top\begin{pmatrix} W_{11}^{KQ} \\ (w_{21}^{KQ})^\top \end{pmatrix} x_{\tau,q}\right\|_2^2 \cdot (w_{21}^V)^\top\right]\Big|_{w_{21}^V=w_{21}^{KQ}=0_{d\times 1}}$$

$$= \frac{4\epsilon^2 M_{\text{train}}}{(N + M_{\text{train}})^2} \cdot \mathbb{E}_\tau\left\|\begin{pmatrix} X_\tau^{\text{sfx}} \\ Y_\tau^{\text{sfx}} \end{pmatrix}^\top\begin{pmatrix} W_{11}^{KQ} \\ 0_{d\times 1}^\top \end{pmatrix} x_{\tau,q}\right\|_2^2 \cdot 0_{d\times 1}^\top = 0_{1\times d}.$$

Besides, for the derivative of $\ell_4(\theta)$ with respect to $w_{21}^{KQ}$, we also have that

$$\partial_{w_{21}^{KQ}} \ell_4(\theta)\Big|_{w_{21}^V=w_{21}^{KQ}=0_{d\times 1}} = \partial_{w_{21}^{KQ}}\left[\frac{2\epsilon^2 M_{\text{train}}}{(N + M_{\text{train}})^2}\|w_{21}^V\|_2^2 \cdot \mathbb{E}_\tau\left\|\begin{pmatrix} X_\tau^{\text{sfx}} \\ Y_\tau^{\text{sfx}} \end{pmatrix}^\top\begin{pmatrix} W_{11}^{KQ} \\ (w_{21}^{KQ})^\top \end{pmatrix} x_{\tau,q}\right\|_2^2\right]\Big|_{w_{21}^V=w_{21}^{KQ}=0_{d\times 1}}$$

$$= \left[\frac{2\epsilon^2 M_{\text{train}}}{(N + M_{\text{train}})^2} \cdot \|w_{21}^V\|_2^2 \cdot \partial_{w_{21}^{KQ}} \mathbb{E}_\tau\left\|\begin{pmatrix} X_\tau^{\text{sfx}} \\ Y_\tau^{\text{sfx}} \end{pmatrix}^\top\begin{pmatrix} W_{11}^{KQ} \\ (w_{21}^{KQ})^\top \end{pmatrix} x_{\tau,q}\right\|_2^2\right]\Big|_{w_{21}^V=w_{21}^{KQ}=0_{d\times 1}}$$

$$= \frac{2\epsilon^2 M_{\text{train}}}{(N + M_{\text{train}})^2} \cdot \|0_{d\times 1}\|_2^2 \cdot \partial_{w_{21}^{KQ}}\left[\mathbb{E}_\tau\left\|\begin{pmatrix} X_\tau^{\text{sfx}} \\ Y_\tau^{\text{sfx}} \end{pmatrix}^\top\begin{pmatrix} W_{11}^{KQ} \\ (w_{21}^{KQ})^\top \end{pmatrix} x_{\tau,q}\right\|_2^2\right]\Big|_{w_{21}^{KQ}=0_{d\times 1}} = 0_{1\times d}.$$

The above two equations justify the claim in Step 4.

**Step 5:** Based on results from previous Steps 1 to 4, we eventually have that

$$\partial_{w_{21}^V}\tilde{\mathcal{L}}^{\text{adv}}(\theta)\Big|_{w_{21}^V=w_{21}^{KQ}=0_{d\times 1}} = \partial_{w_{21}^V}[\ell_1(\theta)+\ell_2(\theta)+\ell_3(\theta)+\ell_4(\theta)]\Big|_{w_{21}^V=w_{21}^{KQ}=0_{d\times 1}} = \sum_{i=1}^{4}0_{1\times d} = 0_{1\times d},$$

$$\partial_{w_{21}^{KQ}}\tilde{\mathcal{L}}^{\text{adv}}(\theta)\Big|_{w_{21}^V=w_{21}^{KQ}=0_{d\times 1}} = \partial_{w_{21}^{KQ}}[\ell_1(\theta)+\ell_2(\theta)+\ell_3(\theta)+\ell_4(\theta)]\Big|_{w_{21}^V=w_{21}^{KQ}=0_{d\times 1}} = \sum_{i=1}^{4}0_{1\times d} = 0_{1\times d}.$$

The proof is completed. $\qquad\square$

With Lemma A.6, we can then simplify the surrogate AT loss $\tilde{\mathcal{L}}^{\text{adv}}(\theta)$, as shown in the following Lemma A.7.

**Lemma A.7.** *Under Assumption 1, the surrogate AT loss $\tilde{\mathcal{L}}^{\text{adv}}(\theta)$ defined in Eq. (9) can be simplified as follows,*

$$\tilde{\mathcal{L}}^{\text{adv}}(\theta) = 2\text{Tr}\Big[(\Gamma(M_{\text{train}})\Lambda + \epsilon^2\psi(M_{\text{train}})I_d)\cdot(w_{22}^V W_{11}^{KQ}\Lambda^{\frac{1}{2}})\cdot(w_{22}^V W_{11}^{KQ}\Lambda^{\frac{1}{2}})^\top\Big]$$

$$- 4\text{Tr}\Big[(w_{22}^V W_{11}^{KQ}\Lambda^{\frac{1}{2}})\cdot\Lambda^{\frac{3}{2}}\Big] + 2\text{Tr}(\Lambda),$$

*where $\Gamma(M) := \frac{N+M+1}{N+M}\Lambda + \frac{\text{Tr}(\Lambda)}{N+M}I_d$ and $\psi(M) := \frac{M^2\text{Tr}(\Lambda)}{(N+M)^2}$ are same functions as that defined in Eq. (10).*

*Proof.* When Assumption 1 holds, by applying Lemma A.6, one can substitute terms $w_{21}^V$ and $w_{21}^{KQ}$ in the surrogate AT loss $\tilde{\mathcal{L}}^{\text{adv}}(\theta)$ with the zero vector $0_{d\times 1}$, which thus simplifies $\tilde{\mathcal{L}}^{\text{adv}}(\theta)$ as follows,

$$\tilde{\mathcal{L}}^{\text{adv}}(\theta) = 2\,\mathbb{E}_\tau\Bigg[\begin{pmatrix}0_{1\times d} & w_{22}^V\end{pmatrix}\frac{\begin{pmatrix}X_\tau & X_\tau^{\text{sfx}} & x_{\tau,q}\\ Y_\tau & Y_\tau^{\text{sfx}} & 0\end{pmatrix}\begin{pmatrix}X_\tau & X_\tau^{\text{sfx}} & x_{\tau,q}\\ Y_\tau & Y_\tau^{\text{sfx}} & 0\end{pmatrix}^\top}{N+M_{\text{train}}}\begin{pmatrix}W_{11}^{KQ}\\ 0_{1\times d}\end{pmatrix}x_{\tau,q}-y_{\tau,q}\Bigg]^2$$

$$+ 0 + \frac{2\epsilon^2 M_{\text{train}}}{(N+M_{\text{train}})^2}\,\mathbb{E}_\tau\Big[\|W_{11}^{KQ}x_{\tau,q}\|_2^2\cdot\|\begin{pmatrix}0_{1\times d} & w_{22}^V\end{pmatrix}\begin{pmatrix}X_\tau^{\text{sfx}}\\ Y_\tau^{\text{sfx}}\end{pmatrix}\|_2^2\Big] + 0$$

$$= \underbrace{2\cdot\mathbb{E}_\tau\Big[w_{22}^V\cdot\frac{Y_\tau X_\tau + Y_\tau^{\text{sfx}}X_\tau^{\text{sfx}}}{N+M_{\text{train}}}\cdot W_{11}^{KQ}x_{\tau,q}-y_{\tau,q}\Big]^2}_{:=B_1(\theta)}$$

$$+ \underbrace{\frac{2\epsilon^2 M_{\text{train}}}{(N+M_{\text{train}})^2}\cdot\mathbb{E}_\tau\Big[\|W_{11}^{KQ}x_{\tau,q}\|_2^2\cdot\|w_{22}^V Y_\tau^{\text{sfx}}\|_2^2\Big]}_{:=B_2(\theta)}. \tag{A.15}$$

**For the term $B_1(\theta)$ in Eq. (A.15), we have that**

$$B_1(\theta) := 2\cdot\mathbb{E}_\tau\Big[w_{22}^V\cdot\frac{Y_\tau X_\tau^\top + Y_\tau^{\text{sfx}}(X_\tau^{\text{sfx}})^\top}{N+M_{\text{train}}}\cdot W_{11}^{KQ}x_{\tau,q}-y_{\tau,q}\Big]^2$$

$$= 2\cdot\mathbb{E}_\tau\Big[\frac{w_\tau^\top\cdot(X_\tau X_\tau^\top + X_\tau^{\text{sfx}}(X_\tau^{\text{sfx}})^\top)}{N+M_{\text{train}}}\cdot w_{22}^V W_{11}^{KQ}\cdot x_{\tau,q}-w_\tau^\top x_{\tau,q}\Big]^2$$

$$= 2\cdot\mathbb{E}_\tau\Big[\Big[\frac{X_\tau X_\tau^\top + X_\tau^{\text{sfx}}(X_\tau^{\text{sfx}})^\top}{N+M_{\text{train}}}\cdot w_{22}^V W_{11}^{KQ}x_{\tau,q}-x_{\tau,q}\Big]^\top\cdot w_\tau w_\tau^\top\cdot\Big[\frac{X_\tau X_\tau^\top + X_\tau^{\text{sfx}}(X_\tau^{\text{sfx}})^\top}{N+M_{\text{train}}}\cdot w_{22}^V W_{11}^{KQ}x_{\tau,q}-x_{\tau,q}\Big]\Big]$$

$$= 2\cdot\mathbb{E}_\tau\Big[\Big[\frac{X_\tau X_\tau^\top + X_\tau^{\text{sfx}}(X_\tau^{\text{sfx}})^\top}{N+M_{\text{train}}}\cdot w_{22}^V W_{11}^{KQ}x_{\tau,q}-x_{\tau,q}\Big]^\top\cdot I_d\cdot\Big[\frac{X_\tau X_\tau^\top + X_\tau^{\text{sfx}}(X_\tau^{\text{sfx}})^\top}{N+M_{\text{train}}}\cdot w_{22}^V W_{11}^{KQ}x_{\tau,q}-x_{\tau,q}\Big]\Big]$$

$$= 2\cdot\mathbb{E}_\tau\Big[x_{\tau,q}^\top\cdot(w_{22}^V W_{11}^{KQ})^\top\cdot\frac{\mathbb{E}_\tau\Big[(X_\tau X_\tau^\top + X_\tau^{\text{sfx}}(X_\tau^{\text{sfx}})^\top)(X_\tau X_\tau^\top + X_\tau^{\text{sfx}}(X_\tau^{\text{sfx}})^\top)\Big]}{(N+M_{\text{train}})^2}\cdot w_{22}^V W_{11}^{KQ}\cdot x_{\tau,q}\Big]$$

$$- 4\cdot\mathbb{E}_\tau\Big[x_{\tau,q}^\top\cdot\frac{\mathbb{E}_\tau\Big[(X_\tau X_\tau^\top + X_\tau^{\text{sfx}}(X_\tau^{\text{sfx}})^\top)\Big]}{N+M_{\text{train}}}\cdot(w_{22}^V W_{11}^{KQ})\cdot x_{\tau,q}\Big] + 2\cdot\mathbb{E}_\tau\Big[x_{\tau,q}^\top\cdot x_{\tau,q}\Big]. \tag{A.16}$$

For $\mathbb{E}_\tau\left[(X_\tau X_\tau^\top + X_\tau^{\text{sfx}}(X_\tau^{\text{sfx}})^\top)(X_\tau X_\tau^\top + X_\tau^{\text{sfx}}(X_\tau^{\text{sfx}})^\top)\right]$ in Eq. (A.16), we have

$$\mathbb{E}_\tau\left[(X_\tau X_\tau^\top + X_\tau^{\text{sfx}}(X_\tau^{\text{sfx}})^\top)(X_\tau X_\tau^\top + X_\tau^{\text{sfx}}(X_\tau^{\text{sfx}})^\top)\right]$$

$$= \mathbb{E}_\tau[X_\tau X_\tau^\top X_\tau X_\tau^\top] + \mathbb{E}_\tau[X_\tau^{\text{sfx}}(X_\tau^{\text{sfx}})^\top] \cdot \mathbb{E}_\tau[X_\tau X_\tau^\top]$$

$$+ \mathbb{E}_\tau[X_\tau X_\tau^\top] \cdot \mathbb{E}_\tau[X_\tau^{\text{sfx}}(X_\tau^{\text{sfx}})^\top] + \mathbb{E}_\tau[X_\tau^{\text{sfx}}(X_\tau^{\text{sfx}})^\top X_\tau^{\text{sfx}}(X_\tau^{\text{sfx}})^\top]$$

$$= \mathbb{E}_\tau\left[\sum_{i,j} x_{\tau,i} x_{\tau,i}^\top x_{\tau,j} x_{\tau,j}^\top\right] + \mathbb{E}_\tau\left[\sum_i x_{\tau,i}^{\text{sfx}}(x_{\tau,i}^{\text{sfx}})^\top\right] \cdot \mathbb{E}_\tau\left[\sum_i x_{\tau,i} x_{\tau,i}^\top\right]$$

$$+ \mathbb{E}_\tau\left[\sum_i x_{\tau,i} x_{\tau,i}^\top\right] \cdot \mathbb{E}_\tau\left[x_{\tau,i}^{\text{sfx}}(x_{\tau,i}^{\text{sfx}})^\top\right] + \mathbb{E}_\tau\left[\sum_{i,j} x_{\tau,i}^{\text{sfx}}(x_{\tau,i}^{\text{sfx}})^\top x_{\tau,j}^{\text{sfx}}(x_{\tau,j}^{\text{sfx}})^\top\right]$$

$$= \mathbb{E}_\tau\left[\sum_i x_{\tau,i} x_{\tau,i}^\top x_{\tau,i} x_{\tau,i}^\top + \sum_{1 \leq i,j \leq N, i \neq j} \Lambda^2\right] + M_{\text{train}}\Lambda \cdot N\Lambda + N\Lambda \cdot M_{\text{train}}\Lambda$$

$$+ \mathbb{E}_\tau\left[\sum_i x_{\tau,i}^{\text{sfx}}(x_{\tau,i}^{\text{sfx}})^\top x_{\tau,i}^{\text{sfx}}(x_{\tau,i}^{\text{sfx}})^\top + \sum_{1 \leq i,j \leq M_{\text{train}}, i \neq j} \Lambda^2\right]$$

$$= \mathbb{E}_\tau\left[\sum_{i=1}^N \underbrace{(2\Lambda^2 + \text{Tr}(\Lambda)\Lambda)}_{\text{by Lemma A.1}}\right] + (N^2 - N) \cdot \Lambda^2 + 2NM_{\text{train}} \cdot \Lambda^2$$

$$+ \mathbb{E}_\tau\left[\sum_{i=1}^{M_{\text{train}}} \underbrace{(2\Lambda^2 + \text{Tr}(\Lambda)\Lambda)}_{\text{by Lemma A.1}}\right] + (M_{\text{train}}^2 - M_{\text{train}}) \cdot \Lambda^2$$

$$= (N^2 + N + M_{\text{train}}^2 + M_{\text{train}} + 2NM_{\text{train}}) \cdot \Lambda^2 + (N + M_{\text{train}}) \cdot \text{Tr}(\Lambda) \cdot \Lambda$$

$$= (N + M_{\text{train}}) \cdot ((N + M_{\text{train}} + 1) \cdot \Lambda^2 + \text{Tr}(\Lambda) \cdot \Lambda) = (N + M_{\text{train}})^2 \cdot \Gamma(M_{\text{train}})\Lambda. \quad \text{(A.17)}$$

For $\mathbb{E}_\tau\left[X_\tau X_\tau^\top + X_\tau^{\text{sfx}}(X_\tau^{\text{sfx}})^\top\right]$ in Eq. (A.16), we have

$$\mathbb{E}_\tau\left[X_\tau X_\tau^\top + X_\tau^{\text{sfx}}(X_\tau^{\text{sfx}})^\top\right]$$

$$= \mathbb{E}_\tau\left[\sum_i x_{\tau,i} x_{\tau,i}^\top\right] + \mathbb{E}_\tau\left[\sum_i x_{\tau,i}^{\text{sfx}}(x_{\tau,i}^{\text{sfx}})^\top\right]$$

$$= N\Lambda + M_{\text{train}}\Lambda = (N + M_{\text{train}}) \cdot \Lambda. \quad \text{(A.18)}$$

Inserting Eqs. (A.17) and (A.18) into Eq. (A.16) leads to

$$B_1(\theta) = 2 \cdot \mathbb{E}_\tau\left[x_{\tau,q}^\top \cdot (w_{22}^V W_{11}^{KQ})^\top \cdot \Gamma(M_{\text{train}})\Lambda \cdot w_{22}^V W_{11}^{KQ} \cdot x_{\tau,q}\right]$$

$$- 4 \cdot \mathbb{E}_\tau\left[x_{\tau,q}^\top \cdot \Lambda \cdot w_{22}^V W_{11}^{KQ} \cdot x_{\tau,q}\right] + 2 \cdot \mathbb{E}_\tau\left[x_{\tau,q}^\top \cdot x_{\tau,q}\right].$$

$$= 2 \cdot \underbrace{\text{Tr}\left[(w_{22}^V W_{11}^{KQ})^\top \cdot \Gamma(M_{\text{train}})\Lambda \cdot w_{22}^V W_{11}^{KQ} \cdot \Lambda\right]}_{\text{by Lemma A.2}}$$

$$- 4 \cdot \underbrace{\text{Tr}\left[\Lambda \cdot w_{22}^V W_{11}^{KQ} \cdot \Lambda\right]}_{\text{by Lemma A.2}} + 2 \cdot \text{Tr}(\Lambda)$$

$$= 2 \cdot \underbrace{\text{Tr}\left[\Gamma(M_{\text{train}})\Lambda \cdot (w_{22}^V W_{11}^{KQ}\Lambda^{\frac{1}{2}}) \cdot (w_{22}^V W_{11}^{KQ}\Lambda^{\frac{1}{2}})^\top\right]}_{\text{by Lemma A.3}}$$

$$- 4 \cdot \underbrace{\text{Tr}\left[(w_{22}^V W_{11}^{KQ}\Lambda^{\frac{1}{2}}) \cdot \Lambda^{\frac{3}{2}}\right]}_{\text{by Lemma A.3}} + 2 \cdot \text{Tr}(\Lambda). \quad \text{(A.19)}$$

**Besides, for the term $B_2(\theta)$ in Eq. (A.15), we have that**

$$B_2(\theta) := \frac{2\epsilon^2 M_{\text{train}}}{(N + M_{\text{train}})^2} \cdot \mathbb{E}_\tau \Big[ \|W_{11}^{KQ} x_{\tau,q}\|_2^2 \cdot \|w_{22}^V Y_\tau^{\text{sfx}}\|_2^2 \Big]$$

$$= \frac{2\epsilon^2 M_{\text{train}}}{(N + M_{\text{train}})^2} \cdot (w_{22}^V)^2 \cdot \mathbb{E}_\tau \Big[ x_{\tau,q}^\top \cdot (W_{11}^{KQ})^\top W_{11}^{KQ} \cdot x_{\tau,q} \Big] \cdot \mathbb{E}_\tau \Big[ w_\tau^\top \cdot X_\tau^{\text{sfx}} (X_\tau^{\text{sfx}})^\top \cdot w_\tau \Big]$$

$$= \frac{2\epsilon^2 M_{\text{train}}}{(N + M_{\text{train}})^2} \cdot (w_{22}^V)^2 \cdot \underbrace{\text{Tr}\Big[ (W_{11}^{KQ})^\top W_{11}^{KQ} \cdot \Lambda \Big]}_{\text{by Lemma A.2}} \cdot \mathbb{E}_\tau \Big[ w_\tau^\top \cdot M_{\text{train}} \Lambda \cdot w_\tau \Big]$$

$$= \frac{2\epsilon^2 M_{\text{train}}}{(N + M_{\text{train}})^2} \cdot (w_{22}^V)^2 \cdot \underbrace{\text{Tr}\Big[ W_{11}^{KQ} \cdot \Lambda \cdot (W_{11}^{KQ})^\top \Big]}_{\text{by Lemma A.3}} \cdot \underbrace{\text{Tr}\Big[ M_{\text{train}} \Lambda \cdot I_d \Big]}_{\text{by Lemma A.2}}$$

$$= 2\epsilon^2 \cdot \frac{M_{\text{train}}^2 \text{Tr}(\Lambda)}{(N + M_{\text{train}} \Lambda^{\frac{1}{2}})^2} \cdot \text{Tr}\Big[ (w_{22}^V W_{11}^{KQ} \Lambda^{\frac{1}{2}}) \cdot \Lambda \cdot (w_{22}^V W_{11}^{KQ})^\top \Big]$$

$$= 2\epsilon^2 \cdot \psi(M_{\text{train}}) \cdot \text{Tr}\Big[ (w_{22}^V W_{11}^{KQ} \Lambda^{\frac{1}{2}}) \cdot \Lambda \cdot (w_{22}^V W_{11}^{KQ})^\top \Big]. \tag{A.20}$$

**Finally, by inserting Eqs. (A.19) and (A.20) into Eq. (A.15), we have**

$$\tilde{\mathcal{L}}^{\text{adv}}(\theta)$$

$$:= 2 \cdot \text{Tr}\Big[ \Gamma(M_{\text{train}})\Lambda \cdot (w_{22}^V W_{11}^{KQ} \Lambda^{\frac{1}{2}}) \cdot (w_{22}^V W_{11}^{KQ} \Lambda^{\frac{1}{2}})^\top \Big] - 4 \cdot \text{Tr}\Big[ (w_{22}^V W_{11}^{KQ} \Lambda^{\frac{1}{2}}) \cdot \Lambda^{\frac{3}{2}} \Big]$$

$$+ 2 \cdot \text{Tr}(\Lambda) + 2\epsilon^2 \cdot \psi(M_{\text{train}}) \cdot \text{Tr}\Big[ (w_{22}^V W_{11}^{KQ} \Lambda^{\frac{1}{2}}) \cdot \Lambda \cdot (w_{22}^V W_{11}^{KQ})^\top \Big]$$

$$= 2 \cdot \text{Tr}\Big[ (\Gamma(M_{\text{train}})\Lambda + \epsilon^2 \psi(M_{\text{train}})I_d) \cdot (w_{22}^V W_{11}^{KQ} \Lambda^{\frac{1}{2}}) \cdot (w_{22}^V W_{11}^{KQ} \Lambda^{\frac{1}{2}})^\top \Big]$$

$$- 4 \cdot \text{Tr}\Big[ (w_{22}^V W_{11}^{KQ} \Lambda^{\frac{1}{2}}) \cdot \Lambda^{\frac{3}{2}} \Big] + 2 \cdot \text{Tr}(\Lambda),$$

which completes the proof. $\qquad\qquad\square$

Based on the simplified surrogate AT loss, the closed-form global minimizer $\theta_*$ for the surrogate AT problem is then calculated in the following Lemma A.8.

**Lemma A.8.** *Suppose Assumption 1 holds. Then, $\theta_* := (W_*^V W_*^{KQ})$ is a minimizer for the surrogate AT loss $\tilde{\mathcal{L}}^{\text{adv}}(\theta)$ in Eq. (8) if and only if $w_{*,22}^V W_{*,11}^{KQ} = (\Gamma(M_{\text{train}})\Lambda + \epsilon^2 \psi(M_{\text{train}})I_d)^{-1}\Lambda$.*

*Proof.* For the simplified surrogate AT loss proved in Lemma A.7, we rewrite it as follows,

$$\tilde{\mathcal{L}}^{\text{adv}}(\theta)$$

$$= 2\text{Tr}\Big[ (\Gamma(M_{\text{train}})\Lambda + \epsilon^2 \psi(M_{\text{train}})I_d) \cdot (w_{22}^V W_{11}^{KQ} \Lambda^{\frac{1}{2}}) \cdot (w_{22}^V W_{11}^{KQ} \Lambda^{\frac{1}{2}})^\top \Big]$$

$$- 4\text{Tr}\Big[ (w_{22}^V W_{11}^{KQ} \Lambda^{\frac{1}{2}}) \cdot \Lambda^{\frac{3}{2}} \Big] + 2\text{Tr}(\Lambda)$$

$$= 2 \cdot \text{Tr}\Big[ (\Gamma_{\text{train}}\Lambda + \epsilon^2 \psi_{\text{train}} I_d) \cdot \Big( w_{22}^V W_{11}^{KQ} \Lambda^{\frac{1}{2}} - (\Gamma_{\text{train}}\Lambda + \epsilon^2 \psi_{\text{train}} I_d)^{-1} \Lambda^{\frac{3}{2}} \Big)$$

$$\cdot \Big( w_{22}^V W_{11}^{KQ} \Lambda^{\frac{1}{2}} - (\Gamma_{\text{train}}\Lambda + \epsilon^2 \psi_{\text{train}} I_d)^{-1} \Lambda^{\frac{3}{2}} \Big)^\top \Big]$$

$$- \text{Tr}\Big[ \Lambda^3 (\Gamma_{\text{train}}\Lambda + \epsilon^2 \psi_{\text{train}} I_d)^{-1} \Big] + 2 \cdot \text{Tr}(\Lambda), \tag{A.21}$$

where $\Gamma_{\text{train}} := \Gamma(M_{\text{train}})$ and $\psi_{\text{train}} := \psi(M_{\text{train}})$.

Notice that the second and third terms in Eq. (A.21) are constants. Besides, the matrix $(\Gamma_{\text{train}}\Lambda + \epsilon^2 \psi I_d)$ in the first term in Eq. (A.21) is positive definite, which means that this first term is non-negative. As a result, the surrogate AT loss $\tilde{\mathcal{L}}^{\text{adv}}(\theta)$ will be minimized when the first term in Eq. (A.21) becomes zero. This can be achieved by setting

$$w_{*,22}^V W_{*,11}^{KQ} \Lambda^{\frac{1}{2}} - (\Gamma(M_{\text{train}})\Lambda + \epsilon^2 \psi(M_{\text{train}})I_d)^{-1} \Lambda^{\frac{3}{2}} = 0,$$

which is

$$w_{*,22}^V W_{*,11}^{KQ} = (\Gamma(M_{\text{train}})\Lambda + \epsilon^2 \psi(M_{\text{train}})I_d)^{-1}\Lambda.$$

The proof is completed. $\qquad\square$

We now turn to prove an PL-inequality for the surrogate AT problem. The proof idea follows that in [69]. Specifically, we will first prove several technical lemmas (*i.e.*, Lemma A.9, Lemma A.10, and Lemma A.11), and then present the PL-inequality in Lemma A.12, which can then enable the surrogate AT model in Eq. (9) approaches its global optimal solution.

**Lemma A.9.** *Suppose Assumption 1 holds and the model $f_{\text{LSA},\theta}$ is trained via minimizing the surrogate AT loss $\tilde{\mathcal{L}}^{\text{adv}}(\theta)$ in Eq. (9) with continuous training flow. Then, for any continuous training time $t \geq 0$, we uniformly have that*

$$(w_{22}^V(t))^2 = \text{Tr}[W_{11}^{KQ}(t)(W_{11}^{KQ}(t))^\top].$$

*Proof.* Since the model is trained via continuous gradient flow, thus $\partial_t W_{11}^{KQ}(t)$ can be calculated based on the simplified surrogate AT loss proved in Lemma A.7 as follows,

$$
\begin{aligned}
\partial_t W_{11}^{KQ}(t) &:= -\partial_{W_{11}^{KQ}}\tilde{\mathcal{L}}^{\text{adv}}(\theta) \\
&= -2 \cdot \partial_{W_{11}^{KQ}}\text{Tr}\Big[(\Gamma(M_{\text{train}})\Lambda + \epsilon^2\psi(M_{\text{train}})I_d) \cdot (w_{22}^V W_{11}^{KQ}\Lambda^{\frac{1}{2}}) \cdot (w_{22}^V W_{11}^{KQ}\Lambda^{\frac{1}{2}})^\top\Big] \\
&\quad + 4 \cdot \partial_{W_{11}^{KQ}}\text{Tr}\Big[(w_{22}^V W_{11}^{KQ}\Lambda^{\frac{1}{2}}) \cdot \Lambda^{\frac{3}{2}}\Big] \\
&= -2 \cdot (w_{22}^V)^2 \cdot \partial_{W_{11}^{KQ}}\text{Tr}\Big[(\Gamma(M_{\text{train}})\Lambda + \epsilon^2\psi(M_{\text{train}})I_d) \cdot W_{11}^{KQ} \cdot \Lambda \cdot (W_{11}^{KQ})^\top\Big] + \underbrace{4w_{22}^V\Lambda^2}_{\text{by Lemma A.4}} \\
&= \underbrace{-4 \cdot (w_{22}^V)^2 \cdot (\Gamma(M_{\text{train}})\Lambda + \epsilon^2\psi(M_{\text{train}})I_d) \cdot W_{11}^{KQ} \cdot \Lambda}_{\text{by Lemma A.4}} + 4w_{22}^V\Lambda^2. \qquad\text{(A.22)}
\end{aligned}
$$

Similarly, for $\partial_t w_{22}^V(t)$, we have

$$
\begin{aligned}
\partial_t w_{22}^V(t) &:= -\partial_{w_{22}^V}\tilde{\mathcal{L}}^{\text{adv}}(\theta) \\
&= -2 \cdot \partial_{w_{22}^V}\text{Tr}\Big[(\Gamma(M_{\text{train}})\Lambda + \epsilon^2\psi(M_{\text{train}})I_d) \cdot (w_{22}^V W_{11}^{KQ}\Lambda^{\frac{1}{2}}) \cdot (w_{22}^V W_{11}^{KQ}\Lambda^{\frac{1}{2}})^\top\Big] \\
&\quad + 4 \cdot \partial_{w_{22}^V}\text{Tr}\Big[(w_{22}^V W_{11}^{KQ}\Lambda^{\frac{1}{2}}) \cdot \Lambda^{\frac{3}{2}}\Big] \\
&= -4w_{22}^V \cdot \text{Tr}\Big[(\Gamma(M_{\text{train}})\Lambda + \epsilon^2\psi(M_{\text{train}})I_d) \cdot (W_{11}^{KQ}\Lambda^{\frac{1}{2}}) \cdot (W_{11}^{KQ}\Lambda^{\frac{1}{2}})^\top\Big] \\
&\quad + 4 \cdot \text{Tr}\Big[(W_{11}^{KQ}\Lambda^{\frac{1}{2}}) \cdot \Lambda^{\frac{3}{2}}\Big]. \qquad\text{(A.23)}
\end{aligned}
$$

Combining Eqs (A.22) and (A.23), we thus have

$$
\begin{aligned}
\text{Tr}&\Big[\partial_t W_{11}^{KQ}(t)(W_{11}^{KQ}(t))^\top\Big] \\
&= -4 \cdot (w_{22}^V)^2 \cdot \text{Tr}\Big[(\Gamma(M_{\text{train}})\Lambda + \epsilon^2\psi(M_{\text{train}})I_d) \cdot (W_{11}^{KQ}\Lambda^{\frac{1}{2}}) \cdot (W_{11}^{KQ}\Lambda^{\frac{1}{2}})^\top\Big] \\
&\quad + 4w_{22}^V \cdot \text{Tr}\Big[\Lambda^{\frac{3}{2}} \cdot (\Lambda^{\frac{1}{2}}W_{11}^{KQ})^\top\Big] \\
&= (\partial_t w_{22}^V(t))w_{22}^V(t),
\end{aligned}
$$

which further indicates that

$$
\begin{aligned}
\partial_t \text{Tr}&\Big[W_{11}^{KQ}(t)(W_{11}^{KQ}(t))^\top\Big] \\
&= \text{Tr}\Big[\partial_t W_{11}^{KQ}(t) \cdot (W_{11}^{KQ}(t))^\top\Big] + \text{Tr}\Big[W_{11}^{KQ}(t) \cdot \partial_t(W_{11}^{KQ}(t))^\top\Big] \\
&= (\partial_t w_{22}^V(t)) \cdot w_{22}^V(t) + W_{22}^V(t) \cdot (\partial_t w_{22}^V(t)) = \partial_t(w_{22}^V(t)^2). \qquad\text{(A.24)}
\end{aligned}
$$

Finally, according to Assumption 1, we have that when the continuous training time is $t = 0$,

$$\text{Tr}\left[W_{11}^{KQ}(0)(W_{11}^{KQ}(0))^\top\right] = \|W_{11}^{KQ}(0)\|_F^2 = \sigma^2 = w_{22}^V(0)^2.$$

Combine with Eq. (A.24), we thus have that

$$\text{Tr}\left[W_{11}^{KQ}(t)(W_{11}^{KQ}(t))^\top\right] = w_{22}^V(t)^2, \quad \forall t \geq 0.$$

The proof is completed. □

**Lemma A.10.** *Suppose Assumption 1 holds and the model $f_{\text{LSA},\theta}$ is trained via minimizing the surrogate AT loss $\tilde{\mathcal{L}}^{\text{adv}}(\theta)$ in Eq. (9) with continuous training flow. Then, if the parameter $\sigma$ in Assumption 1 satisfies*

$$\sigma < \sqrt{\frac{2}{d \cdot \|(\Gamma(M_{\text{train}})\Lambda + \epsilon^2 \psi(M_{\text{train}})I_d)\Lambda^{-1}\|_2}},$$

*we have $w_{22}^V(t) > 0$ holds for any continuous training time $t \geq 0$.*

*Proof.* According to the simplified AT loss calculated in Lemma A.7, we know that if $w_{22}^V(t) = 0$, then $\tilde{\mathcal{L}}^{\text{adv}}(\theta_t) = 2\text{Tr}(\Lambda)$. Besides, under Assumption 1, we have $w_{22}^V(0) = \sigma > 0$. Therefore, if we can show that $\tilde{\mathcal{L}}^{\text{adv}}(\theta_t) \neq 2\text{Tr}(\Lambda)$ for any $t \geq 0$, then it is proved that $w_{22}^V(t) > 0$ for any $t \geq 0$.

To this end, we first analyze the surrogate AT loss $\tilde{\mathcal{L}}^{\text{adv}}(\theta_t)$ at the initial training time $t = 0$. By applying Assumption 1, we have

$$
\begin{aligned}
&\tilde{\mathcal{L}}^{\text{adv}}(\theta_0) \\
&= 2\text{Tr}\left[(\Gamma(M_{\text{train}})\Lambda + \epsilon^2 \psi(M_{\text{train}})I_d) \cdot (w_{22}^V(0)W_{11}^{KQ}(0)\Lambda^{\frac{1}{2}}) \cdot (w_{22}^V(0)W_{11}^{KQ}(0)\Lambda^{\frac{1}{2}})^\top\right] \\
&\quad - 4\text{Tr}\left[(w_{22}^V(0)W_{11}^{KQ}(0)\Lambda^{\frac{1}{2}}) \cdot \Lambda^{\frac{3}{2}}\right] + 2\text{Tr}(\Lambda) \\
&= 2\text{Tr}\left[(\Gamma(M_{\text{train}})\Lambda + \epsilon^2 \psi(M_{\text{train}})I_d) \cdot (\sigma^2\Theta\Theta^\top\Lambda^{\frac{1}{2}}) \cdot (\sigma^2\Theta\Theta^\top\Lambda^{\frac{1}{2}})^\top\right] \\
&\quad - 4\text{Tr}\left[(\sigma^2\Theta\Theta^\top\Lambda^{\frac{1}{2}}) \cdot \Lambda^{\frac{3}{2}}\right] + 2\text{Tr}(\Lambda) \\
&= 2\sigma^4 \cdot \text{Tr}\left[(\Gamma(M_{\text{train}})\Lambda + \epsilon^2 \psi(M_{\text{train}})I_d)\Lambda^{-1} \cdot \Lambda\Theta\Theta^\top\Lambda\Theta\Theta^\top\right] - 4\sigma^2\|\Lambda\Theta\|_F^2 + 2\text{Tr}(\Lambda) \\
&\leq 2\sigma^4 \cdot \underbrace{d \cdot \|(\Gamma(M_{\text{train}})\Lambda + \epsilon^2 \psi(M_{\text{train}})I_d)\Lambda^{-1}\|_2 \cdot \|\Lambda\Theta\Theta^\top\Lambda\Theta\Theta^\top\|_2}_{\text{by Lemma A.5}} - 4\sigma^2\|\Lambda\Theta\|_F^2 + 2\text{Tr}(\Lambda) \\
&\leq 2\sigma^4 \cdot d \cdot \|(\Gamma(M_{\text{train}})\Lambda + \epsilon^2 \psi(M_{\text{train}})I_d)\Lambda^{-1}\|_2 \cdot \|\Lambda\Theta\Theta^\top\Lambda\|_F \cdot \|\Theta\Theta^\top\|_F \\
&\quad - 4\sigma^2\|\Lambda\Theta\|_F^2 + 2\text{Tr}(\Lambda) \\
&\leq 2\sigma^4 \cdot d \cdot \|(\Gamma(M_{\text{train}})\Lambda + \epsilon^2 \psi(M_{\text{train}})I_d)\Lambda^{-1}\|_2 \cdot \|\Lambda\Theta\|_F^2 \cdot 1 - 4\sigma^2\|\Lambda\Theta\|_F^2 + 2\text{Tr}(\Lambda) \\
&= 2\sigma^2 \cdot \|\Lambda\Theta\|_F^2 \cdot (d \cdot \sigma^2 \cdot \|(\Gamma(M_{\text{train}})\Lambda + \epsilon^2 \psi(M_{\text{train}})I_d)\Lambda^{-1}\|_2 - 2) + 2\text{Tr}(\Lambda). \quad (A.25)
\end{aligned}
$$

By Assumption 1, we have $\|\Lambda\Theta\|_F^2 > 0$. Thus, when $(d \cdot \sigma^2 \cdot \|(\Gamma(M_{\text{train}})\Lambda + \epsilon^2 \psi(M_{\text{train}})I_d)\Lambda^{-1}\|_2 - 2) < 0$, which is

$$\sigma < \sqrt{\frac{2}{d \cdot \|(\Gamma(M_{\text{train}})\Lambda + \epsilon^2 \psi(M_{\text{train}})I_d)\Lambda^{-1}\|_2}},$$

we will have $\tilde{\mathcal{L}}^{\text{adv}}(\theta_0) < \text{Tr}(\Lambda)$.

Finally, since the surrogate AT loss $\tilde{\mathcal{L}}^{\text{adv}}(\theta_t)$ is minimized with continuous gradient, thus when the above condition holds, for any $t > 0$, we always have that $\tilde{\mathcal{L}}^{\text{adv}}(\theta_t) \leq \tilde{\mathcal{L}}^{\text{adv}}(\theta_0) < \text{Tr}(\Lambda)$.

The proof is completed. □

**Lemma A.11.** *Suppose Assumption 1 holds and the $\sigma$ in Assumption 1 satisfies $\sigma < \sqrt{\frac{2}{d \cdot \|(\Gamma(M_{\text{train}})\Lambda + \epsilon^2 \psi(M_{\text{train}})I_d)\Lambda^{-1}\|_2}}$. Then, for any continuous training time $t \geq 0$, we have $(w_{22}^V(t))^2 \geq \nu > 0$, where*

$$\nu := \frac{\sigma^2 \cdot \|\Lambda\Theta\|_F^2 \cdot (2 - d \cdot \sigma^2 \cdot \|(\Gamma(M_{\text{train}})\Lambda + \epsilon^2 \psi(M_{\text{train}})I_d)\Lambda^{-1}\|_2)}{2d\|\Lambda^2\|_2} > 0.$$

*Proof.* By applying Eq. (A.25) in Lemma A.10, we have that for any $t \geq 0$,

$$2\sigma^2 \cdot \|\Lambda\Theta\|_F^2 \cdot (d \cdot \sigma^2 \cdot \|(\Gamma(M_{\text{train}})\Lambda + \epsilon^2 \psi(M_{\text{train}})I_d)\Lambda^{-1}\|_2 - 2) + 2\text{Tr}(\Lambda)$$
$$\geq \tilde{\mathcal{L}}^{\text{adv}}(\theta_0) \geq \tilde{\mathcal{L}}^{\text{adv}}(\theta_t)$$
$$= 2\text{Tr}\left[(\Gamma(M_{\text{train}})\Lambda + \epsilon^2 \psi(M_{\text{train}})I_d) \cdot (w_{22}^V W_{11}^{KQ} \Lambda^{\frac{1}{2}}) \cdot (w_{22}^V W_{11}^{KQ} \Lambda^{\frac{1}{2}})^\top\right]$$
$$\quad - 4\text{Tr}\left[(w_{22}^V W_{11}^{KQ} \Lambda^{\frac{1}{2}}) \cdot \Lambda^{\frac{3}{2}}\right] + 2\text{Tr}(\Lambda)$$
$$= 2\|(\Gamma(M_{\text{train}})\Lambda + \epsilon^2 \psi(M_{\text{train}})I_d)^{\frac{1}{2}} \cdot (w_{22}^V W_{11}^{KQ} \Lambda^{\frac{1}{2}})\|_F^2 - 4\text{Tr}\left[w_{22}^V W_{11}^{KQ} \Lambda^2\right] + 2\text{Tr}(\Lambda)$$
$$\geq 0 - \underbrace{4d \cdot |w_{22}^V| \cdot \|\Lambda^2\|_2 \cdot \|W_{11}^{KQ}\|_2}_{\text{by Lemma A.5}} + 2\text{Tr}(\Lambda),$$

which indicates

$$2\sigma^2 \cdot \|\Lambda\Theta\|_F^2 \cdot (d \cdot \sigma^2 \cdot \|(\Gamma(M_{\text{train}})\Lambda + \epsilon^2 \psi(M_{\text{train}})I_d)\Lambda^{-1}\|_2 - 2) \geq -4d \cdot |w_{22}^V| \cdot \|\Lambda^2\|_2 \cdot \|W_{11}^{KQ}\|_F,$$

thus

$$|w_{22}^V| \cdot \|W_{11}^{KQ}\|_F \geq \frac{\sigma^2 \cdot \|\Lambda\Theta\|_F^2 \cdot (2 - d \cdot \sigma^2 \cdot \|(\Gamma(M_{\text{train}})\Lambda + \epsilon^2 \psi(M_{\text{train}})I_d)\Lambda^{-1}\|_2)}{2d\|\Lambda^2\|_2}. \quad (A.26)$$

Besides, by combining Lemma A.9 and Lemma A.10, we know that

$$w_{22}^V(t) = \sqrt{\text{Tr}[W_{11}^{KQ}(t)(W_{11}^{KQ}(t))^\top]} = \sqrt{\|W_{11}^{KQ}(t)\|_F^2} = \|W_{11}^{KQ}(t)\|_F. \quad (A.27)$$

Finally, inserting Eq. (A.27) into Eq. (A.26), we thus have

$$(w_{22}^V(t))^2 \geq \frac{\sigma^2 \cdot \|\Lambda\Theta\|_F^2 \cdot (2 - d \cdot \sigma^2 \cdot \|(\Gamma(M_{\text{train}})\Lambda + \epsilon^2 \psi(M_{\text{train}})I_d)\Lambda^{-1}\|_2)}{2d\|\Lambda^2\|_2} > 0.$$

The proof is completed. $\qquad \square$

**Lemma A.12** (PL-inequality). *Suppose Assumption 1 holds and the LSA model $f_{\text{LSA},\theta}$ is trained via minimizing the surrogate AT loss $\tilde{\mathcal{L}}^{\text{adv}}(\theta)$ in Eq. (9) with continuous training flow. Suppose the $\sigma$ in Assumption 1 satisfies $\sigma < \sqrt{\frac{2}{d \cdot \|(\Gamma(M_{\text{train}})\Lambda + \epsilon^2 \psi(M_{\text{train}})I_d)\Lambda^{-1}\|_2}}$. Then for any continuous training time $t > 0$, we uniformly have that*

$$\|\partial_\theta \tilde{\mathcal{L}}^{\text{adv}}(\theta_t)\|_2^2 \geq \mu \cdot \left(\tilde{\mathcal{L}}^{adv}(\theta_t) - \min_\theta \tilde{\mathcal{L}}^{adv}(\theta)\right),$$

*where*

$$\mu := \frac{8\nu}{\|(\Gamma_{\text{train}}\Lambda + \epsilon^2 \psi_{\text{train}}I_d)^{-\frac{1}{2}}\|_F^2 \cdot \|\Lambda^{-\frac{1}{2}}\|_F^2},$$

*$\nu$ is defined in Lemma A.11, and $\text{Vec}(\cdot)$ denotes the vectorization function.*

*Proof.* From Eq. (A.22) in Lemma A.9, we have that

$$\partial_t W_{11}^{KQ}(t) = -4 \cdot (w_{22}^V)^2 \cdot (\Gamma(M_{\text{train}})\Lambda + \epsilon^2 \psi(M_{\text{train}})I_d) \cdot W_{11}^{KQ} \cdot \Lambda + 4w_{22}^V \Lambda^2$$
$$= -4w_{22}^V \cdot (\Gamma(M_{\text{train}})\Lambda + \epsilon^2 \psi(M_{\text{train}})I_d) \cdot D(\theta_t) \cdot \Lambda^{\frac{1}{2}},$$

where
$$D(\theta_t) := \left( w_{22}^V W_{11}^{KQ} \Lambda^{\frac{1}{2}} - (\Gamma(M_{\text{train}})\Lambda + \epsilon^2 \psi(M_{\text{train}})I_d)^{-1} \Lambda^{\frac{3}{2}} \right) \in \mathbb{R}^{d \times d}. \qquad (A.28)$$

As a result, the gradient norm square $\|\partial_\theta \tilde{\mathcal{L}}^{\text{adv}}(\theta_t)\|_2^2$ can be further lower-bounded as follows,
$$\begin{aligned}
\|\partial_\theta \tilde{\mathcal{L}}^{\text{adv}}(\theta_t)\|_2^2 &:= (\partial_{w_{22}^V} \tilde{\mathcal{L}}^{\text{adv}}(\theta_t))^2 + \|\partial_{W_{11}^{KQ}} \tilde{\mathcal{L}}^{\text{adv}}(\theta_t)\|_F^2 \\
&\geq \|\partial_{W_{11}^{KQ}} \tilde{\mathcal{L}}^{\text{adv}}(\theta_t)\|_F^2 \\
&= \|4 \cdot w_{22}^V \cdot (\Gamma(M_{\text{train}})\Lambda + \epsilon^2 \psi(M_{\text{train}})I_d) \cdot D(\theta_t) \cdot \Lambda^{\frac{1}{2}}\|_F^2 \\
&= 16 \cdot (w_{22}^V)^2 \cdot \|(\Gamma(M_{\text{train}})\Lambda + \epsilon^2 \psi(M_{\text{train}})I_d) \cdot D(\theta_t) \cdot \Lambda^{\frac{1}{2}}\|_F^2 \\
&\geq \underbrace{16 \cdot \nu}_{\text{by Lemma A.11}} \cdot \|(\Gamma(M_{\text{train}})\Lambda + \epsilon^2 \psi(M_{\text{train}})I_d) \cdot D(\theta_t) \cdot \Lambda^{\frac{1}{2}}\|_F^2, \qquad (A.29)
\end{aligned}$$
where $\nu > 0$ is defined in Lemma A.11.

Meanwhile, according to the proof of Lemma A.8, we can rewrite and upper-bound $\left( \tilde{\mathcal{L}}^{\text{adv}}(\theta_t) - \min_\theta \tilde{\mathcal{L}}^{\text{adv}}(\theta) \right)$ as follows,

$$\begin{aligned}
&\left( \tilde{\mathcal{L}}^{\text{adv}}(\theta_t) - \min_\theta \tilde{\mathcal{L}}^{\text{adv}}(\theta) \right) \\
&= 2 \cdot \text{Tr}\Big[ (\Gamma_{\text{train}}\Lambda + \epsilon^2 \psi_{\text{train}}I_d) \cdot \left( w_{22}^V W_{11}^{KQ} \Lambda^{\frac{1}{2}} - (\Gamma_{\text{train}}\Lambda + \epsilon^2 \psi_{\text{train}}I_d)^{-1} \Lambda^{\frac{3}{2}} \right) \\
&\qquad\qquad \cdot \left( w_{22}^V W_{11}^{KQ} \Lambda^{\frac{1}{2}} - (\Gamma_{\text{train}}\Lambda + \epsilon^2 \psi_{\text{train}}I_d)^{-1} \Lambda^{\frac{3}{2}} \right)^\top \Big] \\
&= 2 \cdot \text{Tr}[(\Gamma_{\text{train}}\Lambda + \epsilon^2 \psi_{\text{train}}I_d) \cdot D(\theta_t) \cdot D(\theta_t)^\top] \\
&= 2 \cdot \underbrace{\text{Tr}[(\Gamma_{\text{train}}\Lambda + \epsilon^2 \psi_{\text{train}}I_d)^{\frac{1}{2}} \cdot D(\theta_t) \cdot D(\theta_t)^\top \cdot (\Gamma_{\text{train}}\Lambda + \epsilon^2 \psi_{\text{train}}I_d)^{\frac{1}{2}}]}_{\text{Lemma A.3}} \\
&= 2 \cdot \|(\Gamma_{\text{train}}\Lambda + \epsilon^2 \psi_{\text{train}}I_d)^{\frac{1}{2}} \cdot D(\theta_t)\|_F^2 \\
&\leq 2 \cdot \|(\Gamma_{\text{train}}\Lambda + \epsilon^2 \psi_{\text{train}}I_d)^{-\frac{1}{2}}\|_F^2 \cdot \|\Lambda^{-\frac{1}{2}}\|_F^2 \cdot \|(\Gamma_{\text{train}}\Lambda + \epsilon^2 \psi_{\text{train}}I_d) \cdot D(\theta_t) \cdot \Lambda^{\frac{1}{2}}\|_F^2, \quad (A.30)
\end{aligned}$$
where $\Gamma_{\text{train}} := \Gamma(M_{\text{train}})$ and $\psi_{\text{train}} := \psi(M_{\text{train}})$.

Combining Eqs. (A.29) and (A.30), we thus know that
$$\|\partial_\theta \tilde{\mathcal{L}}^{\text{adv}}(\theta_t)\|_2^2 \geq \frac{8\nu}{\|(\Gamma_{\text{train}}\Lambda + \epsilon^2 \psi_{\text{train}}I_d)^{-\frac{1}{2}}\|_F^2 \cdot \|\Lambda^{-\frac{1}{2}}\|_F^2} \cdot \left( \tilde{\mathcal{L}}^{\text{adv}}(\theta_t) - \min_\theta \tilde{\mathcal{L}}^{\text{adv}}(\theta) \right).$$

The proof is completed. $\qquad\qquad\square$

Finally, we prove Theorem 1 based on Lemma A.8 and Lemma A.12.

*Proof of Theorem 1.* When all the conditions hold, when the surrogate AT problem defined in Eq. (9) is solved via continuous gradient flow, by Lemma A.8 we have
$$\partial_t \left( \tilde{\mathcal{L}}^{\text{adv}}(\theta_t) - \min_\theta \tilde{\mathcal{L}}^{\text{adv}}(\theta) \right) = \partial_\theta \tilde{\mathcal{L}}^{\text{adv}}(\theta_t) \cdot \partial_t \theta_t = \partial_\theta \tilde{\mathcal{L}}^{\text{adv}}(\theta_t) \cdot (-\partial_\theta^\top \tilde{\mathcal{L}}^{\text{adv}}(\theta_t)) = -\|\partial_\theta \tilde{\mathcal{L}}^{\text{adv}}(\theta_t)\|_2^2$$
$$\leq -\mu \cdot \left( \tilde{\mathcal{L}}^{\text{adv}}(\theta_t) - \min_\theta \tilde{\mathcal{L}}^{\text{adv}}(\theta) \right),$$
which means
$$\left( \tilde{\mathcal{L}}^{\text{adv}}(\theta_t) - \min_\theta \tilde{\mathcal{L}}^{\text{adv}}(\theta) \right) \leq \left( \tilde{\mathcal{L}}^{\text{adv}}(\theta_0) - \min_\theta \tilde{\mathcal{L}}^{\text{adv}}(\theta) \right) \cdot e^{-\mu t}.$$

As a result, when performing continuous gradient flow optimization for an infinitely long time, since $\mu > 0$, the surrogate AT loss will eventually converge to the global minima, *i.e.*,
$$\left( \tilde{\mathcal{L}}^{\text{adv}}(\theta_*) - \min_\theta \tilde{\mathcal{L}}^{\text{adv}}(\theta) \right) = \lim_{t \to \infty} \left( \tilde{\mathcal{L}}^{\text{adv}}(\theta_t) - \min_\theta \tilde{\mathcal{L}}^{\text{adv}}(\theta) \right) \leq \left( \tilde{\mathcal{L}}^{\text{adv}}(\theta_0) - \min_\theta \tilde{\mathcal{L}}^{\text{adv}}(\theta) \right) \cdot \lim_{t \to \infty} e^{-\mu t} = 0,$$
where $\theta_* := \lim_{t \to \infty} \theta_t$ is the converged model parameter. Meanwhile, from Lemma A.8, we know that $\theta_*$ is a global minimizer if and only if $w_{*,22}^V W_{*,11}^{KQ} = (\Gamma(M_{\text{train}})\Lambda + \epsilon^2 \psi(M_{\text{train}})I_d)^{-1}\Lambda$, which completes the proof. $\qquad\square$

## A.4 Proofs in Section 4.4

This section collects all proofs that omitted from Section 4.4.

*Proof of Theorem 2.* By substituting all $M_{\text{train}}$ with $M_{\text{test}}$ in proofs of Proposition 1 and Lemma A.7, we immediately have that for any model parameter $\theta$ of the LSA model $f_{\text{LSA},\theta}$,

$$\mathcal{R}(\theta, M_{\text{test}}) \leq 2\text{Tr}\Big[(\Gamma(M_{\text{test}})\Lambda + \epsilon^2\psi(M_{\text{test}})I_d) \cdot (w_{22}^V W_{11}^{KQ}\Lambda^{\frac{1}{2}}) \cdot (w_{22}^V W_{11}^{KQ}\Lambda^{\frac{1}{2}})^\top\Big]$$
$$- 4\text{Tr}\Big[(w_{22}^V W_{11}^{KQ}\Lambda^{\frac{1}{2}}) \cdot \Lambda^{\frac{3}{2}}\Big] + 2\text{Tr}(\Lambda).$$

By inserting the converged model parameter $\theta_*(M_{\text{train}})$, which satisfies $(w_{*,22}^V W_{*,11}^{KQ}) = (\Gamma(M_{\text{train}})\Lambda + \epsilon^2\psi(M_{\text{train}})I_d)^{-1}\Lambda$, into the above robust generalization bound, we thus have that

$$\mathcal{R}(\theta_*(M_{\text{train}}), M_{\text{test}})$$
$$\leq 2\text{Tr}\Big[(\Gamma(M_{\text{test}})\Lambda + \epsilon^2\psi(M_{\text{test}})I_d) \cdot ((\Gamma(M_{\text{train}})\Lambda + \epsilon^2\psi(M_{\text{train}})I_d)^{-1}\Lambda \cdot \Lambda^{\frac{1}{2}})$$
$$\cdot ((\Gamma(M_{\text{train}})\Lambda + \epsilon^2\psi(M_{\text{train}})I_d)^{-1}\Lambda \cdot \Lambda^{\frac{1}{2}})^\top\Big]$$
$$- 4\text{Tr}\Big[(\Gamma(M_{\text{train}})\Lambda + \epsilon^2\psi(M_{\text{train}})I_d)^{-1}\Lambda \cdot \Lambda^{\frac{1}{2}} \cdot \Lambda^{\frac{3}{2}}\Big] + 2\text{Tr}(\Lambda)$$
$$\overset{(*)}{\leq} 2\text{Tr}\Big[(\Gamma(M_{\text{test}})\Lambda + \epsilon^2\psi(M_{\text{test}})I_d) \cdot (\Gamma(M_{\text{train}})\Lambda + \epsilon^2\psi(M_{\text{train}})I_d)^{-1}$$
$$\cdot \Lambda^3 \cdot ((\Gamma(M_{\text{train}})\Lambda + \epsilon^2\psi(M_{\text{train}})I_d)^{-1})^\top\Big] + 0 + 2\text{Tr}(\Lambda)$$
$$\overset{(**)}{\leq} 2\text{Tr}\Big[\Lambda^3 \cdot (\Gamma(M_{\text{test}})\Lambda + \epsilon^2\psi(M_{\text{test}})I_d) \cdot (\Gamma(M_{\text{train}})\Lambda + \epsilon^2\psi(M_{\text{train}})I_d)^{-2}\Big] + 2\text{Tr}(\Lambda),$$

where $(*)$ is due to that the matrix $((\Gamma(M_{\text{train}})\Lambda + \epsilon^2\psi(M_{\text{train}})I_d)^{-1}\Lambda^3)$ is positive definite, and $(**)$ is due to that: (1) $(\Gamma(M_{\text{train}})\Lambda + \epsilon^2\psi(M_{\text{train}})I_d)^{-1}$ is symmetric and is commutative with $\Lambda^3$, and (2) Lemma A.3.

The proof is completed. $\qquad\square$

*Proof of Corollary 1.* Let $\lambda_1, \cdots, \lambda_d$ be the $d$ singular values of the matrix $\Lambda$. Then, the robust generalization bound in Theorem 2 can be rewritten as follows,

$$\mathcal{R}(\theta_*(M_{\text{train}}), M_{\text{test}})$$
$$\leq 2\text{Tr}\Big[\Lambda^3 \cdot (\Gamma(M_{\text{test}})\Lambda + \epsilon^2\psi(M_{\text{test}})I_d) \cdot (\Gamma(M_{\text{train}})\Lambda + \epsilon^2\psi(M_{\text{train}})I_d)^{-2}\Big] + 2\text{Tr}(\Lambda),$$
$$\leq \sum_{i=1}^d \lambda_i^3 \cdot \frac{\frac{N+M_{\text{test}}+1}{N+M_{\text{test}}}\lambda_i + \frac{\text{Tr}(\Lambda)}{N+M_{\text{test}}} + \epsilon^2 \cdot \frac{M_{\text{test}}^2 \text{Tr}(\Lambda)}{(N+M_{\text{test}})^2}}{\left(\frac{N+M_{\text{train}}+1}{N+M_{\text{train}}}\lambda_i + \frac{\text{Tr}(\Lambda)}{N+M_{\text{train}}} + \epsilon^2 \cdot \frac{M_{\text{train}}^2 \text{Tr}(\Lambda)}{(N+M_{\text{train}})^2}\right)^2} + 2\text{Tr}(\Lambda)$$
$$\leq \sum_{i=1}^d \lambda_i^3 \cdot \frac{\frac{N+M_{\text{test}}+1}{N+M_{\text{test}}}\lambda_i + \frac{\text{Tr}(\Lambda)}{N+M_{\text{test}}}}{\left(\frac{N+M_{\text{train}}+1}{N+M_{\text{train}}}\lambda_i\right)^2} + \sum_{i=1}^d \lambda_i^3 \cdot \frac{\epsilon^2 \cdot \frac{M_{\text{test}}^2 \text{Tr}(\Lambda)}{(N+M_{\text{test}})^2}}{\left(\epsilon^2 \cdot \frac{M_{\text{train}}^2 \text{Tr}(\Lambda)}{(N+M_{\text{train}})^2}\right)^2} + 2\text{Tr}(\Lambda)$$
$$\leq \sum_{i=1}^d \lambda_i \cdot \left(\frac{N+M_{\text{train}}}{N+M_{\text{train}}+1}\right)^2 \cdot \left(\frac{N+M_{\text{test}}+1}{N+M_{\text{test}}}\lambda_i + \frac{\sum_{k=1}^d \lambda_k}{N}\right)$$
$$+ \sum_{i=1}^d \frac{\lambda_i^3}{\epsilon^2 \cdot \max_{k=1}^d\{\lambda_k\}} \cdot \frac{(N+M_{\text{train}})^4}{N^2} \cdot \frac{M_{\text{test}}^2}{M_{\text{train}}^4} + 2\sum_{i=1}^d \lambda_i$$
$$\leq \mathcal{O}(d) \cdot \mathcal{O}(1) \cdot \left(\mathcal{O}(1) + \frac{\mathcal{O}(d)}{N}\right) + \mathcal{O}(d) \cdot \mathcal{O}\left(\frac{1}{\epsilon^2}\right) \cdot \frac{(N+M_{\text{train}})^4}{N^2} \cdot \frac{M_{\text{test}}^2}{M_{\text{train}}^4} + \mathcal{O}(d)$$
$$\leq \mathcal{O}(d) + \mathcal{O}\left(\frac{d^2}{N}\right) + \mathcal{O}\left(\frac{d}{\epsilon^2}\right) \cdot \frac{(N+M_{\text{train}})^4}{N^2} \cdot \frac{M_{\text{test}}^2}{M_{\text{train}}^4}.$$

Then, by applying Assumption 2, we further have that

$$\mathcal{R}(\theta_*(M_{\text{train}}), M_{\text{test}}) \leq \mathcal{O}(d) + \mathcal{O}\left(\frac{d^2}{N}\right) + \mathcal{O}\left(\frac{d}{\epsilon^2}\right) \cdot \frac{(N + M_{\text{train}})^4}{N^2} \cdot \frac{M_{\text{test}}^2}{M_{\text{train}}^4}$$

$$\leq \mathcal{O}(d) + \mathcal{O}\left(\frac{d^2}{N}\right) + \mathcal{O}\left(\frac{d}{(\sqrt{d})^2}\right) \cdot \frac{(N + O(N))^4}{N^2} \cdot \frac{M_{\text{test}}^2}{M_{\text{train}}^4}$$

$$= \mathcal{O}(d) + \mathcal{O}\left(\frac{d^2}{N}\right) + \mathcal{O}\left(N^2 \cdot \frac{M_{\text{test}}^2}{M_{\text{train}}^4}\right),$$

which completes the proof. $\square$

## B  Additional experimental details

This section collects experimental details omitted from Section 5.

### B.1  Jailbreak attacks

Our experiments leverage both suffix and non-suffix jailbreak attacks. Specifically, four suffix jailbreak attacks are adopted, which are GCG [73], BEAST [44], AmpleGCG [26], and Zhu's AutoDAN [71]. Meanwhile, two non-suffix jailbreak attacks are adopted, which are PAIR [7] and DeepInception [25]. We re-implemented all attacks except AmpleGCG by ourselves to enable fast batching operations during jailbreak, which can thus improve the efficiency of AT. Besides, other than the adversarial suffix length, we will also tune the following hyperparameters of jailbreak attacks:

- **GCG:** According to Algorithm 1 in [73], hyperparameters that we need to tune for GCG include the iteration number $T$, the top-k parameter $k$, and the "batch-size" $B$.

- **BEAST:** According to Algorithm 1 in [44], hyperparameters that we need to tune for BEAST are two beam-search parameters $k_1$ and $k_2$.

- **AmpleGCG:** According to [26], AmpleGCG is an algorithm for training adversarial suffix generators. Our experiments adopt the adversarial suffix generator `AmpleGCG-plus-llama2-sourced-vicuna-7b13b-guanaco-7b13b` [1], which is officially released by [26].

- **Zhu's AutoDAN:** According to Algorithm 1 and Algorithm 2 in [71], hyperparameters that we need to tune for Zhu's AutoDAN are the iteration number $T$ in each step, objective weights $w_1$ and $w_2$, the top-$B$ parameter $B$, and the temperature $\tau$.

- **GCQ:** According to Algorithm 1 in [17], hyperparameters that we need to tune for GCQ include the iteration number $T$, the proxy batch size $b_p$, the query batch size $b_q$, and the buffer size $B$.

- **PAIR:** According to [7], PAIR adopts LLM-based attacker and judger to iteratively synthesize and refine jailbreak prompts. As a result, one needs to set the base models for the attacker and judger and the number of teratively refining for the PAIR attack.

- **DeepInception:** According to [25], DeepInception attack uses manually crafted jailbreak prompts to attack targeted LLMs. We adopt the role play-based prompt from [25] to perform the attack. No other hyperparameter need to be tuned for the DeepInception attack.

### B.2  Model training

**Jailbreak attacks during AT.** We use GCG to search adversarial prompts during AT. The adversarial suffix token length is fixed to one of $\{5, 10, 20, 30, 40, 50\}$ during the overall AT. For other hyparameters described in Appendix B.1, we set $T$ as 150, $k$ as 256, and $B$ as 64.

**Benign answer $y^{(b)}$ for the safety dataset $D^{(h)}$.** We adopt four benign answers for the safety data during AT, which are:

---

[1] https://huggingface.co/osunlp/AmpleGCG-plus-llama2-sourced-vicuna-7b13b-guanaco-7b13b

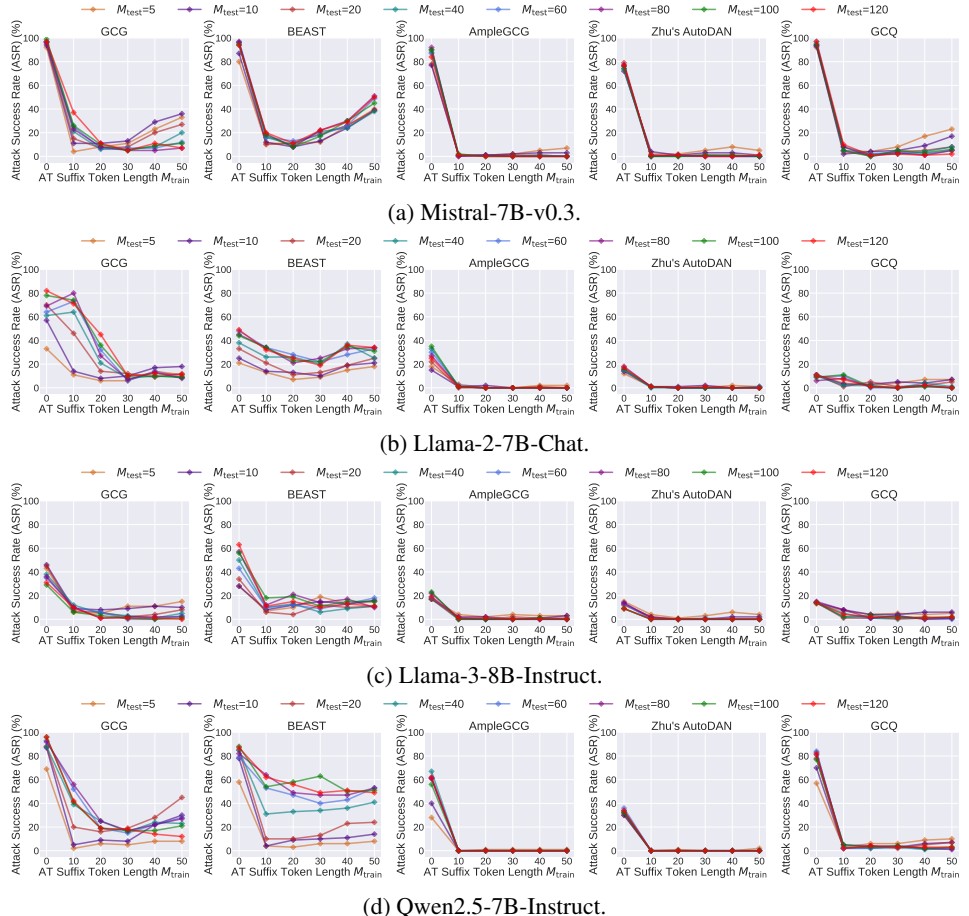

Figure 4: Curves of the ASR versus the adversarial suffix token length during AT (*i.e.*, $M_{\text{train}}$) under jailbreak attacks with different adversarial suffix token lengths (*i.e.*, $M_{\text{test}}$). $M_{\text{train}} = 0$ means that AT is not performed on the evaluated model. A low ASR indicates a strong jailbreak robustness.

- "As a responsible AI assistant, I cannot generate content that promotes illegal activities or violence."
- "Sorry, but I can't do that. Please try other instructions that are safe and responsible."
- "I cannot fulfill your request. I am designed to provide safe and helpful information."
- "This is a topic I am not permitted to discuss since it contain illegal or violent information."

**LoRA setting.** We use the PEFT library [35] to apply LoRA [18] to all query and key projection matrices in attentions. The PEFT hyperparameters are `r=64`, `lora_alpha=32`, and `lora_dropout=0.1`.

**Other hyperparameters for AT.** In every AT experiment, we follow Eq. (3) to perform AT with AdamW for 125 iterations, where the learning rate is set as $5 \times 10^{-5}$ and the factor $\alpha$ is set as $0.2$. Besides, the batch size is set as $64$, in which $8$ samples are jailbreak prompts crafted from data from the safety training set, and the remaining $56$ samples are from the utility training set.

## B.3 Model evaluations

**Robustness evaluation.** We report the Attack Success Rate (ASR) of jailbreak attacks to assess the robustness of models. Specifically, for each instruction from the safety test set, we synthesize the corresponding jailbreak prompt and use it to induce the targeted LLM to generate 10 responses. Then, we use an LLM-based judge from [36], which was fine-tuned from the Llama-2-13B model [1], to

---
[1] https://huggingface.co/cais/HarmBench-Llama-2-13b-cls

determine whether the 10 generated LLM responses are harmful or not. If any of them is determined to be harmful, the jailbreak attack is considered successful.

**Jailbreak attacks for robustness evaluation.** For every suffix attack, the adversarial suffix length is varied within $\{5, 10, 20, 40, 60, 80, 100, 120\}$. Besides, for jailbreak hyperparameters described in Appendix B.1:

- For the GCG attack, we set $T$ as 500, $k$ as 256, and $T$ as 64.
- For the BEAST attack, we set $k_1$ as 64 and $k_2$ as 16.
- For the AmpleGCG attack, we use an official adversarial suffix generator as described in Appendix B.1.
- For the Zhu's AutoDAN attack, we set $T$ as 3, $w_1$ as 10, $w_2$ as 100, $B$ as 256, and $\tau$ as 2.
- For the GCQ attack, we set $T$ as 200 and $b_p$, $b_q$, and $B$ all as 128.
- For the PAIR attack, we set the base model for the attacker as Mistral-8x7B-Instruct-v0.1, the base model for the judger as Llama-3-70B-Instruct, and the number of iteratively refining is fixed to 10.
- For the DeepInception, as explained in Appendix B.1, we use a role-play-based prompt to perform the attack, and there are no other hyperparameters that need to be tuned for this attack.

**Utility evaluation.** We use the AlpacaEval2 framework [10] to report the Length-controlled WinRate (LC-WinRate) of targeted models against a reference model based on their output qualities on the utility test set. An LC-WinRate of $50\%$ means that the output qualities of the two models are equal, while an LC-WinRate of $100\%$ means that the targeted model is consistently better than the reference model. We use Davinci003 as the reference model and use the Llama-3-70B model to judge output quality. The official code of the AlpacaEval2 framework is used to conduct the evaluation. Additionally, the Llama-3-70B judger is run locally via the vLLM model serving framework [23].

### B.4 Additional experimental results

This section collects additional experimental results (*i.e.*, Figure 4) omitted from Section 5.2.

From Figure 4, we find that GCG-based AT is extremely effective in improving model robustness against GCG, AmpleGCG, and Zhu's AutoDAN. For the BEAST attack, GCG-based AT can also suppress the ASR to no more than $50\%$. Further, when the AT adversarial suffix token length is set to 20, AT is already able to reduce the ASR by at least $30\%$ under all settings. It is worth noting that the adversarial suffix length during AT is only up to 50, while that during jailbreaking can vary from 5 to 120. All these results indicate the effectiveness of defending against long-length jailbreaking with short-length AT.

## C   More experiments

This section presents experiments beyond those in Section 5.

### C.1   Comparison with other jailbreak defense baselines

Here, we compare the jailbreak defense performance of short-length LLM AT with that of another jailbreak defense baseline, the Circuit Breakers method [72]. Specifically, we adopt GCG and BEAST attacks to assess the jailbreak robustness of Mistral-7B and Llama-3-8B LLMs protected by short-length LLM AT or the Circuit Breakers defense. For short-length LLM AT, we set the adversarial suffix length $M_{\text{train}}$ during AT to a small value of 20 or 30. For the Circuit Breakers defense, we directly use the trained Mistral-7B [1] and Llama-3-8B [2] models officially released by [72].

The resulting jailbreak ASRs are collected and presented in Table 4, from which we observe that: (1) When the base model is Mistral-7B, short-length LLM AT consistently achieves better jailbreak

---

[1] https://huggingface.co/GraySwanAI/Mistral-7B-Instruct-RR
[2] https://huggingface.co/GraySwanAI/Llama-3-8B-Instruct-RR

Table 4: ASR (%) of suffix jailbreaking against LLMs trained with Circuit Breakers [73] or LLM AT. A low ASR suggests a strong jailbreak robustness of the targeted model.

(a) ASRs of different jailbreak attacks against Mistral-7B.

| Attack | Defense | Adversarial Suffix Token Length $M_{\text{test}}$ in Jailbreaking | | | | | | | |
|---|---|---|---|---|---|---|---|---|---|
| | | 5 | 10 | 20 | 40 | 60 | 80 | 100 | 120 |
| GCG | Circuit Breakers [72] | 21.0 | 20.0 | 21.0 | 23.0 | 23.0 | 28.0 | 28.0 | 23.0 |
| | LLM AT ($M_{\text{train}} = 20$) | **8.0** | **11.0** | **7.0** | **6.0** | **7.0** | 8.0 | 10.0 | 11.0 |
| | LLM AT ($M_{\text{train}} = 30$) | 11.0 | 13.0 | 8.0 | **6.0** | **7.0** | **5.0** | **5.0** | **5.0** |
| BEAST | Circuit Breakers [72] | 19.0 | 21.0 | 20.0 | 24.0 | 25.0 | 25.0 | 25.0 | 27.0 |
| | LLM AT ($M_{\text{train}} = 20$) | **11.0** | **8.0** | **11.0** | **10.0** | **13.0** | **8.0** | **8.0** | **11.0** |
| | LLM AT ($M_{\text{train}} = 30$) | 12.0 | 13.0 | 19.0 | 21.0 | 18.0 | 22.0 | 17.0 | 22.0 |

(b) ASRs of different jailbreak attacks against Llama-3-8B.

| Model | Defense | Adversarial Suffix Token Length $M_{\text{test}}$ in Jailbreaking | | | | | | | |
|---|---|---|---|---|---|---|---|---|---|
| | | 5 | 10 | 20 | 40 | 60 | 80 | 100 | 120 |
| GCG | Circuit Breakers [72] | **3.0** | **5.0** | 3.0 | 4.0 | 3.0 | 5.0 | 5.0 | 7.0 |
| | LLM AT ($M_{\text{train}} = 20$) | 5.0 | 8.0 | 6.0 | 5.0 | 6.0 | **1.0** | 3.0 | 1.0 |
| | LLM AT ($M_{\text{train}} = 30$) | 11.0 | 9.0 | **2.0** | 3.0 | **0.0** | 2.0 | **1.0** | **1.0** |
| BEAST | Circuit Breakers [72] | 12.0 | **9.0** | 11.0 | 12.0 | 16.0 | 15.0 | 17.0 | 15.0 |
| | LLM AT ($M_{\text{train}} = 20$) | **10.0** | 12.0 | **4.0** | 13.0 | 12.0 | 21.0 | 19.0 | 15.0 |
| | LLM AT ($M_{\text{train}} = 30$) | 19.0 | 15.0 | 12.0 | **6.0** | **9.0** | **14.0** | **11.0** | **10.0** |

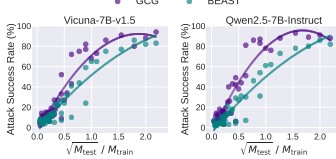

(a) Scatter plots of ASR to the ratio $\sqrt{M_{\text{test}}}/M_{\text{train}}$.

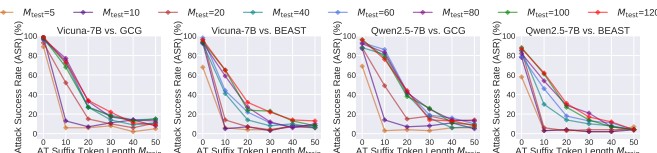

(b) ASR versus $M_{\text{train}}$ under different $M_{\text{test}}$. $M_{\text{train}} = 0$ means that AT is not performed on the evaluated model.

Figure 5: ASR of models trained from the BEAST-based LLM AT. A low ASR indicates a strong jailbreak robustness of the model.

robustness than Circuit Breakers under different jailbreak attack adversarial suffix lengths. (2) When the base model is Llama-3-8B, the two defense methods achieve similar performance.

### C.2   LLM AT with the BEAST attack

In our main experiments in Section 5, we solely use the GCG attack to synthesize jailbreak prompts for LLM AT. In this section, we investigate whether our theoretical results still empirically hold for AT with jailbreak attacks other than GCG. Specifically, we now perform LLM AT with the BEAST attack on Vicuna-7B-v1.5 and Qwen2.5-7B-Instruct models. For the hyperparameters of BEAST described in Appendix B.1, we vary the adversarial suffix token length within $\{5, 10, 20, 30, 40, 50\}$, and set $k_1$ to $64$ and $k_2$ to $16$. All other settings of LLM AT follow those described in Section B.2.

Experimental results are presented in Figure 5 and Table 5. From Figure 5a and Table 5, we observe a statistically significant positive correlation between the suffix jailbreak robustness and the ratio $\sqrt{M_{\text{test}}}/M_{\text{train}}$ in every experiment, which indicates that our ICL-AT theory still holds for BEAST-based LLM AT. Besides, from Figure 5b, one can find that AT with a short adversarial suffix length $M_{\text{train}}$ of 30 can already reduce the ASR from nearly $100\%$ to around $20\%$ in every evaluation case, which demonstrates the effectiveness of short-length BEAST-based LLM AT in defending against jailbreak attacks.

### C.3   LLM AT on larger models

We also perform short-length LLM AT on Vicuna-13B-v1.5, which is a model larger than those 7B/8B LLMs adopted in our main experiments in Section 5. All hyperparameters for LLM AT follow those described in Appendix B.2. Results are presented in Table 6, which shows that AT with an

Table 5: PCCs and $p$-values calculated between ASR and ratio $\sqrt{M_{\text{test}}}/M_{\text{train}}$ on LLMs adversarially trained with the BEAST attack. $p < 5.00 \times 10^{-2}$ means that the correlation between ASR and the ratio is considered statistically significant.

| Model | GCG Attack | | BEAST Attack | |
|---|---|---|---|---|
| | PCC($\uparrow$) | $p$-value($\downarrow$) | PCC($\uparrow$) | $p$-value($\downarrow$) |
| Vicuna-7B | 0.91 | $\mathbf{5.3 \times 10^{-19}}$ | 0.94 | $\mathbf{6.7 \times 10^{-24}}$ |
| Qwen2.5-7B | 0.88 | $\mathbf{2.2 \times 10^{-16}}$ | 0.95 | $\mathbf{5.0 \times 10^{-25}}$ |

Table 6: ASR (%) of the GCG attack against Vicuna-13B-v1.5 trained with LLM AT. A low ASR suggests a strong jailbreak robustness of the targeted model.

(a) ASRs of different jailbreak attacks against Mistral-7B.

| Attack | Defense | Adversarial Suffix Token Length $M_{\text{test}}$ in Jailbreaking | | | | | | | |
|---|---|---|---|---|---|---|---|---|---|
| | | 5 | 10 | 20 | 40 | 60 | 80 | 100 | 120 |
| | None | 92.0 | 94.0 | 99.0 | 96.0 | 98.0 | 96.0 | 99.0 | 98.0 |
| GCG | LLM AT ($M_{\text{train}} = 5$) | **11.0** | 19.0 | 30.0 | 53.0 | 55.0 | 67.0 | 70.0 | 68.0 |
| | LLM AT ($M_{\text{train}} = 20$) | 12.0 | **9.0** | 11.0 | **6.0** | **6.0** | **6.0** | **8.0** | **7.0** |

adversarial suffix token length as short as 20 can already reduce the ASR of the GCG attack from nearly 99% to around 10% in the worst case. This suggests the generalization of our theoretical findings beyond 7B/8B models.

