# OpenReview forum: "Short-length Adversarial Training Helps LLMs Defend Long-length Jailbreak Attacks: Theoretical and Empirical Evidence"
_NeurIPS.cc/2025/Conference — NeurIPS 2025 poster_

### Official Review · Reviewer_gkox · 2025-07-01

**Clarity:** 3
**Significance:** 2
**Originality:** 3
**Rating:** 4
**Confidence:** 3

**Summary:**

This paper explores the challenges of using adversarial training (AT) to defend against jailbreak attacks on large language models (LLMs).
The core contribution is to demonstrate the effectiveness of short-length adversarial training for defending against long-length adversarial suffix attacks from both theoretical and empirical perspectives.
Combining practical results with theory, this paper conducts empirical verification on five LLMs covering four attack types, and the results show that short suffixes used in training (e.g., 20 tokens) can defend against significantly longer attack suffixes (up to 120 tokens).

**Questions:**

1. Scope limitation: This paper focuses on suffix-based jailbreak attacks, but a large number of LLM jailbreak attacks use non-suffix patterns (such as deepinception), which reduces the practical value of this paper.
It would be even better if the author could discuss and clarify the limitations on the scope of this paper.
2. Transferability to large models: The largest model in the experiments is Llama-3-8b.
Given that the parameter scale of commonly used LLMs typically range from 30B to 70B, this paper could discuss whether its findings and conclusions have the potential to be extended to larger models.
3. Impact of suffix generation strategy:
It remains unclear the impact of suffix generation methods (i.e., GCG attack, token level generation) used in AT.
It would be great if the author could conduct additional experiments, such as comparing the use of token level and character level methods to generate suffix in AT.
Alternatively, the author can discuss the impact of different methods on the experimental results and conclusions.
This could help future research on efficient and effective construction of AT.
4. In Figure 1, AutoDAN's ASR seems to be always low.
Is this related to the judgment method to determine the success of the attack (for example, judging by whether the output result has specific keywords or starts with a specific string)?
Improper judgment methods may affect experimental results and observations。
The author should provide some explanations and clarifications to improve the persuasiveness of the experiment.

**Ethical Concerns:**

["NO or VERY MINOR ethics concerns only"]

**Final Justification:**

The authors have answered all questions. About the practiability, although the authors have conducted additional experiments, I still have concerns about this aspect. I will maintain the score.

**Limitations:**

Yes

**Quality:**

3

**Strengths And Weaknesses:**

Strengths:
1. Theoretical evidence and empirical validation: The derivation of a generalization bound based on suffix length is both mathematically rigorous and practically meaningful. The experiments on five open-source LLMs and four jailbreak methods further offer strong support for the main claim.
2. Efficiency insight: Demonstrates that short adversarial suffixes during training can yield robustness, which could reduce training cost significantly.
3. Clear analysis: Uses ASR VS. √(Mtest/Mtrain) ratio across multiple models/attacks to show the correlation between the jailbreak robustness and the ratio.

Weaknesses:
1. Limited practical value: Focuses only on suffix-style jailbreaks, while many attack samples are in other prevalent formats (e.g., roleplay, templates).
2. Insufficient exploration of model scale: Unclear if conclusions hold for larger LLMs (e.g., 30B) beyond the tested 7–8B range.

---

> ### Author Rebuttal · Authors · 2025-07-30
>
> Thank you for your thorough comments and kind support!
>
> **Q1: (From Weakness#1 & Question#1) Limited practical value: Focuses only on suffix-style jailbreaks, while many attack samples are in other prevalent formats (e.g., roleplay, templates).
> Scope limitation: This paper focuses on suffix-based jailbreak attacks, but a large number of LLM jailbreak attacks use non-suffix patterns (such as deepinception), which reduces the practical value of this paper. It would be even better if the author could discuss and clarify the limitations on the scope of this paper.**
>
> **A1:** Thanks. We agree that our current theoretical and empirical analyses are limited to suffix-based jailbreak attacks. To further investigate how our current results can be generalized to non-suffix-based jailbreaking settings, we have adopted an additional non-suffix-based jailbreak attack in our experiments, i.e., the PAIR attack [r1]. Specifically, the PAIR attack is an agent-based jailbreak attack that generates jailbreak prompts directly using LLM-based agents. It has been widely adopted as a jailbreak baseline in many existing works [r4, r5, r6, r9]. The preliminary results of the PAIR attack on adversarially trained Vicuna-7B models are presented in Table r1 below:
>
> | $M_{\text{train}}$ | 0 (No AT) | 5 | 10 | 20 | 30 | 40 | 50 |
> | ---- | ---- | ---- | ---- | ---- | ---- | ---- | ---- |
> | Vicuna-7B + AT v.s. **PAIR Attack [r1]** | 84 | 53 | 48 | 42 | 50 | 44 | 55 |
>
> *Table r1: ASR (%) of PAIR attack [r1] against adversarially trained Vicuna-7B.*
>
> From Table r1, we find that while GCG-based LLM AT can reduce the ASR of the PAIR attack from more than 80% to around 50%, further increasing $M_{\text{train}}$ does not help much to improve LLM robustness against PAIR. This suggests that the mechanisms behind suffix-based and non-suffix-based jailbreak attacks might share different properties.
>
> Nevertheless, we believe a potential solution to improve our work for the non-suffix-based attack setting is to extend our current ICL-AT theoretical framework to the setting of "continuous-AT" [r5]. Specifically, continuous-AT [r5] is a more recent LLM AT method that has been shown to improve LLM robustness against both suffix-based and non-suffix-based jailbreak attacks. Since continuous-AT is implemented by adding adversarial perturbations to the continuous embeddings of input tokens, our ICL-AT theoretical framework can be more easily generalized to this setting. We will leave this generalization for future studies.
>
> **Q2: (From Weakness#2 & Question#2) Insufficient exploration of model scale: Unclear if conclusions hold for larger LLMs (e.g., 30B) beyond the tested 7–8B range.
> Transferability to large models: The largest model in the experiments is Llama-3-8b. Given that the parameter scale of commonly used LLMs typically range from 30B to 70B, this paper could discuss whether its findings and conclusions have the potential to be extended to larger models.**
>
> **A2:** Thanks. To address your concern, we have added an additional 13B model for experiments, which is Vicuna-13B-v1.5. We will also add experiments on 30B+ models (e.g., Qwen2.5-32B), but not for now due to the limited rebuttal period.
>
> Preliminary results of Vicuna-13B-v1.5 with adversarial training are presented in the following Table r3.
>
> | $M_{\text{test}}$ (GCG Attack) | 5 | 10 | 20 | 40 | 60 | 80 | 100 | 120 |
> | ---- | ---- | ---- | ---- | ---- | ---- | ---- | ---- | ---- |
> | Vicuna-13B (Original) | 92 | 94 | 99 | 96 | 98 | 96 | 99 | 98 |
> | Vicuna-13B + LLM AT ($M_{\text{train}}=5$) | 11 | 19 | 30 | 53 | 55 | 67 | 70 | 68 |
> | Vicuna-13B + LLM AT ($M_{\text{train}}=20$) | 12 | 9 | 11 | 6 | 6 | 6 | 8 | 7 |
>
> *Table r3: ASR (%) of the GCG attack against adversarially trained Vicuna-13B.*
>
> Table r3 shows that AT with a short adversarial suffix length $M_{\text{train}} = 20$ can already reduce the ASR of the GCG attack from nearly 99% to around 10% in the worst case. This justifies the generalization of our theoretical findings beyond 7B/8B models.
>
> A complete analysis for larger models (including both Vicuna-13B and another 30B+ model) will be added in the revision.
>
> **Q3: Impact of suffix generation strategy: It remains unclear the impact of suffix generation methods (i.e., GCG attack, token level generation) used in AT. It would be great if the author could conduct additional experiments, such as comparing the use of token level and character level methods to generate suffix in AT. Alternatively, the author can discuss the impact of different methods on the experimental results and conclusions. This could help future research on efficient and effective construction of AT.**
>
> **A3:** Thanks. Following your suggestion, we have conducted experiments on LLM adversarial training with a different suffix generation method, the BEAST attack [r9]. Preliminary results of BEAST-based LLM AT on Vicuna-7B are presented in the following Table r4.
>
> | $M_{\text{test}}$ (GCG/BEAST Attack) | 5 | 10 | 20 | 40 | 60 | 80 | 100 | 120 |
> | ---- | ---- | ---- | ---- | ---- | ---- | ---- | ---- | ---- |
> | Vicuna-7B (Original) v.s. GCG Attack | 89 | 99 | 93 | 98 | 96 | 97 | 98 | 99 |
> | Vicuna-7B + BEAST-based LLM AT ($M_{\text{train}}=20$) v.s. GCG Attack | 6 | 7 | 15 | 27 | 27 | 33 | 27 | 34 |
> | Vicuna-7B (Original) v.s. BEAST Attack | 68 | 93 | 96 | 92 | 98 | 93 | 94 | 96 |
> | Vicuna-7B + BEAST-based LLM AT ($M_{\text{train}}=20$) v.s. BEAST Attack | 4 | 7 | 7 | 14 | 22 | 27 | 24 | 32 |
>
> *Table r4: ASR (%) of different jailbreak attacks against BEAST-based adversarially trained Vicuna-7B.*
>
> From Table r4, we find that LLM AT via the BEAST attack with a short adversarial suffix length of $M_{\text{train}}=20$ can already reduce the ASR of GCG or BEAST attacks from nearly 100% to 30+% in the worst cases, which demonstrates the effectiveness of BEAST-based short-length AT. Furthermore, when compared with the results of GCG-based AT in Figure 2 of our original paper, Table r4 also seems to suggest that BEAST-based AT can achieve a better robustness balance in defending against both GCG and BEAST attacks simultaneously. We will leave the study of how different adversarial suffix generation methods in LLM AT affect LLM robustness for future research.
>
> Complete results of the BEAST-based AT will be added in the revision.
>
> **Q4: In Figure 1, AutoDAN's ASR seems to be always low. Is this related to the judgment method to determine the success of the attack (for example, judging by whether the output result has specific keywords or starts with a specific string)? Improper judgment methods may affect experimental results and observations. The author should provide some explanations and clarifications to improve the persuasiveness of the experiment.**
>
> **A4:** Thanks. Firstly, we would like to clarify that the ASR calculation in our paper is implemented using an open-source LLM-based judger [r2] (originally proposed by [r3]) to determine whether a given piece of LLM-generated content contains harmful information. It is **NOT** implemented by judging whether output results contain specific keywords or start with specific strings. Please see Appendix B.3 in our paper for more details on the ASR calculation.
>
> Secondly, for the LLM-based judger used in our ASR calculation, we note that [r3] has shown that judgments made by this judger align well with human judges. This judger has also been widely adopted for jailbreak robustness evaluation in many existing works, such as [r4], [r5], and [r6]. Thus, we think the ASR calculation method we used is reliable enough and is unlikely to be the cause of the low ASR of Zhu's AutoDAN in Figure 1.
>
> Thirdly, for our reproduced version of Zhu's AutoDAN, since the authors of Zhu's AutoDAN did not release their official experimental code publicly (the GitHub link [r7] in their original paper [r8] is unfortunately empty), we had to reproduce this attack solely based on the pseudocode provided in their original paper [r8]. We tried various combinations of hyperparameters (see Appendix B.1 for all tunable hyperparameters in our reproduced version of Zhu's AutoDAN), but failed to find a strong one that could achieve effective attack performance against adversarially trained LLMs.
>
> Finally, we deduce that the low ASR of Zhu's AutoDAN in Figure 1 is mainly due to the fact that this attack is very similar to the GCG attack, as both utilize gradients with respect to one-hot token-space embeddings to find adversarial suffix tokens. Zhu's AutoDAN further needs to preserve the semantics of synthesized jailbreak suffixes, which may potentially reduce its attack performance compared with the GCG attack. Since our LLM AT is performed with GCG, the trained LLMs may be very robust to attacks similar to GCG, such as Zhu's AutoDAN.
>
> **References:**
>
> [r1] Chao et al. "Jailbreaking Black Box Large Language Models in Twenty Queries". arXiv 2023.
>
> [r2] https://huggingface.co/cais/HarmBench-Llama-2-13b-cls
>
> [r3] Mazeika et al. "HarmBench: A Standardized Evaluation Framework for Automated Red Teaming and Robust Refusal". arXiv 2024.
>
> [r4] Paulus et al. "AdvPrompter: Fast Adaptive Adversarial Prompting for LLMs". ICML 2025.
>
> [r5] Xhonneux et al. "Efficient Adversarial Training in LLMs with Continuous Attacks". NeurIPS 2024.
>
> [r6] Yu et al. "Robust LLM safeguarding via refusal feature adversarial training". ICLR 2025.
>
> [r7] http://autodan-jailbreak.github.io/
>
> [r8] Zhu et al. "AutoDAN: Interpretable Gradient-Based Adversarial Attacks on Large Language Models". COLM 2024. https://arxiv.org/abs/2310.15140
>
> [r9] Sadasivan et al. "Fast Adversarial Attacks on Language Models In One GPU Minute". ICML 2024.

---

> > ### Comment · Reviewer_gkox · 2025-08-03
> >
> > The authors have answered all questions: (1) acknowledged the scope and limitations of this paper and proposed feasible future research directions; (2) supplemented their experiments with additional results using the 13B model and BEAST-based AT to further support the scalability and transferabililty of their approach, and (3) clarified the AutoDAN experimental results.
> >
> > While the authors have addressed part of my concerns, the paper still primarily focuses on suffix jailbreak attacks, and there is limited evidence that such attacks are prevalent in real-world scenarios. Therefore, despite the additional clarifications and experiments, I believe the paper's practical impact remains somewhat limited.
> >
> > Therefore, I will maintain the score of 4: Borderline Acceptance.

---

> ### Author Response · Authors · 2025-08-03
> **Thank you for your feedback & Further response**
>
> Dear Reviewer gkox,
>
> Thank you for your kind feedback!
>
> We are glad that our rebuttal addressed part of your concerns. We will include all the additional experiments from the rebuttal in our revision.
>
> **Q5: While the authors have addressed part of my concerns, the paper still primarily focuses on suffix jailbreak attacks, and there is limited evidence that such attacks are prevalent in real-world scenarios. Therefore, despite the additional clarifications and experiments, I believe the paper's practical impact remains somewhat limited.**
>
> **A5:** Regarding your concern that our experiments on LLM-AT focus more on suffix-based jailbreak attacks than on non-suffix-based ones, we would like to note that we have made the following efforts to address your concern: (1) (in **A1**) We added experiments on a non-suffix-based jailbreak attack, PAIR [r1], and (2) (in **A1**) we clarified how to generalize our current ICL-AT theory to defend against non-suffix-based jailbreak attacks (via theoretically analyzing continuous-AT).
>
> To better address your concern, we have added another non-suffix-based jailbreak attack, DeepInception [r10], which you also mentioned in your original review, as an attack baseline in our experimental evaluation. Specifically, DeepInception [r10] is a jailbreak attack achieved by inserting harmful instructions into carefully designed prompt templates to bypass the safeguards of targeted LLMs. The preliminary results are shown in Table r5:
>
> | $M_{\text{train}}$ | 0 (No AT) | 5 | 10 | 20 | 30 | 40 | 50 |
> | ---- | ---- | ---- | ---- | ---- | ---- | ---- | ---- |
> | Vicuna-7B + AT v.s. **DeepInception Attack [r10]**  | 76 | 39 | 15 | 0 | 0 | 0 | 0 |
> | Llama3-8B + AT v.s. **DeepInception Attack [r10]**  | 44 | 0 | 0 | 0 | 0 | 0 | 0 |
> | Qwen2.5-7B + AT v.s. **DeepInception Attack [r10]** | 89 | 0 | 0 | 0 | 0 | 0 | 0 |
>
> *Table r5: ASR (%) of DeepInception attack [r10] against adversarially trained LLMs.*
>
> From Table r5, we find that LLM AT with a short adversarial suffix length $M\_{\text{train}} = 30$ can already suppress the ASR of the DeepInception attack to 0%, which indicates that short-length AT is extremely effective against the DeepInception attack. Additionally, when compared with the results of the PAIR attack in Table r1, we find that while both PAIR and DeepInception attacks are non-suffix-based jailbreak attacks, the PAIR attack is more robust to LLM-AT than the DeepInception attack.
>
> Finally, we would like to note that our paper is one of the very first works that try to theoretically understand the effect of adversarial training in defending against LLM jailbreak attacks. Our main theoretical contribution is that we have proposed the first ICL-based theoretical framework for analyzing LLM AT. While the theoretical analysis currently mainly focuses on suffix-based attacks, we believe this does not diminish the merit of our work.
>
> **Reference:**
>
> [r10] Li et al. "DeepInception: Hypnotize Large Language Model to Be Jailbreaker". arXiv 2023.

---

### Official Review · Reviewer_6htW · 2025-07-02

**Clarity:** 3
**Significance:** 3
**Originality:** 3
**Rating:** 5
**Confidence:** 4

**Summary:**

This paper presents both theoretical and empirical evidence to show the quadratic relation between the length of adversarial training and robustness for LLMs against jailbreaking. Theoretically, the authors demonstrate this insight by simplifying the problem as an instance of adversarial in-context learning. Empirically, this insight is validated by 4 jailbreaking attacks.

**Questions:**

1. How can this theory generalize to non-suffix jailbreaking attacks?
2. What are the connection between this theory and other theoretical insights from ICL-based jailbreaking attacks?

**Ethical Concerns:**

["NO or VERY MINOR ethics concerns only"]

**Final Justification:**

I think this is an interesting paper and keep my score for acceptance. Please incorporate the discussed points in your revision.

**Limitations:**

Yes.

**Quality:**

3

**Strengths And Weaknesses:**

# Strengths
1. Theoretical understanding of adversarial suffix-based jailbreaking attacks remains an open and important problem.
2. It is interesting to study this problem through the lens of in-context learning.
3. This paper also presents comprehensive empirical results to support the theoretical insights.

# Weakness
1. While the theory mainly focuses on suffix-based attacks, its extension to non-suffix attacks is not well-discussed.
2. The connection between the theory presented in this paper and other theoretical insights regarding ICL-based jailbreak attacks [1,2] is also not addressed. Additionally, in line 31, [1], which first presented this discovery before [2], was not cited.

[1] Jailbreak and Guard Aligned Language Models with Only Few In-Context Demonstrations https://arxiv.org/pdf/2310.06387
[2] Many-shot jailbreaking. NeurIPS 2024

---

> ### Author Rebuttal · Authors · 2025-07-30
>
> Thank you for your thorough comments and kind support!
>
> **Q1: (From Weakness#1 & Question#1) While the theory mainly focuses on suffix-based attacks, its extension to non-suffix attacks is not well-discussed.
> How can this theory generalize to non-suffix jailbreaking attacks?**
>
> **A1:** Thank you for this good question. We believe a potential solution is to extend our current ICL-AT theoretical framework to the setting of "continuous-AT" [r3].
>
> Specifically, continuous-AT [r3] is a more recent LLM AT method that has been shown to improve LLM robustness against both suffix-based and non-suffix-based jailbreak attacks. Since continuous-AT is implemented by adding adversarial perturbations to **the continuous embeddings of input tokens**, our ICL-AT theoretical framework is naturally easier to generalize to this setting.
>
> Generalizing our current ICL-AT theoretical framework to analyze continuous-AT would require two steps: (1) introduce an additional trainable matrix into the LSA model $f_{\text{LSA},\theta}$ to map the ICL input $E_\tau$ to a new embedding space, and (2) perform the training-time adversarial perturbation within the new embedding space (rather than in the original space of $E_\tau$). We will leave the generalization for future studies.
>
> **Q2: (From Weakness#2 & Question#2) The connection between the theory presented in this paper and other theoretical insights regarding ICL-based jailbreak attacks [1,2] is also not addressed. Additionally, in line 31, [1], which first presented this discovery before [2], was not cited.
> What are the connection between this theory and other theoretical insights from ICL-based jailbreaking attacks?**
>
> **A2:** Firstly, regarding [r2], we respectfully note that this work is purely experimental and does not present any theoretical results/insights.
>
> Secondly, regarding [r1], the main connection between their theoretical insight and our ICL-AT theory is that **we both noticed that the number of adversarial in-context samples plays a critical role in ICL-based adversarial attacks**. Specifically, [r1] theoretically characterizes the minimum number of in-context adversarial samples required to increase the safety loss of ICL to some extent. On the other hand, our ICL-AT theory illustrates a “square-root connection” between the number of in-context adversarial (suffix) samples during training (i.e., $M_{\text{train}}$) and during testing (i.e., $M_{\text{test}}$).
>
> While [r1] and our work share similarities, the main difference is that [r1] focuses on studying the adversarial robustness of fixed ICL models, whereas our work analyzes how adversarial training affects the robustness of ICL models.
>
> We will cite and discuss [r1] in the revision.
>
> **References:**
>
> [r1] Wei et al. "Jailbreak and Guard Aligned Language Models with Only Few In-Context Demonstrations". arXiv 2023.
>
> [r2] Anil et al. "Many-shot Jailbreaking". NeurIPS 2024.
>
> [r3] Xhonneux et al. "Efficient Adversarial Training in LLMs with Continuous Attacks". NeurIPS 2024.

---

> > ### Comment · Reviewer_6htW · 2025-08-01
> >
> > I thank the authors for the clarifications. I think this is an interesting paper and keep my score for acceptance. Please incorporate the discussed connection with [r1] in your revision.

---

> > > ### Author Response · Authors · 2025-08-01
> > > **Thanks for your support and suggestion!**
> > >
> > > Thank you very much for recognizing our contributions and kind support! We will include the discussion of [r1] in the revision as you suggested.

---

### Official Review · Reviewer_q37y · 2025-07-02

**Clarity:** 1
**Significance:** 3
**Originality:** 3
**Rating:** 4
**Confidence:** 4

**Summary:**

This paper investigates the effectiveness of short-length adversarial training in defending LLMs against long-length jailbreak attacks.

The authors provide very detailed theoretical analysis and some empirical experimental results to support their findings that there exists a positive correlation between the attack success rate and the ratio of the square root of the adversarial suffix length during jailbreaking to the length during adversarial training. Additionally, they establish a robust generalization bound under a linear transformer framework.

**Questions:**

1. How does the effectiveness of short-length AT compare when defending against other types of jailbreak attacks beyond suffix attacks? Can the authors provide insights or preliminary results on the robustness against more complex or multi-token attacks?

2. Could the authors provide more details or quantitative analysis regarding the computational efficiency of short-length AT compared to longer-length AT during training?

**Ethical Concerns:**

["NO or VERY MINOR ethics concerns only"]

**Final Justification:**

Raised my score. Although I am not fully convinced, I recognize this paper meets the bar of most ML conferences on this topic...

**Limitations:**

As noted above, I am not optimistic about the generalizability of the proposed approach. The paper does not present experimental evidence on newer or larger-scale LLMs, different architectures, or diverse types of jailbreak attacks. Further validation in these directions would be necessary to demonstrate the broader applicability of the method.

**Quality:**

2

**Strengths And Weaknesses:**

# Strengths

1. The paper offers a rigorous theoretical treatment of adversarial in-context learning for linear transformers. The derived robust generalization bound sheds light on how adversarial prompt length affects model robustness.

2. The empirical results, though limited in scale, show that short-length AT can help defend against long-length jailbreak attacks on several open-source LLMs.

# Weaknesses

1. Despite the strong theoretical contributions, the writing can be hard to follow in several parts, making it difficult for readers to understand the proofs. For example, some notational choices are confusing, such as the use of both M and m in Equations (2) and (3).

2. The empirical evaluation is not comprehensive enough to fully support the paper’s claims. The authors only test on small-scale LLMs (7B) and focus on similar types of token-level attacks. It would strengthen the paper to include experiments on more diverse LLM families and a broader range of attack types.

---

> ### Author Rebuttal · Authors · 2025-07-30
>
> Thank you for your thorough comments.
>
> **Q1: Despite the strong theoretical contributions, the writing can be hard to follow in several parts, making it difficult for readers to understand the proofs. For example, some notational choices are confusing, such as the use of both M and m in Equations (2) and (3).**
>
> **A1:** Thanks. The uppercase "M" in Eqs. (2) and (3) should be lowercase "m". These are typos, and we apologize for any confusion.
>
> We will improve our writing in the revision to help readers better understand our proofs. Specifically, we will: (1) recheck all notations in Section 3 and fix potential typos; (2) add a notation table for Section 4 to clarify all necessary notations in this section; and (3) add a "Proof Skeleton" section for Section 4 to better summarize the main idea and contribution of our theoretical analysis.
>
> **Q2: The empirical evaluation is not comprehensive enough to fully support the paper’s claims. The authors only test on small-scale LLMs (7B) and focus on similar types of token-level attacks. It would strengthen the paper to include experiments on more diverse LLM families and a broader range of attack types.**
>
> **A2:** Thanks. To address the concern about **testing only 7B/8B LLMs**, we have added an additional 13B model for experiments, which is Vicuna-13B-v1.5. Preliminary results are presented in the following Table r1.
>
> | $M_{\text{test}}$ (GCG Attack) | 5 | 10 | 20 | 40 | 60 | 80 | 100 | 120 |
> | ---- | ---- | ---- | ---- | ---- | ---- | ---- | ---- | ---- |
> | Vicuna-13B (Original) | 92 | 94 | 99 | 96 | 98 | 96 | 99 | 98 |
> | Vicuna-13B + LLM AT ($M_{\text{train}}=5$) | 11 | 19 | 30 | 53 | 55 | 67 | 70 | 68 |
> | Vicuna-13B + LLM AT ($M_{\text{train}}=20$) | 12 | 9 | 11 | 6 | 6 | 6 | 8 | 7 |
>
> *Table r1: ASR (%) of the GCG attack against adversarially trained Vicuna-13B.*
>
> Table r1 shows that AT with a short adversarial suffix length $M_{\text{train}} = 20$ can already reduce the ASR of the GCG attack from nearly 99% to around 10% in the worst case. This justifies the generalization of our theoretical findings beyond 7B/8B models. Besides, we would also like to note that our original paper has already adopted models from 5 different LLM families (i.e., Vicuna-7B, Mistral-7B, Llama-2-7B, Llama-3-7B, and Qwen2.5-7B), which we believe is diverse enough to empirically verify our theoretical findings.
>
> To address the concern about **testing with only token-level attacks**, we have additionally adopted a non-token-level jailbreak attack, i.e., PAIR attack [r1]. Specifically, PAIR attack is an agent-based jailbreak attack that generates jailbreak prompts directly using LLM-based agents. It has been widely adopted as a jailbreak baseline in many existing works [r2, r3, r4, r5]. Preliminary results of the PAIR attack on adversarially trained Vicuna-7B models are presented in Table r2 below:
>
> | $M_{\text{train}}$ | 0 (No AT) | 5 | 10 | 20 | 30 | 40 | 50 |
> | ---- | ---- | ---- | ---- | ---- | ---- | ---- | ---- |
> | Vicuna-7B + AT v.s. **PAIR Attack [r1]** | 84 | 53 | 48 | 42 | 50 | 44 | 55 |
>
> *Table r2: ASR (%) of PAIR attack [r1] against adversarially trained Vicuna-7B.*
>
> From Table r2, we find that while GCG-based LLM AT can reduce the ASR of the PAIR attack from more than 80% to around 50%, further increasing $M_{\text{train}}$ does not help much to improve LLM robustness against PAIR. We will leave the theoretical analysis of defending attacks other than suffix-based token-level jailbreaking for future studies.
>
> The complete analysis and discussion of the Vicuna-13B-v1.5 model and the PAIR jailbreak attack will be included in the revision.
>
>
>
> **Q3: How does the effectiveness of short-length AT compare when defending against other types of jailbreak attacks beyond suffix attacks? Can the authors provide insights or preliminary results on the robustness against more complex or multi-token attacks?**
>
> **A3:** Thanks. We have added a new baseline attack, named PAIR [r1], to our empirical analysis. As explained in **A2**, the PAIR attack [r1] is an agent-based jailbreak attack but not a suffix-based attack. Preliminary results are presented in Table r2 in **A2**, which show that while short-length AT can improve LLM robustness against PAIR, increasing the adversarial suffix length during AT does not help much to further improve LLM robustness. These results indicate the limitations of both our ICL-AT theory and short-length AT.
>
> Nevertheless, we believe there is a potential solution to further improve our work by extending our current ICL-AT theoretical framework to the setting of "continuous-AT" [r4]. Specifically, continuous-AT [r4] is a more recent LLM AT method that has been shown to improve LLM robustness against both suffix-based and non-suffix-based jailbreak attacks. Since continuous-AT is implemented by adding adversarial perturbations to the continuous embeddings of input tokens, our ICL-AT theoretical framework is naturally easier to generalize to this setting. We will leave this generalization for future studies.
>
> **Q4: Could the authors provide more details or quantitative analysis regarding the computational efficiency of short-length AT compared to longer-length AT during training?**
>
> **A4:** Here, we present time costs of performing AT with different adversarial suffix lengths $M_{\text{train}}$ on Vicuna-7B in Table r3 below:
>
> | $M_{\text{train}}$ | 5     | 10    | 20    | 30    | 40    | 50    |
> | ------------------ | ----- | ----- | ----- | ----- | ----- | ----- |
> | Vicuna-7B          | 10.2h | 11.3h | 13.8h | 16.0h | 18.2h | 20.4h |
>
> *Table r3: Time costs of AT on Vicuna-7B.*
>
> From Table r3, we find that when the adversarial suffix length during AT is as long as $50$, the time cost of AT can reach around 20 hours, which is almost twice that when the suffix length is set to $20$. However, according to Figure 2 in our paper, AT with an adversarial suffix length of $20$ can already make Vicuna-7B achieve strong jailbreak robustness. These results clearly demonstrate the advantages of using short-length AT instead of long-length AT.
>
> The complete time-cost analysis for AT will be added in the revision.
>
> **References:**
>
> [r1] Chao et al. "Jailbreaking Black Box Large Language Models in Twenty Queries". arXiv 2023.
>
> [r2] Paulus et al. "AdvPrompter: Fast Adaptive Adversarial Prompting for LLMs". ICML 2025.
>
> [r3] Sadasivan et al. "Fast Adversarial Attacks on Language Models In One GPU Minute". ICML 2024.
>
> [r4] Xhonneux et al. "Efficient Adversarial Training in LLMs with Continuous Attacks". NeurIPS 2024.
>
> [r5] Yu et al. "Robust LLM safeguarding via refusal feature adversarial training". ICLR 2025.

---

> > ### Comment · Reviewer_q37y · 2025-08-04
> > **ack**
> >
> > Thanks for the authors' additional results, which addressed most of my concerns. I will consider raise my score during discussion with AC

---

> > > ### Author Response · Authors · 2025-08-04
> > >
> > > Dear Reviewer q37y,
> > >
> > > Thank you for your kind follow-up!
> > >
> > > We are glad that our rebuttal addressed your concerns. If you have any further questions, please let us know and we would be happy to provide clarification.

---

### Official Review · Reviewer_W4gv · 2025-07-06

**Clarity:** 3
**Significance:** 3
**Originality:** 3
**Rating:** 4
**Confidence:** 2

**Summary:**

This paper aims to address the challenge of defending LLMs against jailbreak attacks, a type of adversarial prompt designed to circumvent safety mechanisms and induce harmful behavior. Specifically, the authors focus on adversarial suffix jailbreaks and investigate how the length of adversarial prompts used during adversarial training (AT) affects the model’s robustness to such attacks.

The core contribution is a theoretical and empirical finding that adversarial training with shorter suffixes (length ∝ $\sqrt M$) is sufficient to defend against longer attack suffixes (length ∝ M). The authors support this with:

1. A theoretical analysis of in-context learning using linear transformers on linear regression tasks, yielding a robust generalization bound dependent on the ratio $\sqrt {M_{test}} / M_{train}$.

2. Empirical validation on five open-source LLMs using the GCG jailbreak attack, showing that shorter suffixes in AT significantly reduce attack success rates against longer adversarial prompts.

Overall, the paper introduces a computationally efficient strategy for enhancing LLM safety by demonstrating that short-length AT can generalize effectively to defend against long-length jailbreak attacks.

**Questions:**

None

**Ethical Concerns:**

["NO or VERY MINOR ethics concerns only"]

**Final Justification:**

issues resolved:
1. Agreed to shorten theoretical part for more experimental studies
2. Conducted more experiments to demonstrate the generalisation of the proposed methodI

issues unresolved:
1. Evaluate the performance on standard benchmark to demonstrate that the proposed defense doesn't hurt the generic capability of LLMs.
2. Focus only on suffix-based attacks thus not sure the generalisation of the proposed defense to other attacks.

**Limitations:**

Please refer to the weaknesses

**Quality:**

3

**Strengths And Weaknesses:**

**Strengths**

- The paper makes a strong theoretical contribution by establishing a novel generalization bound that rigorously connects adversarial training with in-context learning.

- The paper evaluates five widely used open-source LLMs across a range of adversarial suffix lengths under four different jailbreak attacks. In all cases, the results consistently demonstrate the effectiveness of the proposed defense strategy.

- Clear writing, solid structure, and well-motivated problem framing.

**Weaknesses**

- While the paper presents a rigorous and well-developed theoretical analysis, a significant portion of the main content is devoted to the theoretical framework, resulting in a relatively brief empirical section (approximately two pages). To improve balance and accessibility, some of the more detailed proof steps could be moved to the appendix, allowing more space for expanded experimental analysis and discussion in the main text.

- The experimental evaluation focuses primarily on GCG-based jailbreak attacks. While GCG is a strong and representative method, it would strengthen the work to demonstrate the effectiveness of the proposed defense against a broader range of jailbreak techniques (e.g., [1, 2, 3]). This would help confirm the generalizability of the findings.

- The paper does not provide a direct comparison between the proposed approach and existing defense mechanisms (e.g., [4, 5, 6]). Including such comparisons would clarify the relative strengths and potential trade-offs of the proposed method, particularly in terms of robustness and efficiency.

- The current evaluation uses ASR as the primary metric. However, in practice, LLMs may generate responses that engage with irrelevant content in the adversarial suffix rather than directly following the harmful instruction. Such cases could still constitute successful jailbreaks under the intended threat model. A more detailed analysis or clarification of how these borderline cases are handled would provide a fuller understanding of the defense’s real-world effectiveness.

References:

[1] Jailbreaking Leading Safety-Aligned LLMs with Simple Adaptive Attacks. Andriushchenko et al. 2024

[2] Tree of attacks: Jailbreaking black-box llms automatically. Mehrotra et al. 2023

[3] Query-based adversarial prompt generation. Hayase et al. 2024

[4] Smoothllm: Defending large language models against jailbreaking attacks. Robey et al. 2023

[5] Baseline defenses for adversarial attacks against aligned language models. Jain et al. 2023

[6] Improving alignment and robustness with circuit breakers. Zou et al. 2024

---

> ### Author Rebuttal · Authors · 2025-07-30
>
> Thank you for your thorough comments and kind support!
>
> **Q1: While the paper presents a rigorous and well-developed theoretical analysis, a significant portion of the main content is devoted to the theoretical framework, resulting in a relatively brief empirical section (approximately two pages). To improve balance and accessibility, some of the more detailed proof steps could be moved to the appendix, allowing more space for expanded experimental analysis and discussion in the main text.**
>
> **A1:** Thanks for your suggestion. We will shorten the theoretical analysis (Section 3) in the main text and add more experimental results to the main text.
>
> **Q2: The experimental evaluation focuses primarily on GCG-based jailbreak attacks. While GCG is a strong and representative method, it would strengthen the work to demonstrate the effectiveness of the proposed defense against a broader range of jailbreak techniques (e.g., [r1, r2, r3]). This would help confirm the generalizability of the findings.**
>
> **A2:** Thanks. Following your suggestion, we have added GCQ [r3], which is a suffix-based jailbreaking, as an additional jailbreak baseline. Preliminary results on the Vicuna-7B model are presented in Table r1 below:
>
> | $M_{\text{test}}$ (GCQ Attack [r3]) | 5 | 10 | 20 | 40 | 60 | 80 | 100 | 120 |
> | ---- | ---- | ---- | ---- | ---- | ---- | ---- | ---- | ---- |
> | Vicuna-7B (Original) | 85 | 95 | 92 | 96 | 95 | 97 | 96 | 94 |
> | Vicuna-7B + LLM AT ($M_{\text{train}}=5$)  | 7 | 12 | 15 | 17 | 21 | 20 | 19 | 19 |
> | Vicuna-7B + LLM AT ($M_{\text{train}}=20$) | 7 | 4 | 3 | 3 | 4 | 6 | 3 | 3 |
>
> *Table r1: ASR (%) of the GCQ attack [r3] against adversarially trained Vicuna-7B.*
>
>
>
> From the table, we find that short-length AT (where $M_{\text{train}} = 20$) can already reduce the ASR of the GCQ attack from around 90% to less than 10% in the worst case.
>
> Furthermore, we have also adopted a non-suffix-based jailbreak attack, named PAIR [r7], into our experiments. Results on Vicuna-7B are shown in the following Table r2:
>
> | $M_{\text{train}}$ | 0 (No AT) | 5 | 10 | 20 | 30 | 40 | 50 |
> | ---- | ---- | ---- | ---- | ---- | ---- | ---- | ---- |
> | Vicuna-7B + AT v.s. **PAIR Attack [r2]** | 84 | 53 | 48 | 42 | 50 | 44 | 55 |
>
> *Table r2: ASR (%) of PAIR attack against adversarially trained Vicuna-7B.*
>
> From Table r2, we find that while LLM AT with a short adversarial suffix length $M_{\text{train}}$ can reduce the ASR of the (non-suffix) PAIR attack from more than 80% to around 50%,but further increasing $M_{\text{train}}$ does not help much to improve LLM robustness against PAIR. We will leave the theoretical analysis of defending against non-suffix jailbreaking attacks for future studies.
>
> Complete experimental results of GCQ and PAIR attacks will be added in the revision.
>
> **Q3: The paper does not provide a direct comparison between the proposed approach and existing defense mechanisms (e.g., [r4, r5, r6]). Including such comparisons would clarify the relative strengths and potential trade-offs of the proposed method, particularly in terms of robustness and efficiency.**
>
> **A3:** Thanks. Following your suggestion, we have added “Circuit Breakers” [r6] as an additional LLM jailbreak defense baseline. Specifically, we adopted the robust Mistral-7B and Llama3-8B models officially released by [r6] in our analysis and performed various jailbreak attacks against these robust models. Preliminary experimental results are presented in Tables r4 and r5 below:
>
> | $M_{\text{test}}$ (GCG Attack)              | 5    | 10   | 20   | 40   | 60   | 80   | 100  | 120  |
> | ------------------------------------------- | ---- | ---- | ---- | ---- | ---- | ---- | ---- | ---- |
> | Mistral-7B + LLM AT ($M_{\text{train}}=20$) | 6    | 9    | 16   | 25   | 19   | 25   | 19   | 19   |
> | Mistral-7B + Circuit Breakers [r6]          | 21   | 20   | 21   | 23   | 23   | 28   | 28   | 23   |
> | Llama3-8B + LLM AT ($M_{\text{train}}=20$)  | 5    | 8    | 6    | 5    | 6    | 1    | 3    | 1    |
> | Llama3-8B + Circuit Breakers [r6]           | 3    | 5    | 3    | 4    | 3    | 5    | 5    | 7    |
>
> *Table r4: ASR(%) of GCG attack against different protection methods (i.e., LLM-AT and Circuit-Breaker [r6]).*
>
> | $M_{\text{test}}$ (BEAST Attack) | 5 | 10 | 20 | 40 | 60 | 80 | 100 | 120 |
> | ---- | ---- | ---- | ---- | ---- | ---- | ---- | ---- | ---- |
> | Mistral-7B + LLM AT ($M_{\text{train}}=20$) | 11 | 8 | 11 | 10 | 13 | 8 | 8 | 11 |
> | Mistral-7B + Circuit Breakers [r6] | 19 | 21 | 20 | 24 | 25 | 25 | 25 | 27 |
> | Llama3-8B + LLM AT ($M_{\text{train}}=20$)  | 10 | 12   | 4 | 13 | 12 | 21 | 19 | 15 |
> | Llama3-8B + Circuit Breakers [r6] | 12 | 9 | 11 | 12 | 16 | 15 | 17 | 15 |
>
> *Table r5: ASR(%) of BEAST attack against different protection methods (i.e., LLM-AT and Circuit-Breaker [r6]).*
>
> From the two tables, we find that when the base model is Mistral-7B, short-length LLM-AT achieves better jailbreak robustness than Circuit Breaker. Meanwhile, when the base model is Llama3-8B, the two defense methods achieve similar performance.
>
> We will add more baselines in the revision.
>
> **Q4: The current evaluation uses ASR as the primary metric. However, in practice, LLMs may generate responses that engage with irrelevant content in the adversarial suffix rather than directly following the harmful instruction. Such cases could still constitute successful jailbreaks under the intended threat model. A more detailed analysis or clarification of how these borderline cases are handled would provide a fuller understanding of the defense’s real-world effectiveness.**
>
> **A4:** We would like to clarify that the ASR calculation adopted in our paper is directly based on **whether the generated content contains harmful information**, but NOT on whether it follows the harmful input instruction. As a result, those borderline cases mentioned by the reviewer would not be a problem for our ASR calculation.
>
> Specifically, we use an open-source LLM-based judger [r8], originally proposed in [r9], to determine whether a given piece of LLM-generated content contains harmful information. This LLM-based judger does not take the input harmful instruction into account during the ASR calculation process. Please see Appendix B.3 in our paper for more details on the ASR calculation.
>
> We would also like to note that [r9] has shown that judgments made by this LLM-based judger align well with human judges. This judger has also been widely adopted for jailbreak robustness evaluation in many existing works, such as [r10], [r11], and [r12]. Therefore, we believe our ASR calculation is reliable enough to justify our finding that "short-length LLM AT can help defend against long-length jailbreaking".
>
> **References:**
>
> [r1] Andriushchenko et al. "Jailbreaking Leading Safety-Aligned LLMs with Simple Adaptive Attacks". ICLR 2025.
>
> [r2] Mehrotra et al. "Tree of attacks: Jailbreaking black-box LLMs automatically". NeurIPS 2024.
>
> [r3] Hayase et al. "Query-based adversarial prompt generation". NeurIPS 2024.
>
> [r4] Robey et al. "Smoothllm: Defending large language models against jailbreaking attacks". arXiv 2023.
>
> [r5] Jain et al. "Baseline defenses for adversarial attacks against aligned language models". arXiv 2023.
>
> [r6] Zou et al. "Improving alignment and robustness with circuit breakers". NeurIPS 2024.
>
> [r7] Chao et al. "Jailbreaking Black Box Large Language Models in Twenty Queries". arXiv 2023.
>
> [r8] https://huggingface.co/cais/HarmBench-Llama-2-13b-cls
>
> [r9] Mazeika et al. "HarmBench: A Standardized Evaluation Framework for Automated Red Teaming and Robust Refusal". arXiv 2024.
>
> [r10] Paulus et al. "AdvPrompter: Fast Adaptive Adversarial Prompting for LLMs". ICML 2025.
>
> [r11] Xhonneux et al. "Efficient Adversarial Training in LLMs with Continuous Attacks". NeurIPS 2024.
>
> [r12] Yu et al. "Robust LLM safeguarding via refusal feature adversarial training". ICLR 2025.

---

> > ### Comment · Reviewer_W4gv · 2025-08-05
> > **thanks for the additional experiments**
> >
> > Thank you for conducting the additional experiments. I recommend incorporating the new results into your revision. My evaluation remains unchanged.

---

> > > ### Author Response · Authors · 2025-08-05
> > > **Thank you for your feedback**
> > >
> > > Thank you very much for your feedback and kind support! We will include all additional results from the rebuttal in the revision as you suggested.

---

> ### Author Response · Authors · 2025-08-04
> **Follow-up on rebuttal**
>
> Dear Reviewer W4gv,
>
> Thanks for your time in reviewing our paper and valuable comments!
>
> Since the reviewer-author discussions is about to end, we are writing to kindly ask if our replies have satisfactorily addressed your concerns. Please kindly let us know if you have any further questions or if you require any additional clarification.
>
> Thank you very much!

---

### Note · Authors · 2025-08-13

We would like to thank again to the Area Chair for coordinating the reviewing process and all reviewers for giving valuable comments on our paper. All these will significantly help to improve the quality of our paper.

As the overall reviewing process is nearing its end, we would like to summarize the updates that will be made in our revision (some of the new results and discussions have already been presented in our rebuttal). Specifically, we will:

- Improve the presentation of the theoretical part in the main text and add more experimental results to the main text. (Reviewer W4gv)

- Add additional jailbreaking baselines: GCG attack [1], PAIR attack [2], and DeepInception attack [3]. (Reviewer W4gv, Reviewer q37y, and Reviewer gkox)

- Add additional jailbreak defense baselines: Circuit Breakers [4] and one additional defense. (Reviewer W4gv)

- Add experiments on LLMs larger than 7B/8B: Vicuna-13B and one additional larger LLM. (Reviewer q37y and Reviewer gkox)

- Add the time usage analysis of LLM AT. (Reviewer q37y)

- Discuss the connection between [5] and our work. (Reviewer 6htW)

- Analyze the robustness of LLMs that were adversarially trained via the BEAST attack. (Reviewer gkox)

We appreciate Reviewer W4gv, Reviewer 6htW, and Reviewer gkox for their positive support for our paper. We also appreciate Reviewer q37y for their willingness to reconsider their rating and respectfully hope that our revision will merit their support.

With best regards,

Authors

**References**

[1] Hayase et al. "Query-based adversarial prompt generation". NeurIPS 2024.

[2] Chao et al. "Jailbreaking Black Box Large Language Models in Twenty Queries". arXiv 2023.

[3] Li et al. "DeepInception: Hypnotize Large Language Model to Be Jailbreaker". arXiv 2023.

[4] Zou et al. "Improving alignment and robustness with circuit breakers". NeurIPS 2024.

[5] Wei et al. "Jailbreak and Guard Aligned Language Models with Only Few In-Context Demonstrations". arXiv 2023.

---

### Decision · Program_Chairs · 2025-09-17

**Decision:**

Accept (poster)

**Comment:**

This paper studies the effectiveness of short-length adversarial training for defending LLMs against long-length jailbreak attacks, providing both theoretical analysis and empirical validation of a quadratic relationship between training suffix length and model robustness.

**Strengths.** All reviewers acknowledge the mathematical analysis establishing a novel generalization bound for adversarial in-context learning under linear transformer frameworks. The paper demonstrates that short adversarial suffixes during training (e.g., 20 tokens) can effectively defend against significantly longer attack suffixes (up to 120 tokens), offering substantial training cost reductions.

**Weaknesses.**  All reviewers highlight the exclusive focus on suffix-based attacks as a major limitation, questioning generalizability to other prevalent jailbreak formats. The paper also misses direct comparisons with existing defense mechanisms, limiting assessment of relative performance and trade-offs.

Overall, the paper conducted an interesting study to understand adversarial training dynamics in LLMs and provides a mathematically rigorous framework, representing a meaningful contribution to the literature. Yet, its practical impact is constrained by the narrow focus on suffix-based attacks. The work would also benefit from broader experimental evaluation across attack types and model scales, along with comparative analysis against existing defenses to better establish its position in the defense landscape.